# Metabolic drug survey highlights cancer cell dependencies and vulnerabilities

Tea Pemovska [1,2], Johannes W. Bigenzahn[1,3], Ismet Srndic[1], Alexander Lercher [1,7], Andreas Bergthaler [1], Adrián César-Razquin[1], Felix Kartnig[1], Christoph Kornauth[2,4], Peter Valent [2,5], Philipp B. Staber[2,4] & Giulio Superti-Furga [1,6✉]

Interrogation of cellular metabolism with high-throughput screening approaches can unravel contextual biology and identify cancer-specific metabolic vulnerabilities. To systematically study the consequences of distinct metabolic perturbations, we assemble a comprehensive metabolic drug library (CeMM Library of Metabolic Drugs; CLIMET) covering 243 compounds. We, next, characterize it phenotypically in a diverse panel of myeloid leukemia cell lines and primary patient cells. Analysis of the drug response profiles reveals that 77 drugs affect cell viability, with the top effective compounds targeting nucleic acid synthesis, oxidative stress, and the PI3K/mTOR pathway. Clustering of individual drug response profiles stratifies the cell lines into five functional groups, which link to specific molecular and metabolic features. Mechanistic characterization of selective responses to the PI3K inhibitor pictilisib, the fatty acid synthase inhibitor GSK2194069, and the SLC16A1 inhibitor AZD3965, bring forth biomarkers of drug response. Phenotypic screening using CLIMET represents a valuable tool to probe cellular metabolism and identify metabolic dependencies at large.

[1] CeMM—Research Center for Molecular Medicine of the Austrian Academy of Sciences, Vienna, Austria. [2] Department of Medicine I, Division of Hematology & Hemostaseology, Medical University of Vienna, Vienna, Austria. [3] Department of Laboratory Medicine, Medical University of Vienna, Vienna, Austria. [4] Comprehensive Cancer Center Vienna, Vienna General Hospital, Medical University of Vienna, Vienna, Austria. [5] Ludwig Boltzmann Institute for Hematology and Oncology, Medical University of Vienna, Vienna, Austria. [6] Center for Physiology and Pharmacology, Medical University of Vienna, Vienna, Austria. [7] Present address: Laboratory of Virology and Infectious Disease, The Rockefeller University, New York, NY, USA. ✉email: GSuperti@cemm.oeaw.ac.at

Cellular metabolism represents a dynamic network of regulated pathways that is frequently reprogrammed in cancer and has been recognized as an emerging hallmark[1,2]. The first observations that transformed cells exhibit a distinct metabolic program came from Otto Warburg almost hundred years ago. The Warburg effect depicts the phenomenon of cancer cells preferentially undergoing glycolysis and converting carbon to lactate even in high oxygen conditions[3]. In cancer, genetic events activate signaling pathways that subsequently modulate cellular metabolism to satisfy the increased bioenergetic, biosynthetic and redox demands[4–6]. Moreover, this supports cancer initiation and progression and is typically accompanied with changes in expression of metabolic enzymes and transporters, which has implications for nutrient uptake, distribution of nutrients to pathways for biomass generation ultimately affecting therapy response[2,7,8]. Therefore, cancer-specific metabolic changes confer a selective advantage for survival, but also introduce metabolic liabilities that provide a unique opportunity for therapeutic targeting.

Considering the matter closer, it becomes apparent that there are different individual metabolic signatures of cancer cells. Transformed cells display greater flexibility in catabolite utilization, contingent on nutrient accessibility in the environment, cell lineage or tissue of origin, and genetic events[7,9,10]. Hence, despite glycolysis many more metabolic dependencies are altered to support cancer cell proliferation[2,8]. A greater insight in the metabolic differences between cancer and nonmalignant cells may lead to development of improved cancer therapies and ultimately patient outcomes. However, normal cells often physiologically activate metabolic pathways that are upregulated in cancer. Thus, identifying metabolic processes that offer sufficient therapeutic window and patients most likely to respond to a given therapy remains a challenge[9,11].

Obtaining a collection of chemical agents affecting different aspects of cancer metabolism would represent a toolbox allowing to interrogate cell lines, primary samples and animal models in a versatile mode and may suggest therapeutic strategies. Here, we assembled a custom metabolic drug library covering 243 compounds allowing to systematically identify metabolic dependencies in high-throughput phenotypic screens. We focused on myeloid leukemias as a model disease, as they are heterogeneous clonal malignancies for which there is a need for comprehensive understanding of disease-specific molecular mechanisms. Even though, there has been substantial advancement in mapping the genetic landscapes of AML, the first approvals for targeted agents came only in the last few years[12,13]. While the hematologists toolbox has increased, the survival of AML patients remains poor[14]. Moreover, prior studies have illustrated metabolic pe3culiarities in myeloid leukemias that can be exploited for either the development of novel therapies or for particular stratification rationales[15–21]. For instance, the development and clinical utility of IDH1/2 inhibitors for *IDH1/2* mutant AML[22–25] as wells as BCL-2 inhibitors[26] illustrates that untangling metabolic changes could provide therapeutic avenues in AML. Here, we profiled the metabolic drug library phenotypically in a panel of 15 diverse myeloid leukemia cell lines for drug-induced effects on cell growth and survival.

We functionally grouped the cell lines and drugs based on metabolic drug efficacy patterns and associated them with distinct genomic and metabolic attributes. Moreover, we identified a number of differential vulnerabilities, such as the sensitivity to the monocarboxylate transporter (MCT) SLC16A1 (also known as MCT1) inhibitor AZD3965, that could be rationalized by differential expression as well as target essentiality. Our data provides a primer for the use of a focused metabolic drug library in phenotypic screening platforms, thereby identifying metabolic vulnerabilities and pinpointing that targeted metabolic perturbations may provide promising new therapeutic strategies for myeloid leukemias and beyond.

## Results

**Assembly and characteristics of the metabolic drug library**. To systematically study the consequences of metabolic perturbations in healthy and disease-altered cell states, we assembled a comprehensive metabolic drug screening collection (the CeMM Library of Metabolic Drugs; CLIMET). While probing cellular metabolism could also be achieved with genetic screening technologies such as CRISPR, compound libraries are more readily applicable to challenging cell and tissue samples including primary patient cells and provide rapid result outputs. Furthermore, they can easily be combined with other interventions (e.g. other drugs, metabolites, genetic lesions) for higher-order perturbations[27]. CLIMET was compiled in a stepwise fashion, starting from 8000 candidate compounds, after a survey of public drug-target databases, and ending with 243 highly curated compounds after extensive crosschecking for approval status, structural information, compound's potency and selectivity for the intended target, pathway/target redundancy, and commercial availability (Fig. 1a; Supplementary Data 1).

Eighty-eight (36.2%) compounds are clinically approved for various disease indications allowing for rapid translation, roughly half (113; 46.5%) are chemical probes, 10 in clinical trials and 25 either withdrawn or their clinical development discontinued (Fig. 1b). CLIMET compounds target 191 different metabolic enzymes and/or processes with the most common drug-target categories being reductases, dehydrogenases, nonreceptor serine/threonine protein kinases, glycosidases, acetyltransferases, and carboxylesterases (Fig. 1c). The metabolic target space of these compounds was categorized to eight different processes with the majority targeting lipid and fatty acid (FA) ($n = 84$), protein and amino acid ($n = 43$), and glycolysis/sugar metabolism ($n = 39$) (Fig. 1d).

**Phenotypic characterization of CLIMET**. To assess the potency of CLIMET compounds, we screened the full collection against a panel of myeloid leukemia cell lines (10 AML with different disease-causing genetic alterations and 5 CML cell lines). Each compound was tested for its effect on cell growth and survival over a 10,000-fold concentration range (1 nM to 10 μM) enabling the generation of dose–response curves and area under the curve (AUC) calculation (Fig. 2a; Supplementary Data 2). From the 243 compounds, 77 (32%) markedly affected viability in at least one tested concentration. These 77 hits covered the eight metabolic processes, with autophagy (67%), nucleotide metabolism (52%), and oxidative stress (50%) having the highest hits to category process ratio (Supplementary Fig. 1a). We then ranked the compounds based on mean AUC values across all cell lines screened and found that among the top 25 most effective drugs were antimetabolites (e.g., gemcitabine, topotecan, antifolates), PI3K/mTOR pathway (e.g., rapalogs, Torin1, LY-294002), oxidative stress (e.g., daporinad, STF-31, ML175), acid–base balance (e.g., digoxin, digitoxigenin, indisulam), and energy homeostasis inhibitors (e.g., rotenone, CPI-0610) (Fig. 2b).

The average drug sensitivity profiles between AML and CML cell lines were highly concordant ($r = 0.78$), with the exception of CML cells being more sensitive to Torin1 (mTOR inhibitor), cytarabine (nucleoside analog), and rotenone (complex I inhibitor) (Supplementary Fig. 1b). Hence, the disease subtype did not play a significant role in the cell line clustering.

**Metabolic vulnerabilities facilitate functional taxonomy of cells and drugs**. To assess differences in drug sensitivity profiles, unsupervised hierarchical complete linkage clustering with

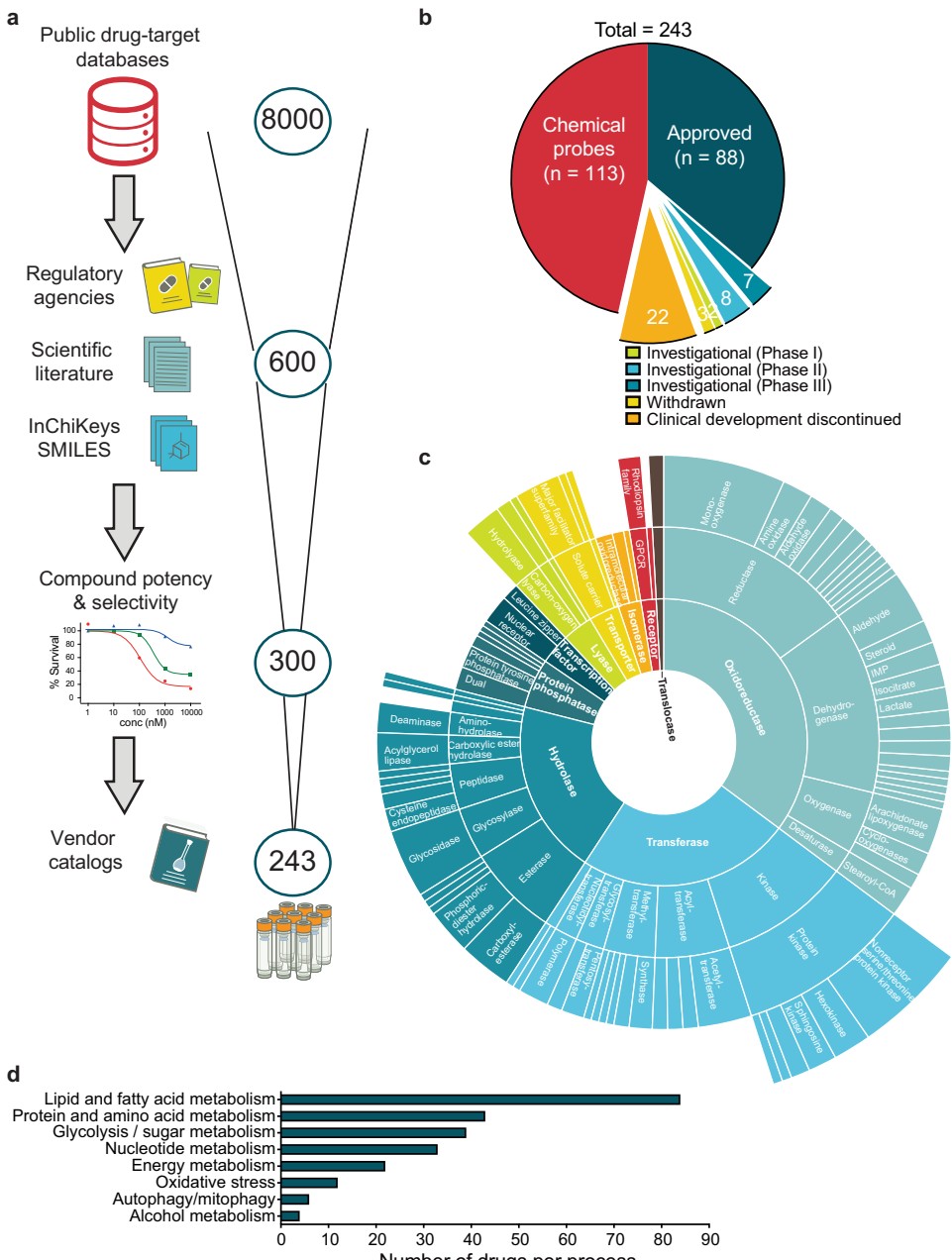

**Fig. 1 Workflow and features of the CeMM library of metabolic drugs (CLIMET). a** The process of library creation is shown, with compounds selected based on approval status, scientific literature, structural information, potency and selectivity toward the intended target, and commercial availability. Selected graphics were modified from Servier Medical Art, licensed under a Creative Common Attribution 3.0 Generic License, http://smart.servier.com/. **b** Pie chart depicting the approval status of the compounds included in the library. **c** Sunburst visualization of protein function hierarchy categorization with growing specificity of function relating to distance from the figure center and size of each segment comparative to the fraction of the library targeting each protein class. The library is enriched with compounds targeting oxidoreductases, transferases and hydrolases. **d** Number of compounds targeting different metabolic pathways and processes.

Spearman correlation distance measure of AUC was applied (Supplementary Fig. 2a). This facilitated grouping of cell lines based on drug efficacy patterns across all viability affecting drugs and taxonomy of compounds in terms of metabolic addictions in the context of myeloid malignancies. While each cell line had a unique metabolic vulnerability profile, the activity of 18 compounds in particular stratified the cells lines into five robust and distinct functional groups (Fig. 2c and Supplementary Fig. 2a, b). Consequently, the pathways targeted by the drugs shared in a specific group may shed light on the nature of the metabolic dependency.

Group I (MV4-11 and MOLM-13) was selectively sensitive to a cluster of compounds targeting processes and enzymes capable of producing reactive oxygen species such as cytochrome P450 oxidases (econazole), NADPH oxidases (GKT-136901), mitochondrial complex II (3-nitropropionic acid), and phosphodiesterases (PF-02545920) (Supplementary Fig. 2c). Moreover, Group I exhibited selective sensitivity to 5-fluorouracil (Supplementary Fig. 2d), lestaurtinib, GW 4064 (farnesoid X receptor agonist), and PI3K/mTOR inhibitors (Supplementary Fig. 2e, f). Group II (Mono-Mac-6, LAMA-84, KU-812, and KCL-22) exhibited no response to the PI3K inhibitor LY-294002 but showed

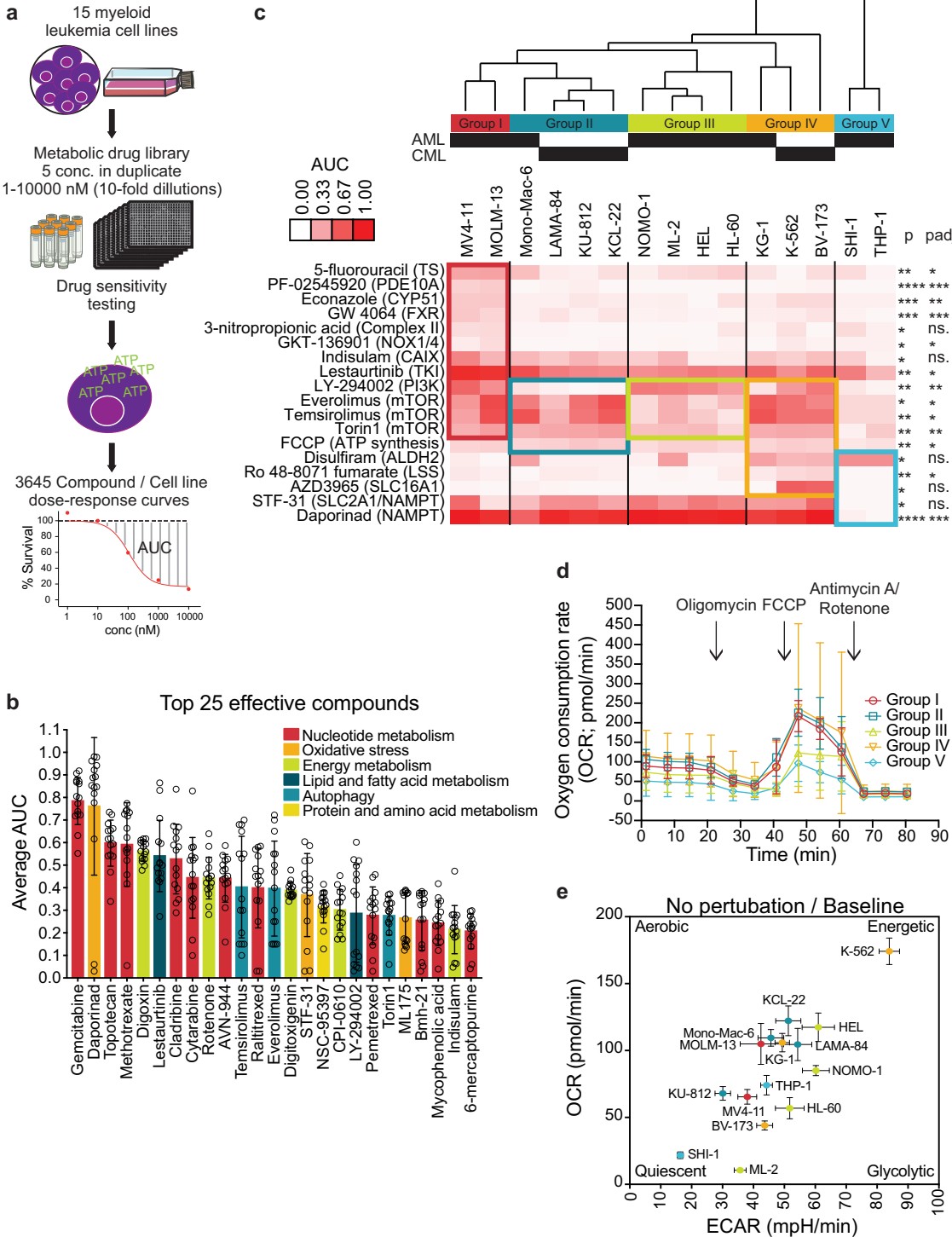

**Fig. 2 Stratification of myeloid cancer cells by metabolic vulnerabilities. a** Overview depicting the phenotypic characterization of the metabolic drug library. Selected graphics were modified from Servier Medical Art, licensed under a Creative Common Attribution 3.0 Generic License, http://smart.servier.com/. **b** Scatter dot plot showing the average top 25 effective compounds in the myeloid cell lines tested with each circle representing one cell line ($n = 15$ biologically independent cell lines in total) and bars labeled per metabolic process the corresponding compounds target. Graph shows mean ± SD. **c** Heatmap illustrating the functional grouping of myeloid cancer cells in five taxonomic groups based on metabolic vulnerability profiles. While each cell line had an overall distinct drug sensitivity profile, the activity of 18 compounds in particular significantly functionally stratified the myeloid cancer cells lines (an ordinary one-way ANOVA analysis *$P \leq 0.05$; **$P \leq 0.01$; ***$P \leq 0.001$; ****$P \leq 0.0001$; FDR of 10% was deemed significant *$P$adj $\leq 0.1$; **$P$adj $\leq 0.05$; ***$P$adj $\leq 0.01$; ns not significant). **d** Measurement of oxygen consumption rate using Seahorse analyzer at basal level and after consecutive injections of oligomycin (1 µM), FCCP (1 µM), and antimycin A/rotenone (1 µM) ($n = 8$ technical replicates for each cell line $n = 15$). Error bars indicate mean ± SD of the data of cell lines falling in the same group. **e** XF PhenoGram profile of each of the myeloid leukemia cell lines is shown by plotting the basal OCR and ECAR providing a snap-shot of the bioenergetics profiles of the cell lines. Cell lines are colored based on which group they fall in as shown in **c**. Data presented as mean ± SD; $n = 8$ technical replicates).

comparable sensitivity to rapalogs and the mTOR inhibitor Torin1 as Group I and increased vulnerability to the ATP synthesis disruptor FCCP. Group III (NOMO-1, ML-2, HEL, and HL-60) displayed a similar profile with respect to mTOR inhibitor sensitivity as Group I and II, albeit with lower intensity. In addition to the PI3K/mTOR inhibitor sensitivity, Group IV (KG-1, K-562, and BV-173) was characterized with selective sensitivity to disulfiram (alcohol dehydrogenase) (Supplementary Fig. 2g), the cholesterol biosynthesis inhibitor Ro 48-8071 fumarate, and the SLC16A1 inhibitor AZD3965. Group V (SHI-1 and THP-1) had a largely lower sensitivity pattern in comparison to all other groups with the exception of disulfiram, but could be specifically distinguished by displaying no sensitivity to NAMPT inhibitors STF-31 and daporinad (Supplementary Fig. 2h).

**Metabolic phenotypes of the myeloid cancer cell line panel**. To uncover the basis for these differences, we first decided to investigate the basal metabolic profiles of our cancer cell line panel. We determined the oxygen consumption rate (OCR) and extracellular acidification rate (ECAR) as proxies for mitochondrial respiration and glycolysis, respectively, by performing the Seahorse mitochondrial stress test (Supplementary Fig. 3a, b). Generally, cell lines in Groups I, II, and IV were predominately oxidative, whereas cell lines in Groups III and V were primarily glycolytic as assessed by differences in basal OCR, maximum respiration rate achieved, and the basal OCR/ECAR ratio (Fig. 2d; Supplementary Fig. 3c). Comparison of overall metabolic profiles (OCR vs. ECAR) showed that the examined cell lines have analogous baseline metabolic phenotypes with few exceptions, suggesting that the mitochondrial cell function as measured by Seahorse analysis did not significantly influence the cell line clustering (Fig. 2e). Evaluation of the spare respiratory capacity, as an indication for cellular fitness or flexibility implied that Group I and II had the highest energetic stress response capability consistent with the comparable drug response profiles detected (Supplementary Fig. 3d).

Since the mitochondrial stress test functionally evaluated basal metabolic activity of cells, it is likely that it did not fully capture the complete cellular metabolic profile. Thus, it is plausible to assume that the test output had impact only on sensitivity of inhibitors targeting energy metabolism and glycolysis. From the effective compounds affecting cellular energetics, indisulam, digitoxigenin, rotenone, and FCCP had a significant stronger effect in predominantly oxidative cell lines (Groups I, II and IV) in comparison to largely glycolytic cell lines (Groups III and V) (Supplementary Fig. 3e). To gain a deeper understanding of the metabolic wiring of the cell lines included in this study, we analyzed a targeted metabolomics dataset from the Cancer Cell Line Encyclopedia (CCLE) resource[28]. A liquid chromatography–mass spectrometry (LC–MS) approach was applied on 225 metabolites in 928 cell lines from more than 20 different cancer types. Data were available for 13 of the 15 cell lines and the levels of 57 metabolites were found to be significantly different between at least two groups using a two-way ANOVA analysis (Supplementary Fig. 4). The abundance of specific metabolites could be linked to the drug sensitivity defined groups, suggesting that the similarities in functional phenotypes of the cell lines in our analysis may be anchored to particular metabolic vulnerabilities.

Specifically, Group I had higher levels of several metabolites involved in the pentose phosphate pathway, which is largely responsible for generation of reducing equivalents such as NADPH, and production of intermediates for nucleic and amino acid synthesis. This is in accordance with the metabolic drug sensitivity profile and the more oxidative phenotype detected.

Group II showed enrichment in metabolites involved in arginine and proline metabolism and the urea cycle. The metabolites defining Group III were primarily involved in tryptophan metabolism, whereas alpha-glycerophosphocholine was the most abundant metabolite in Group IV. Taken together, the identified cell line groups have distinct metabolic phenotypes, which suggest different nutrient acquisition dependencies that could be associated to the drug sensitivity profiles.

**Metabolic coregulation revealed by genotype to phenotype associations**. Next, we asked whether the metabolic phenotypes of the cell lines tested within the study reflected their mutational status. The most recurrent mutational events in the cell line panel were collected from the Cancer Dependency Map portal[29] (Fig. 3a). In addition, AML cell lines were annotated with the French–American–British (FAB) classification system[30]. Group I harbored activating mutations in *FLT3*, *MLL* fusions and are classified as AML M5; Group II was enriched for loss of function *TP53* mutations and three of the four cell lines were CML and thus had *BCR-ABL1* fusions; Group III was characterized by activating mutations in genes involved in growth factor signaling such as *RAS* or *JAK2*, whereas no clear pattern was detected for Group IV. Finally, Group V carried activating mutations in *RAS*, inactivating mutations in *TP53*, *MLL* fusions, and are classified as AML M5. The mutational status did not considerably drive the functional stratification of the cell lines. Systematic comparison of mutant vs. wild-type cases (where at least three mutant cell lines could be identified), revealed statistically significant novel and previously known correlations between *FLT3* mutations and sensitivity to 5-fluorouracil, lestaurtinib[31,32], and PF-02545920. Moreover, *CREBBP* mutations linked to sphingosine kinase inhibitor (N,N-dimethylsphingosine) sensitivity, whereas *RAS* and *TP53* mutations conferred a resistance phenotype especially to energy and lipid metabolism inhibitors such as PI3K/mTOR and mitochondrial respiration inhibitors (Fig. 3b; Supplementary Fig. 5a–p). Several of these associations could be confirmed in an independent AML patient sample dataset[33] (Fig. 3c). This analysis revealed metabolic and genotype relationships, which might imply that cells adapt their metabolic programs driven by activation or inactivation of particular signaling events or, in a different causal relationship, that particular metabolic enzyme and nutrient transporters are among the targets of oncogenic pathways[5,6,34,35].

**Co-occurring drug sensitivities and mutual exclusivity**. Co-occurring and mutually exclusive drug sensitivities were detected by pair-wise Spearman's correlation analysis of the 77 active drugs (Fig. 4a). Inhibitors with related mode of action in general correlated highly, such as antimetabolites, mTOR, inosine-5′-monophosphate dehydrogenase, NAMPT, and FA metabolism inhibitors (Supplementary Fig. 2a; Fig. 4a) demonstrating the robustness and reproducibility of the obtained dataset. Nevertheless, we also observed unforeseen correlations with the response to Bmh-21, a DNA-directed RNA polymerase I subunit inhibitor. Bmh-21 inversely correlated with the response to two inhibitors targeting different aspects of mitochondrial respiration, 3-nitropropionic acid and MKT 077 (that positively correlated among each other). Conversely, the sensitivity to the FA synthase (FASN) inhibitor GSK2194069 correlated with sensitivity to CP-640186 (targeting acetyl-CoA carboxylase (ACC) also involved in FA synthesis), pictilisib (PI3K inhibitor), and as1949490 (targeting phosphatidylinositol 3,4,5-trisphosphate 5-phosphatase 2 (SHIP2)). This suggested convergence on the FA synthesis and PI3K pathways. Moreover, the drug sensitivity profile of rosiglitazone (peroxisome proliferator-activated receptor gamma

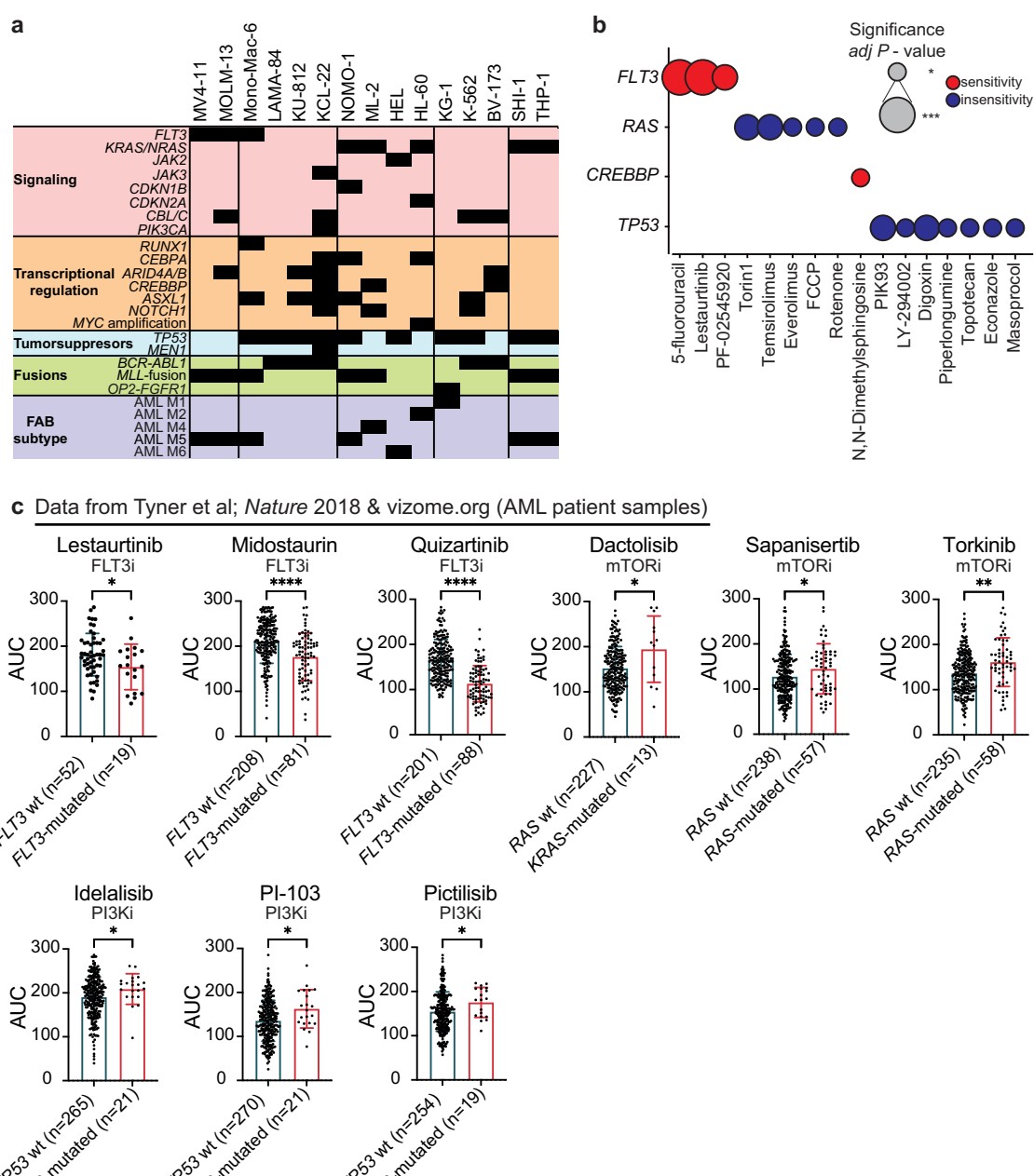

**Fig. 3 Metabolic coregulation revealed by genotype to phenotype associations. a** Summary of the most recurrent mutational events as well as FAB subtype annotation of the cell line panel included in the study. **b** Bubble plot describing the significant genotype to phenotype associations discerned with a two-sided equal variance *t*-test between mutant and wild-type cases. The size of each bubble corresponds to adjusted *P* value significance level with FDR of 10% (*Padj ≤ 0.1; **Padj < 0.05; ***Padj < 0.01). **c** Validation of selected phenotype to genotype associations (where data available) from the Beat AML study[33] (data retrieved from http://www.vizome.org/aml/). Box plots showing sensitivity of several metabolic modifiers that have significantly higher or lower sensitivity in *FLT3*, *RAS*, and *TP53* mutant vs. wild-type AML patient samples. The number of independent samples for each comparison is indicated in the figure. Higher values, here, indicate lack of sensitivity and lower values indicate sensitivity. Error bars signify mean ± SD and the difference in response was assessed with either a two-tailed unpaired *t*-test or Mann–Whitney test (*P ≤ 0.05; **P ≤ 0.01; ***P ≤ 0.001; ****P ≤ 0.0001).

agonist) matched the effect of zinterol hydrochloride (β₂-adrenoceptor agonist; triggers arachidonic acid release) and dorsomorphin (AMPK inhibitor). These findings imply that the activity of these compounds in the setting of myeloid malignancies could be functionally related.

To investigate the identified drug interactions further we performed drug combinatorial screens in four different cell lines (HL-60, Mono-Mac-6, BV-173, and MV4-11) that best captured the significant drug–drug correlations. The results showed that in majority of cases combinatorial effects were observed as hypothesized (Fig. 4b and Supplementary Fig. 6). However, there were a few drug combination pairs (e.g., lestaurtinib and 5-fluorouracil, and dorsomorphin and rosiglitazone) that exhibited marked single agent activity but an antagonistic combinatorial effect. Taken together this analysis could facilitate elucidation of the context-specific molecular mechanisms of action and foster a rationale for context-dependent drug combinations, though alternative bases for the observed interactions should also be contemplated.

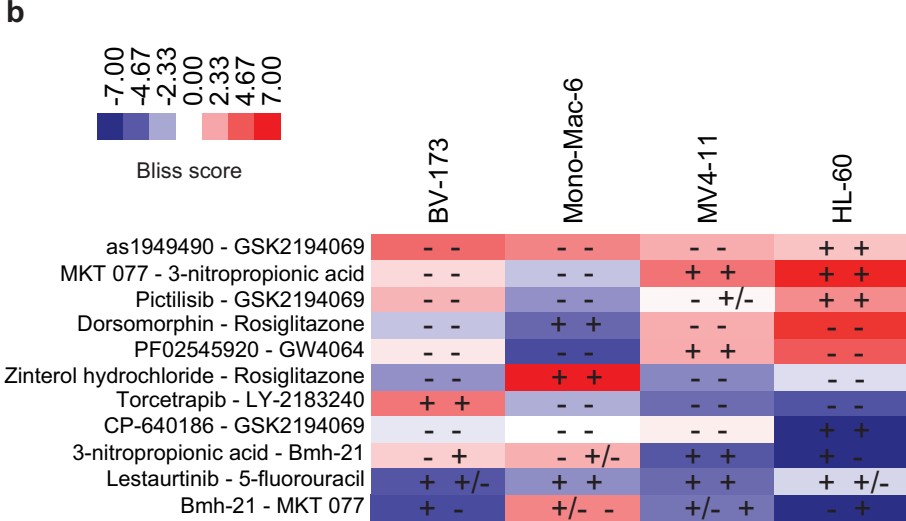

**Functional linkage of PI3K signaling and de novo FA synthesis.** To identify cell line-specific metabolic drug sensitivities, we compared the AUC scores in each individual cell line with the average AUC score per drug (selective AUC; sAUC) (Fig. 5a). For instance, HL-60 cells showed strong selective response to the pan-PI3K inhibitor (higher potency toward α and δ isoforms)

pictilisib (Fig. 5b) in line with data from the Genomics of Drug Sensitivity in Cancer portal (GDSC1)[36], where pictilisib was tested across 937 cell lines and HL-60 scored as 20th most sensitive cell line (Supplementary Fig. 7a). However, among AML and CML cell lines, HL-60 was one of the most sensitive lines (Supplementary Fig. 7b). Exploration of genome-wide CRISPR

**Fig. 4 Co-occurring drug sensitivities reveal metabolic coregulation. a** Corrplot visualization of Spearman's correlation analysis of each drug against each other plotted onto a clustered heatmap, exposing drugs with similar or dissimilar patterns of sensitivity. This analysis revealed metabolic processes and/or targets that were highly concordant (red) and ones that are quite discordant (blue). The size and color of the dots is based on the Spearman correlation coefficient. Significant positive and negative correlations were identified after correcting for multiple testing at FDR of 10% and are marked with thick black border. Targets of compounds exhibiting significant drug–drug associations are shown to the right. **b** Heatmap showing the deviation from Bliss independence score for each tested combination and cell lines. Synergy is denoted in red while antagonism is shown in blue. Analysis was performed using the SynergyFinder tool[82,83]. The individual drug sensitivity is shown within the heatmap squares per cell line with + indicating sensitivity; +/− indicating moderate sensitivity, and − indicating no sensitivity. The first sign refers to the sensitivity of the first listed drug of the combination and the second sign to the sensitivity of the second drug.

dropout screening results[37] revealed that HL-60 cells are dependent on *PIK3C2A* (Fig. 5c). This is in accordance with previous genome-wide RNAi screens assessing cancer cell line genetic dependencies estimated using the DEMETER2 model[38]. Here, among 648 cell lines, HL-60 was one of the cell lines most dependent on *PIK3C2A* (Supplementary Fig. 7c). Comparison of publicly available gene expression data of class I and II PI3K family members, showed that with the exception of *PIK3CB* and *PIK3CD*, HL-60 cells had an overall lower expression level of PI3K family members (Fig. 5d). Thus, we initially hypothesized this to likely explain the vulnerability of HL-60 cells to PI3K inhibition as pictilisib is not effective in inhibiting PIK3C2A[39].

To explore determinants to pictilisib sensitivity, we overexpressed the different class I PI3K isoforms in HL-60 and MV4-11 cells, which have a comparable expression pattern of those genes. Pictilisib sensitivity was enhanced in HL-60 cells overexpressing *PIK3CA, PIK3CB* or *PIK3CD* in comparison to HL-60 cells lentivirally transduced with an empty vector (Supplementary Fig. 7d). In contrast, we detected a moderate increase in pictilisib sensitivity in *PIK3CA* and *PIK3CG* MV4-11 overexpressing cells, while there was no difference in sensitivity in *PIK3CB* and a minor loss in sensitivity in *PIK3CD* overexpressing cells (Supplementary Fig. 7e). These findings suggest that our original postulate was incorrect and that there may not be one specific PI3K isoform that influences sensitivity to pictilisib in these cells. Rather, it is likely that the combined activity of several isoforms may determine pictilisib susceptibility. Comparison of gene essentiality[37], gene expression[40], reverse phase protein array (RPPA)[29], and drug sensitivity data[36] in HL-60 cells and cluster I cell lines (MV4-11 and MOLM-13) revealed that HL-60 cells exhibit higher dependency and activity on the MAPK pathway and mTORC2 complex signaling resulting in reduced sensitivity to AKT and mTOR inhibitors (Supplementary Fig. 7f–i). These features corresponded with lower gene expression and RPPA signal of several mTORC1 complex members and its downstream effectors, suggesting that mTORC1 activation status influences sensitivity to pictilisib as shown previously in the context of breast cancer[41,42].

Besides the susceptibility to pictilisib, HL-60 cells were also selectively sensitive to GSK2194069 (Fig. 5e). While no specific genetic dependency on the *FASN* gene could be established, HL-60 cells appear to rely on a number of genes involved in lipid metabolism and FA metabolism (Supplementary Fig. 8a; data[37]). Varied gene expression analysis identified 49 most differently expressed genes in relation to the gene expression profile of the remaining 13 cell lines (Fig. 5f). Nine genes were upregulated and enriched in alpha-defensins and neutrophil degranulation (Fig. 5g), whereas 40 genes were downregulated and primarily involved in FA metabolism and biosynthesis of unsaturated FA (Fig. 5h; Supplementary Fig. 8b). Although, C75 and orlistat, first generation FASN inhibitors, did not show an effect in the original drug screen, retesting of the respective compounds at an increased dose led to a viability reduction, whereas stearoyl-CoA desaturase 1 inhibitors exhibited limited activity in HL-60

cells (Supplementary Fig. 8c). We, further, detected a combinatorial effect of pictilisib and GSK2194069 in HL-60 cells (Supplementary Fig. 6). Moreover, pictilisib exposure led to a dose-dependent reduction of *FASN* expression (Supplementary Fig. 8d). Taken together, our data illustrate that HL-60 are dependent on PI3K signaling and de novo FA synthesis and conceivably these pathways and processes are functionally linked in this cellular and disease context.

**The sensitivity AZD3965 is dependent on differential expression of SLC16A1 and SLC16A3.** Two CML cell lines, K-562 and BV-173, exhibited a unique vulnerability to the SLC16A1 transporter inhibitor AZD3965 (Fig. 6a). Genome-wide CRISPR dropout screening data[43,44] revealed that *SLC16A1* was essential in K-562 cells, while HEL, MOLM-13, MV4-11 or THP-1 cells were not affected by genetic *SLC16A1* inactivation (Supplementary Fig. 9a, b). Comparison of *SLC16A1* and *SLC16A3* expression levels indicated that K-562 and BV-173 cells had markedly lower expression of *SLC16A3* mRNA (Fig. 6b). Thus, inhibition of SLC16A1 by AZD3965 may provide additional selective pressure, raising the possibility that this dependency on SLC16A1 may represent a proper Achilles' heel for these cells. Hence, changes in the expression of SLC16A1 and its paralog SLC16A3 should differ in their modulatory ability on AZD3965 action.

We first overexpressed *SLC16A1* or *SLC16A3* in K-562 and BV-173 cells and tested AZD3965 sensitivity (Fig. 6c and Supplementary Fig. 9c). While overexpression of *SLC16A3* conferred resistance to AZD3965, overexpression of *SLC16A1* did not impact the sensitivity to AZD3965 (Fig. 6d; Supplementary Fig. 9d). We also performed the opposite experiment and genetically inactivated *SLC16A1* or *SLC16A3* using CRISPR technology in previously insensitive cell lines, MV4-11 (Fig. 6e) and LAMA-84 (Supplementary Fig. 9e). Knockout of *SLC16A3* re-sensitized MV4-11 and LAMA-84 cells partially to AZD3965, whereas knockout of *SLC16A1* did not have any effect (Fig. 6f; Supplementary Fig. 9f). These experiments illustrated the selective targeting of SLC16A1 by AZD3965 and the synthetic lethal relationship between SLC16A1 and SLC16A3. Evaluation of differentially expressed genes in the AZD3965 sensitive and not sensitive cell lines identified 17 genes that were significantly upregulated and 12 genes that were significantly downregulated including *SLC16A3* (Fig. 6g). Analysis of *SLC16A1/SLC16A3* expression ratio in 205 hematological cancer cell lines, identified 44 cell lines, primarily of lymphoid origin, that are predicted to be sensitive to AZD3965. This cutoff was made as K-562 cells, shown here to be sensitive to AZD3965, had the lowest *SLC16A1/SLC16A3* expression ratio among those 44 cell lines (Methods; Fig. 6h and Supplementary Fig. 9g).

Our findings show that the combined expression of *SLC16A1* (high) and SLC16A3 (low/no) may serve as an actionable biomarker for AZD3965 response in hematological malignancies. To explore the potential clinical value of this biomarker, we turned to The Cancer Genome Atlas (TCGA) gene expression dataset accessed via the UCSF Xena project[45]. Utilizing the gene expression profiling interactive analysis server (GEPIA2)[46], we

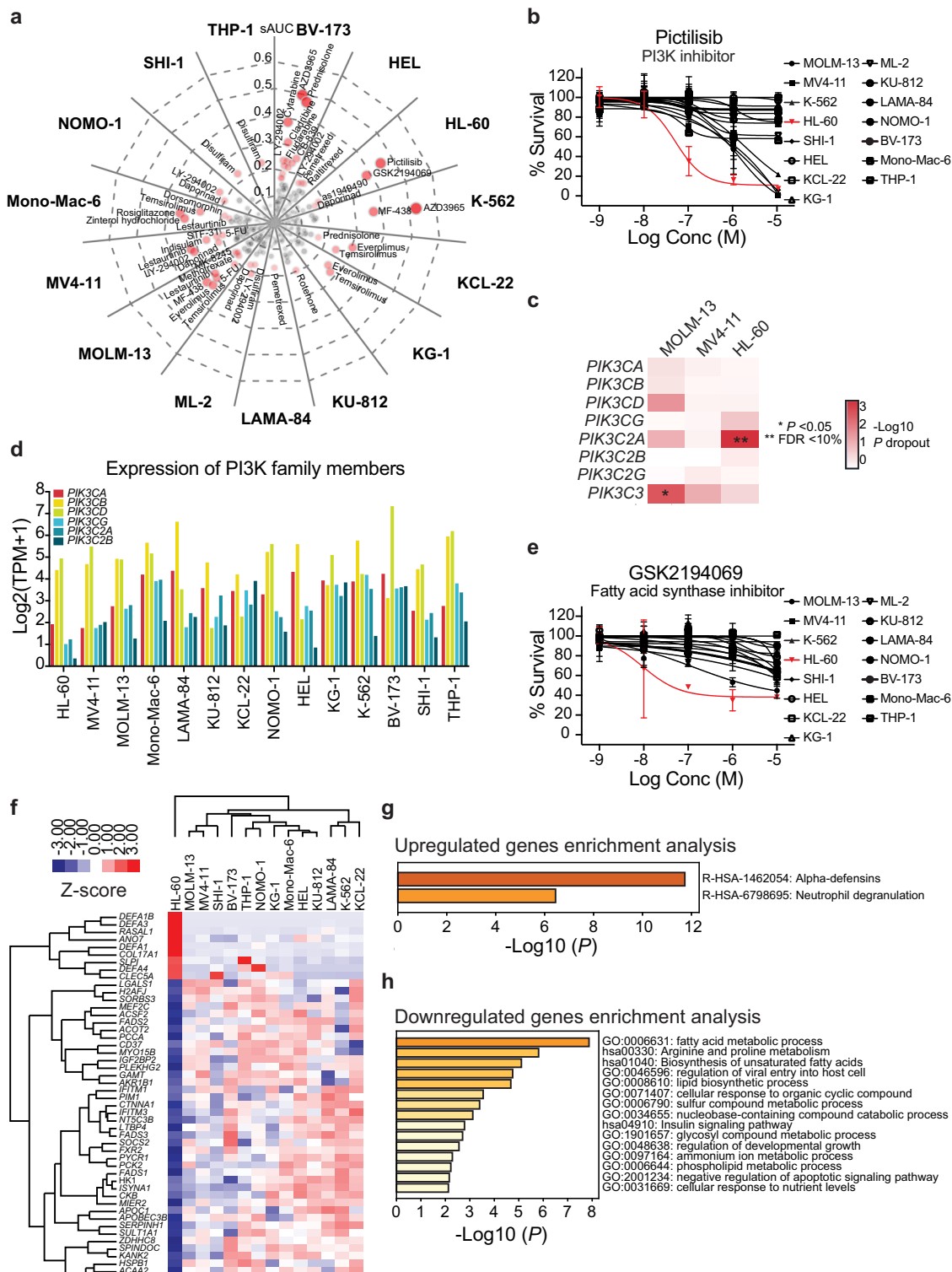

assessed the expression pattern of *SLC16A1* and *SLC16A3* in cancer patients in comparison to normal samples (TCGA-matching and the Genotype-Tissue Expression project[47]) (Fig. 7a, b). For instance, *SLC16A1* is significantly upregulated in diffuse large B-cell lymphoma (DLBCL) and downregulated in AML, while the opposite is true for *SLC16A3*. Moreover, the expression of *SLC16A1* and *SLC16A3* is negatively correlated on a pan-cancer and AML-specific level (Fig. 7c–e), indicating that in a subset of cancer patients the expression pattern of these MCTs is divergent and suggestive of eligibility for AZD3965 therapy or

other inhibitors specific for one of the two transporters. Thus, this case study demonstrates that functional drug testing with metabolic modifiers is a powerful approach to identify metabolic vulnerabilities without the need of gene editing technologies and facilitate uncovering of disease-specific drug mode of action.

**Metabolic vulnerabilities can be altered by chemical perturbation.** To determine whether metabolic vulnerabilities can be influenced by drug exposure we re-tested the entire drug library in HL-60 cells in the presence of vehicle (DMSO) or 100 nM

**Fig. 5 Determinants of sensitivity to pictilisib and GSK2194069. a** Spider plot summarizing the selective responses to metabolic modifiers in each cell line by comparing the average AUC values to ones detected in each individual cell line (sAUC). Identified hits are ranked according to sAUC value (cutoff ≥ 0.16), which is also represented with bubble size and color intensity. **b** Average dose–response data of pictilisib in the myeloid leukemia cell lines expressed as percentage survival ($n = 15$ biologically independent cell lines in technical duplicate of each concentration). The sensitivity of pictilisib in HL-60 cells is highlighted in red. Error bars indicate SD. **c** Depletion of PI3K family member genes in MOLM-13, MV4-11, and HL-60 cells highlighting that HL-60 cells are dependent on *PIK3C2A*. Data (*P* values and FDR-based multiple testing corrections) was retrieved from Tzelepis et al. [37]. **d** Level of gene expression (expressed as log2 transcripts per million (TPM+1)) of PI3K family members in publicly available datasets from the Dependency Map Portal[29,80]. **e** Average dose–response data of GSK2194069 expressed as percentage survival ($n = 15$ biologically independent cell lines in technical duplicate of each concentration). The sensitivity of GSK2194069 in HL-60 cells is highlighted in red. Error bars indicate SD. **f** Clustering of the most variedly expressed genes in HL-60 cells (*Z* score normalized expression values) in comparison to the remaining 13 cells lines for which gene expression data were available in the Dependency Map Portal. **g, h** Pathway and process gene set enrichment analysis of the upregulated and downregulated genes in HL-60 cells. *P* values were derived with a hypergeometric test and Benjamini–Hochberg *P* value correction algorithm with Metascape[84].

AZD3965, pictilisib, or GSK2194069 for 72 h (Fig. 8a, b). Even though HL-60 do not exhibit sensitivity to AZD3965 at baseline, treatment with AZD3965 resulted in reduced sensitivity to the mitochondrial complex I inhibitor rotenone and increased sensitivity to antifolates, PI3K pathway and oxidative stress inhibitors. This finding suggests that upon SLC16A1 inhibition the cells undergo a metabolic switch thereby becoming more reliant on glycolysis for their metabolic needs and SLC16A3 for lactate transport. Moreover, in response to pictilisib exposure we detected an increased sensitivity to a number of inhibitors targeting lipid metabolism (TOFA, Ro 48-8071 fumarate, perhexiline, caffeic acid phenethyl ester, SKI II, lestaurtinib) and nucleotide metabolism (cladribine and cytarabine). While we did not observe any pronounced difference in drug sensitivity in response to FASN inhibition, these results indicate that drugs can modify the metabolic phenotype of cells and this could serve as a plausible mechanism of drug resistance.

**Comparison of metabolic pharmacological profiles between myeloid cell lines and patient samples.** To evaluate the representativity of the myeloid cell lines of primary patient cells ($n = 15$; Supplementary Table 1), we first assessed differential drug responses of 70 metabolic inhibitors (Supplementary Data 3). We detected significantly higher efficacy of numerous nucleotide metabolism inhibitors in the cell lines (Fig. 9a), likely as a result of higher cell proliferation during the drug screening assay as compared to the primary patient cells. In contrast, patient samples were more susceptible to several lipid and FA metabolism drugs (Fig. 9a). Overall, the metabolic vulnerability profiles between patient samples and cell line were comparable (Spearman's $r = 0.5716$) with ~64% of compounds not exhibiting a differential response (Fig. 9a, b). These included 14 (77.8%) out of the 18 inhibitors that strongly contributed to the metabolic functional stratification of the cell lines depicted in Fig. 2c (Fig. 9b).

Unsupervised hierarchical clustering of the metabolic drug responses in the 15 different patient samples, also stratified the patient samples in five different functional groups (Fig. 9c). The patient sample groupings were not generally driven by age, diagnosis, or mutational status. Analogously to the cell line grouping, the clustering gave an indication of drug-target distinctiveness or relatedness (e.g., clustering of digoxin and digitoxigenin, nucleotide analogs, rapalogs, PI3K, or oxidative stress inhibitors). From the 70 screened compounds, only 5 did not exhibit a response in any of the patient samples, whereas 4 (Bmh-21, lestaurtinib, daporinad, and STF-31) were active in all cases. Thus, the majority of compounds exhibited sample/ group selective responses, suggesting associations with particular metabolic dependencies. Taken together, these data show that metabolic drug response profiling may decode disease-relevant biology and targeting cellular metabolism may be a

complementary therapeutic strategy for a subset of myeloid leukemia patients.

## Discussion

We describe the generation and functional characterization of a custom, focused metabolic drug library, CLIMET, as a powerful tool to identify cell-specific metabolic dependencies. While previous efforts to functionally profile cancer cell vulnerabilities, such as the CCLE[48] and GDSC[36,49], have primarily focused on assessing the sensitivity to growth factor signaling, epigenetic, and cell cycle inhibitors, our study provides a new layer by specifically characterizing metabolic drug responses. Very few previous studies have presented a broad pharmacologically focused characterization of metabolic dependencies of cancer cells[28,50]. The availability of clinical and preclinical metabolism-modifying agents provides an opportunity to functionally profile cancer cells and uncover metabolic liabilities that could be of high therapeutic value.

Cellular metabolism, while plastic in nature, cannot be escaped and must be considered in the development of treatment strategies. However, one of the biggest questions for modern cancer drug development is how to select the patients most likely to benefit to a given therapy. Here, we illustrated the use of the CLIMET metabolic perturbation library for the identification of metabolic susceptibilities, grouping of cancer cells based on metabolic dependencies, and understanding context-dependent mechanism of drug action. Drug screening represents an orthogonal approach to genetic methods in assessing cellular vulnerabilities and provides the opportunity to target whole enzyme groups that can circumvent cellular redundancy, often an issue in CRISPR/Cas9-based genetic screens. The metabolic drug sensitivity profiles generated in a panel of myeloid leukemia cell lines highlight several effective metabolic modifiers such as GSK2194069, AZD3965, and PI3K/mTOR inhibitors that could be clinically explored for a subset of myeloid leukemia patients.

The tested cell lines were functionally grouped in five taxonomic groups that could further be associated with somatic mutational patterns as well as cellular metabolic profiles. For instance, cell lines harboring activating mutations in the *RAS* genes (groups III and group V) were predominantly glycolytic and displayed lack of sensitivity to mTOR and mitochondrial respiration inhibitors in line with previous reports that *RAS*-mutated cancers display a Warburg effect phenotype exemplified by increased glucose utilization and lactate production, and upregulation of the glucose transporter SLC2A1[51–54]. While some genotype to phenotype links were identified, for the majority of drug responses no clear genotype associations could be established, in accordance to prior studies showing a better association between multiple molecular datasets and drug sensitivities[49,55,56]. The sensitivity of the PI3K inhibitor pictilisib could be associated with lower activation of the mTORC1 complex, as

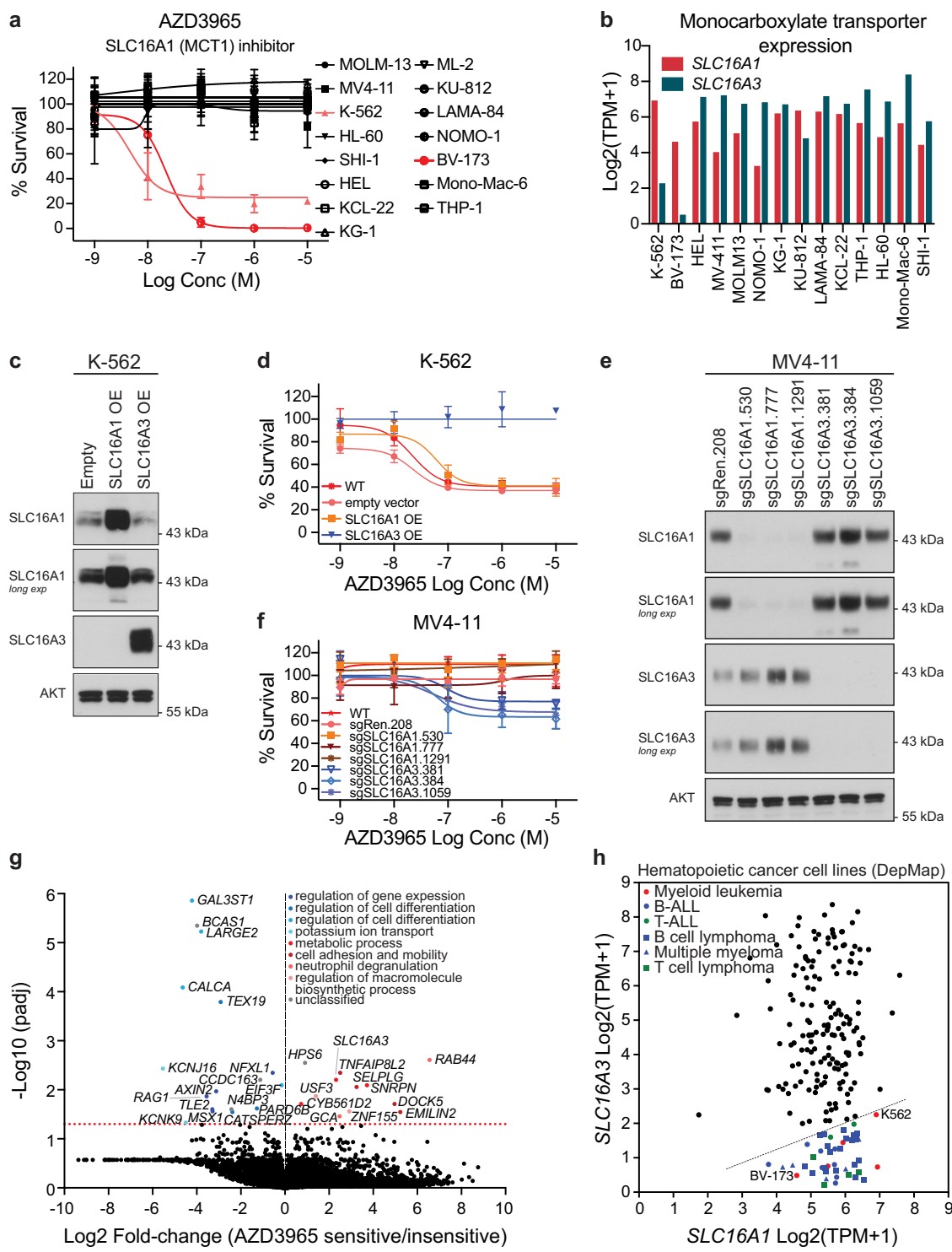

mTORC1 activation has been described to be a crucial event in the resistance to PI3K inhibitors in many tumor types[41,42]. A recent study found that signaling models based on multiplex single-cell mass cytometry data were more predictive of pictilisib sensitivity than gene expression alone in breast cancer cell lines[57]. We, further, detected a significant positive correlation between the FASN inhibitor, GSK2194069, and pictilisib, which also showed a combinatorial effect. This finding suggests a functional link between PI3K signaling and de novo FA synthesis. In breast cancer cells synergistic responses between rapamycin and FASN inhibition have been reported[58]. The PI3K/AKT pathway could stimulate the expression of enzymes necessary for FA synthesis

and promote activation of ATP-citrate lyase, an enzyme that catalyzes the production of acetyl-CoA from cytoplasmic citrate[59,60]. PI3K signaling in known to induce lipogenic enzymes, including FASN[61]. Cancer cells with upregulated PI3K signaling have been linked to increased FASN expression levels and enhanced glucose uptake thus fulfilling the increased need for membrane synthesis of rapidly dividing cells[62,63].

We also observed a differential sensitivity pattern to AZD3965, which could be rationalized by dependency on SLC16A1 in sensitive cell lines due to near or complete absent expression of the functionally related transporter SLC16A3. Hence, inhibition of SLC16A1 in cells lacking SLC16A3 led to cell death due to the

**Fig. 6 Differential expression of *SLC16A1* and *SLC16A3* determines sensitivity to AZD3965. a** Average dose–response data of AZD3965 expressed as percentage survival ($n = 15$ biologically independent cell lines in technical duplicate of each concentration). The sensitivity of AZD3965 in the two sensitive CML cell lines, K-562 and BV-173, is highlighted in red. Error bars indicate mean ± SD. **b** Level of gene expression of *SLC16A1* and *SLC16A3* in publicly available datasets from the Dependency Map Portal[29,80]. **c** Immunoblot analysis of SLC16A1 and SLC16A3 expression in K-562 cells lentivirally transduced with empty vector, *SLC16A1* or *SLC16A3*-cDNA. Data are representative of two independent experiments. **d** Average response to AZD3965 in K-562 wild type, empty vector, and *SLC16A1* or *SLC16A3* overexpressing cells. Error bars indicate standard deviation of the mean of technical triplicates. Data are representative of two independent experiments. **e** Expression of *SLC16A1* and *SLC16A3* in MV4-11 AML cells transduced with the indicated sgRNAs. Data are representative of two independent experiments. **f** Average response to AZD3965 in MV4-11 wild type, sgRen control infected cells, and *SLC16A1* or *SLC16A3* deficient cells. Error bars indicate standard deviation of the mean of technical triplicates. Data are representative of two independent experiments. **g** Volcano plot showing differences in gene expression between AZD3965 sensitive (K-562 and BV-173) and nonsensitive lines (the remaining 12 for which gene expression data were available). Each point represents one gene. An unpaired two-sample, two-sided *t*-test was performed for each gene. The *y*-axis corresponds to FDR adjusted *P* values (using Benjamini–Hochberg correction). The horizontal red dotted line signifies the 5% FDR cutoff. **h** A correlation plot of *SLC16A1* and *SLC16A3* expression in the 205 hematological cancer cell lines covered by the Dependency Map from the Broad Institute (expression data 20Q2)[40]. The diagonal dotted line indicates an *SLC16A1/SLC16A3* expression ratio of 3 or higher highlighting cell lines predicted to be sensitive to AZD3965 ($n = 44$) that are annotated per lineage and disease subtype. BV-173 and K-562 are labeled for reference. The names of the remaining cell lines and the corresponding *SLC16A1/SLC16A3* expression ratio are shown in Supplementary Fig. 9g.

synthetic lethal relationship of these transporters and functional redundancy[64]. Similar observations have been made in small cell lung cancer, DLBCL, and Burkitt lymphoma[65,66]. The MCTs are bidirectional and execute proton-linked transport of lactate, pyruvate, and ketone bodies across the plasma membrane thereby playing a crucial role in preventing intracellular acidification[67,68]. Importantly, several cancers use lactate as a metabolic energy source[69,70] and MCTs have been reported to impact cancer cell growth and survival[71]. Thus, as our data also show, MCTs represent attractive potential therapeutic cancer targets. Several SLC16A1-specific inhibitors have been developed[72] with AZD3965 being in phase I clinical trials for advanced cancers (NCT01791595). The structural basis for its specificity and mode of action was also recently identified[73]. Our study illustrates that a relatively small percentage of myeloid malignancies (10–13%) would be susceptible to SLC16A1 inhibition, but highlights phenotypic drug screening as attractive means of identifying responding cells/patients harboring metabolic dependencies.

While the genetic diversity of cancers is substantial, changes in numerous oncogenes and tumor suppressor genes stimulate a limited number of metabolic changes converging on a metabolic phenotype. In fact, previous studies have linked lack of drug response or the presence of drug resistance to specific intrinsic molecular and/or metabolic cellular properties[21,74], that could guide the exploitation of novel metabolic vulnerabilities by alternative chemical agents. Nonetheless, any therapy success of metabolic inhibitors would be contingent on an adequate integrative patient stratification. Several limitations of our study should be contemplated for future metabolic drug profiling efforts. First, the inactivity of roughly 70% of CLIMET compounds might be due to the concentration range used as some compounds are known to have activity in higher concentrations. Second, we may have missed out on interesting compounds that affect cellular metabolism by not impacting cell viability. Third, we envisage a wider collection of metabolic inhibitors would create insights into larger array of metabolic pathways/processes. Lastly, the composition of cell culture media is likely distinct from in vivo settings and in future, this parameter would need to be assessed more carefully. Nonetheless we uncovered functional and clinically relevant determinants of metabolic drug response. In-depth characterization of cancer metabolic dependencies and differences between normal and cancer cells will be a crucial step toward translating metabolic inhibitors to the clinic.

## Methods

**Cell lines and cell culture.** MOLM-13, MV4-11, Mono-Mac-6, KG-1, SHI-1, ML-2, NOMO-1, HEL, LAMA-84, KU-812, KCL-22, and BV-173 were obtained from

DSMZ, K-562 and HL-60 from NCI 60 panel and THP-1 from ATCC. All cell lines were maintained in RPMI 1640 medium (Gibco) with 10% fetal bovine serum (Gibco), penicillin (100 U/mL), and streptomycin (100 μg/mL). Cell lines were authenticated with STR profiling and mycoplasma tested by PCR prior to the study.

**Assembly of metabolic drug library.** Information on metabolic enzymes and drug-target annotations were integrated from publicly available databases such as Kyoto Encyclopedia of Genes and Genomes (KEGG), the Small Molecule Pathway Database (SMPDB) and ChEMBL target database (filtering on compounds with potency < 5 μM $k_D$ or $IC_{50}$) together with PubChem commercial chemical vendor records, which led to initial 8000 compound candidates (Fig. 1a). We then narrowed down this list to 600 by selecting compounds in order of preference for approved drugs, compounds in pre/clinical investigation, compounds tested in cell-based assays, and compounds tested in biochemical assays. The compound with the highest activity with drug-like structure was taken (1–3 leads per target). The compound list was further refined by literature evidence for biophysical interaction (X-ray, SPR, etc.), in vitro activity on isolated enzyme, in vitro activity in cell-based systems, in vivo activity in animal models, clinical activity, and number of PubMed publications, which gave us 300 candidate compounds. We then cross-referenced our candidates with the Fox Chase Cancer Center Cellular Metabolic library, metabolic inhibitors outlined by Martinez-Outschoorn et al.[9], and commercial vendor records to the final list of 243 compounds. The compounds were manually annotated with CHEMBL IDs, database/literature-reported target information, target CHEMBL IDs, approval status, the metabolic process they inhibit or stimulate, molecular weight, and SMILE (simplified molecular-input line-entry system) information. The target annotation was performed by collecting information from publicly available databases (e.g., DrugBank, KEGG) and additional manual curation. The compounds were ordered via the Sigma-Aldrich Marker Select service and sourced at 10 mM concentration in DMSO directly on Echo 384-well plates (LP-0200; Labcyte). In addition, we obtained 1 mL of 10 mM stock in vials for future use. The detailed description of the compounds is depicted in Supplementary Data 1.

**Phenotypic characterization of the metabolic drug library.** The optimal cell number per well for each cell line was assessed for cell viability analysis in 384-well plates (Corning 3764). Briefly, cells were suspended in RPMI 1640 and seeded in increasing concentrations in triplicate (500 cells/well to 25,000 cells/well) and cell viability was assessed with CellTiter-Glo® Luminescent Cell Viability Assay (Promega) after 72 h incubation at 37 °C according to the manufacturer's instructions. The highest cell number at which the growth curve was in linear range was chosen for drug testing. The compounds were printed on tissue culture-treated 384-well plates (Corning) using Echo 550 (Labcyte Inc.) in five different concentrations in duplicate in tenfold dilutions encompassing a 10,000-fold concentration range (1–10,000 nM). Twenty-five microliters of single-cell suspension were seeded to each well with a Multidrop Combi peristaltic dispenser (Thermo Scientific). The plates were incubated at 37 °C and 5% $CO_2$ for 72 h after which cell viability was measured using CellTiter-Glo in a multilabel plate reader (EnVision, PerkinElmer). Data were normalized to DMSO wells serving as negative controls and 10 μM bortezomib wells (a proteasome inhibitor; effectively killing all cells at this dose) serving as a positive control. A four-parameter log-logistic model was fit for each compound's dose–response data using the drc[75] R package and corresponding AUC values calculated for each compound using the compute AUC function of the PharmacoGx[76] R package. Unsupervised hierarchical clustering of the active compounds (ones that had an effect on viability in at least one tested concentration; $n = 77$) was performed with Gene Cluster 3.0 software (complete linkage method and Spearman (drugs) and Euclidean (cell lines) distance measures). The resulting

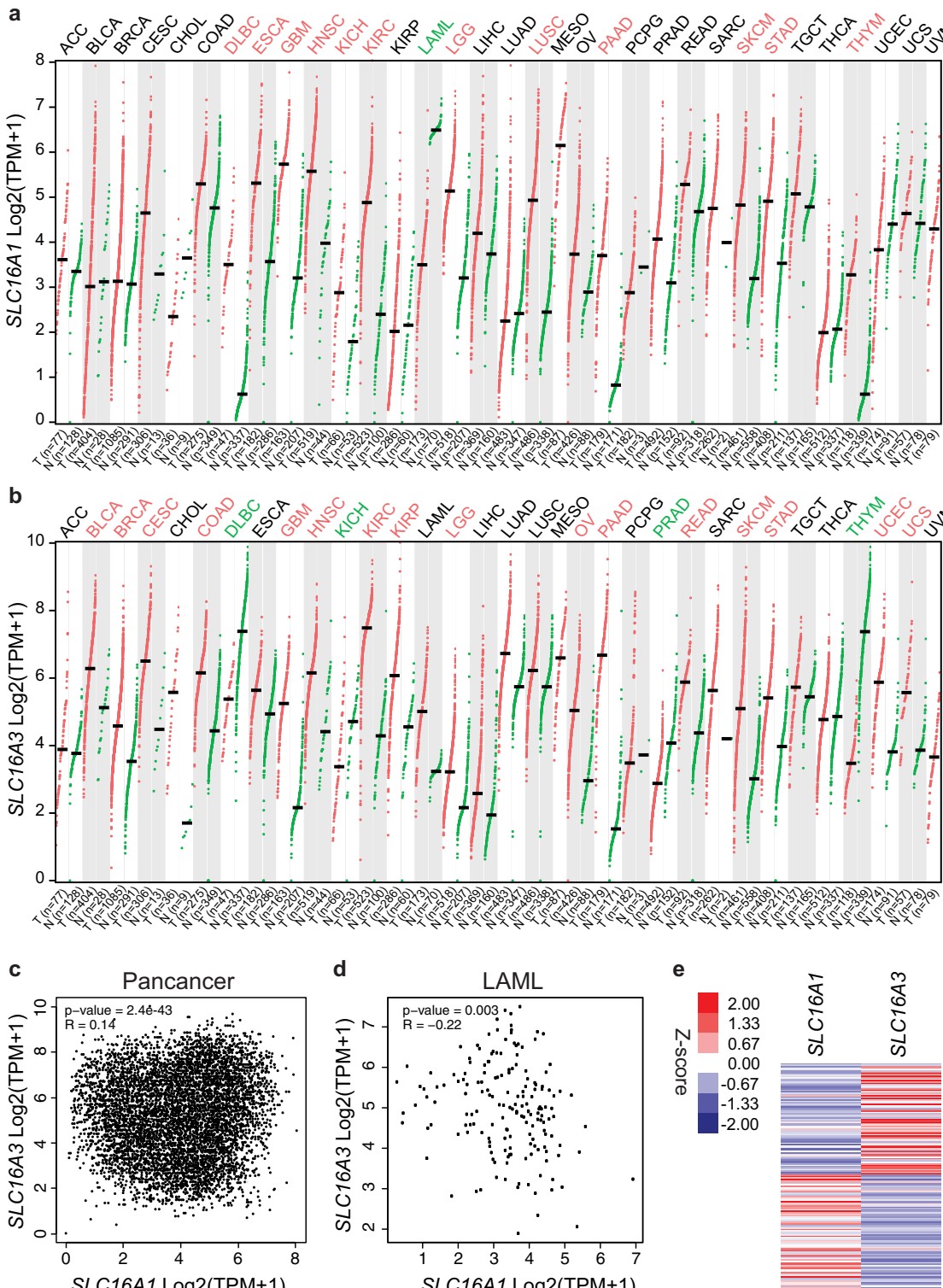

**Fig. 7 Gene expression pattern of monocarboxylate transporters in cancer patients. a, b** Comparison of *SLC16A1* and *SLC16A3* expression in cancer patient and normal samples (derived from matching TCGA or Genotype-Tissue Expression (GTEx) project[47]. The analysis was performed with the gene expression profiling interactive analysis (GEPIA2) server[46] utilizing RNA-Seq datasets based on the UCSC Xena project[45] (http://xena.ucsc.edu), which are computed by a standard pipeline. Disease entities highlighted in red indicate significantly higher expression in cancer patients, whereas highlighted in green indicate significantly higher expression in normal samples (log2 fold-change cutoff 1; *q* cutoff 0.01). The median gene expression is shown. Abbreviations details for the cancer types are available on the GEPIA server website http://gepia.cancer-pku.cn/index.html. LAML indicates Acute Myeloid Leukemia TCGA dataset. **c** Spearman correlation analysis of *SLC16A1* and *SLC16A3* expression across 33 cancer types (TCGA data) performed with the GEPIA2 server. GEPIA uses the non-log scale for calculation and the log-scale axis for visualization. **d, e** Spearman correlation analysis and heatmap of *SLC16A1* and *SLC16A3* expression in AML patient samples (data from the AML TCGA study[91]). For the heatmap *Z* score normalized expression data were used.

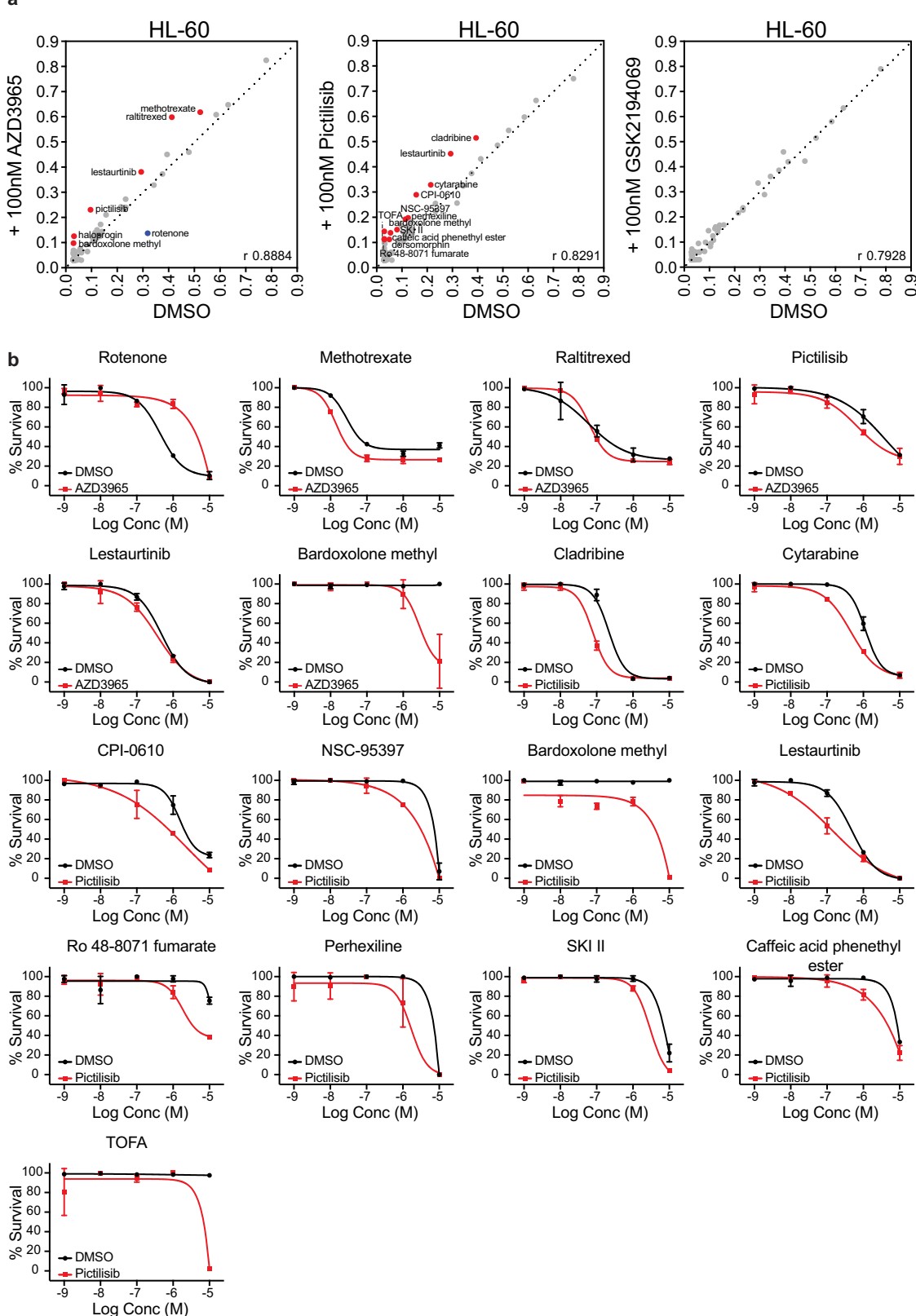

**Fig. 8 Metabolic inhibitors can alter metabolic dependencies. a** Spearman correlation of drug responses in HL-60 cells in response to 72 h 100 nM AZD3965, 100 nM pictilisib, or 100 nM GSK2194069 treatment quantified as AUC values. Drugs that showed a reduced sensitivity in response to drug treatment are marked in blue, whereas drugs that showed an increased sensitivity are marked in red. **b** Dose–response curves of the drugs highlighted in **a** as percentage survival. Black curves are baseline and red curves are in response to the respective drugs. Error bars indicate standard deviation of the mean of technical duplicates.

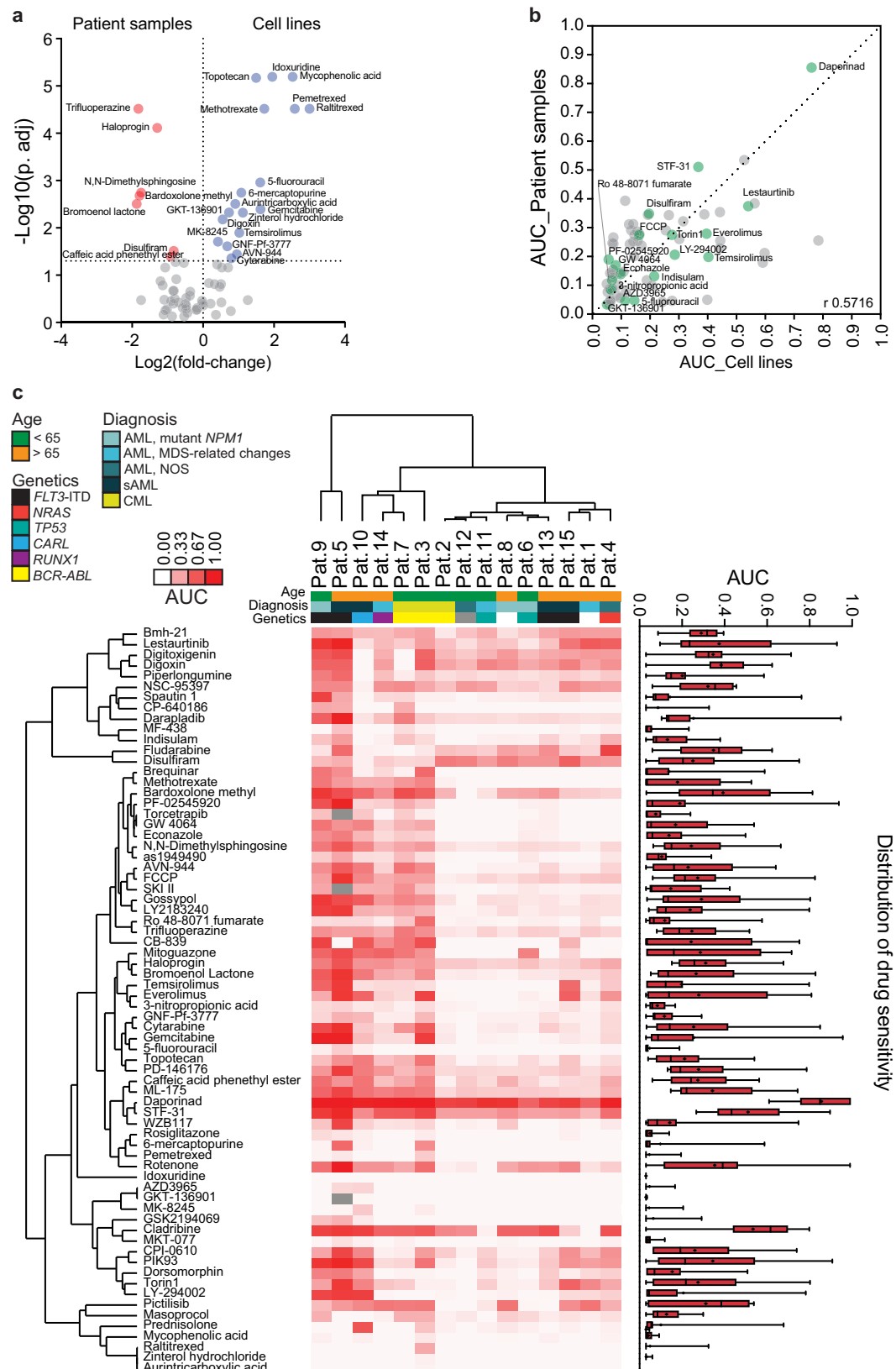

heatmap was visualized with Java Tree View software Version 1.1.6r4. Reproducibility of the clustering was assessed with the bootstrap resampling method using the Pvclust R package[77]. Similarity of drug sensitivity was evaluated by SD of AUC values for a particular compound within a cluster with SD ≤ 0.05 indicating concordance. The results were visualized with the Venn diagrams with the Venn webtool from the University of Gent (http://bioinformatics.psb.ugent.be/webtools/Venn/). For detecting selective drug sensitivities in each cell line, the average AUC per drug

was subtracted from the AUC value of the drug in each individual cell line and visualized with a spider plot using R statistical environment.

Evaluation if metabolic inhibitors can alter metabolic vulnerabilities was performed by rescreening the entire drug library as described above (each drug tested in five different concentrations in duplicate) in HL-60 cells in the presence of either DMSO, 100 nM AZD3965, 100 nM pictilisib, or 100 nM GSK2194069. Data analysis was the same as for the original screen with overall drug response profiles

**Fig. 9 Metabolic drug sensitivity profiles are generally comparable between myeloid cancer patient samples and cell lines. a** A volcano plot depicting the comparison of 70 drug responses between 15 myeloid cancer patient samples and 15 myeloid cancer cell lines. Each point represents one drug and statistical significance was assessed by Mann–Whitney test for each drug. The log2 AUC (fold change) is plotted on the x-axis and negative log10 of the adjusted *P* values (using the two-stage step-up Benjamini, Krieger, and Yekutieli correction) is plotted on the y-axis. The horizontal dotted line signifies the 5% significance cutoff. Colored dots indicate drugs that exhibit higher sensitivity in either patient samples (red) or cell lines (blue). **b** Spearman correlation of drug response profiles between myeloid leukemia patient samples and cell lines. Drugs that were highlighted to strongly contribute to the cell line grouping are marked in green. **c** Heatmap of metabolic drug sensitivity profiles expressed as AUC in 12 AML and 3 CML patient samples. Clustering was performed with the complete linkage method and Spearman (drugs) and Euclidean (patient samples) distance measures. The color bars to the left and above show sample annotations with gray box depicting data not available and white box depicting no mutation detected. Drug AUC distribution (min to max) box plot is shown on the right side of the graph with the black line in the middle of box depicting the median (the mean is indicated with a + sign).

being compared with a Spearman correlation analysis performed in GraphPad Prism 9.

**Cellular metabolic (Seahorse) analysis**. To assess basal cellular metabolic activity, we utilized the Seahorse XF Cell Mito Stress Test Kit (Agilent) according to the manufacturer's instructions. Briefly, cells suspended in Seahorse XF base medium, supplemented with glucose (10 mM), sodium pyruvate (1 mM) and L-glutamine (2 mM), were plated at 150,000 cells per well in XF 96-well cell culture microplates (Agilent). After 1 h of incubation at 37 °C the plate was loaded into the XF96 Extracellular Flux Analyzer (Agilent) and the Seahorse XF Cell Mito Stress Test was performed using oligomycin (1 μM), FCCP (1 μM) and rotenone (0.5 μM).

**Metabolomics analysis**. Publicly available targeted metabolomics dataset from the CCLE[28] was utilized for assessment of the metabolic profiles of the cell lines included in this study. A LC–MS approach was used to measure 225 metabolites in 928 cell lines from more than 20 cancer types. For analysis the imputed batch-corrected dataset with log10-transformed values (clean data) was used. To determine metabolite levels that were significantly different between at least two groups, a two-way ANOVA analysis and multiple testing correction with false discovery rate (FDR) of 10% was performed with GraphPad Prism version 9. The results were normalized with a Z score for a heatmap visualization. Metabolite enrichment and pathway analysis was performed with MetaboAnalyst[78,79].

**Genotype to phenotype correlation**. Somatic mutation data for the cell lines included in this study was obtained from The Cancer Dependency Map portal (DepMap; 19Q3 data release)[29,80]. For each recurrent mutation in at least three cell lines an unpaired two-sided parametric Student's *t* test was performed to identify significant associations between mutations and drug response. FDR was computed using the Benjamini–Hochberg method over all drugs (an FDR of 10% was deemed statistically significant).

**Drug response correlations**. Co-occurring and mutually exclusive drug sensitivities were identified by pair-wise Spearman's correlation analysis using the corplot[81] R package. The analysis was applied for the 77 active drugs and the Spearman's correlation coefficient of each drug was plotted onto a clustered heatmap visualized with a correlogram. Positive correlations are showed in red and negative in blue. The intensity of the color and size of the circles are proportional to the correlation coefficients. The significant drug–drug correlations were identified after correcting for multiple testing with Benjamini–Hochberg at FDR of 10% and highlighted with thick black border.

**Drug combination screens**. The significant drug–drug interactions identified in Fig. 4a were evaluated by drug synergy screens in four different cell lines HL-60, Mono-Mac-6, BV-173, and MV4-11. Briefly, 8 × 8 synergy matrices were tested for each drug combination and cell lines with majority of drugs being tested at 5, 10, 50, 100, 500, 1000, and 5000 nM concentration with the exception of rosiglitazone and lestaurtinib that were tested at 1, 5, 10, 50, 100, 500, and 1000 nM concentration. The experimental procedure and readouts were as described above for single agent screening. Combinatorial effects were evaluated and visualized using the SynergyFinder[82,83] web portal (https://synergyfinder.fimm.fi) using the Bliss independent model.

**Gene expression analysis**. Gene expression data for the cell lines included in this study were obtained from The Cancer Dependency Map portal (DepMap; 19Q3 and 20Q2 data release)[29,40,80] as log2 (TPM+1). Data were available for 14 of the 15 cell lines included in this study. Genes that were not expressed in any of the cell lines of interest (expression level log2 (TPM) < 1) were excluded from further analysis. Most variedly expressed genes in HL-60 cells were identified by assessing the SD between gene expression levels in HL-60 in relation to the average gene expression across the remaining 13 cell lines. Moreover, genes for which the SD was above 1.7 were considered and additionally filtered for unique expression

pattern in HL-60 cells. Pathway and process gene set enrichment analysis was performed with Metascape[84].

Differentially expressed genes in AZD3965 sensitive and nonsensitive cell lines were identified by performing an unpaired two-sample, two-sided *t*-test for each expressed gene. The resulting *P* values were adjusted with a Benjamini–Hochberg correction with FDR of 5% deemed significant. A volcano plot was generated with Prism 8. Biological process and molecular function gene annotation was retrieved from the gene ontology resource[85,86]. An *SLC16A1/SLC16A3* expression ratio of 3 or higher was considered predictive of AZD3965 sensitivity, as K-562 cells were tested to be sensitive in this study but had the lowest *SLC16A1/SLC16A3* expression ration of 3.1 among the 44 cell lines deemed sensitive to AZD3965. Thus, the threshold cutoff was placed at *SLC16A1/SLC16A3* expression ratio of 3. Lineage and disease subtype of the cell lines was retrieved from the DepMap portal (20Q2 data release).

**Plasmid cloning**. *SLC16A1* and *SLC16A3* deficient cells were generated by CRISPR/Cas9 technology as previously described[87] using the pLentiCRISPRv2 vector (Addgene plasmid #52961)[88]. In brief, guide RNAs (sgRNA) were designed using CHOPCHOP[89] and sgRNA designer tools[90], oligonucleotides harboring *BsmBI* restriction site-compatible overhangs were annealed, phosphorylated and ligated into pLentiCRISPRv2 by standard cloning methods and sequences were confirmed by Sanger sequencing. An sgRNA targeting the Renilla luciferase coding sequence:

sg*Ren*.208 (F: CACCGGTATAATACACCGCGCTAC; R: AAACGTAGCGCGGTGTATTATACC) was used as a negative control. Three different sgRNAs per gene were used and the sequences are provided bellow:

sg*SLC16A1*.530 (F: CACCGGGATATCCATGACACTTCGC; R: AAACGCGAAGTGTCATGGATATCCC)

sg*SLC16A1*.777 (F: CACCGTTTCTACAAGAGGCGACCAT; R: AAACATGGTCGCCTCTTGTAGAAAC)

sg*SLC16A1*.1291 (F: CACCGATGGTAGCCCGACCATCTAT; R: AAACATAGATGGTCGGGCTACCATC)

sg*SLC16A3*.381 (F: CACCGTGCCGGCCCGTCATGCTTGT; R: AAACACAAGCATGACGGGCCGGCAC)

sg*SLC16A3*.384 (F: CACCGCGGCCCGTCATGCTTGTGGG; R: AAACCCCACAAGCATGACGGGCCGC)

sg*SLC16A3*.1059 (F: CACCGTCTACGGCGGGCGACTACGG; R: AAACCCGTAGTCGCCCGCCGTAGAC).

Generation of cells overexpressing *SLC16A1*, *SLC16A3*, *PIK3CA*, *PIK3CB*, *PIK3CD*, and *PIK3CG* was performed by lentiviral cDNA delivery experiments. SLC-encoding cDNAs were obtained as codon-optimized versions from Genscript and transferred into pDONR221 entry vector using BP reaction gateway cloning (Invitrogen). PI3K isoform cDNAs were ordered from Addgene (#81736, #82221, #82222, and #81843).

In the case of Addgene plasmid #81843 the stop codon was removed by site directed mutagenesis to enable C-terminal tagging (E0554S, New England Biolabs, Ipswich, USA). Constructs were subsequently transferred into lentiviral expression vectors LEgwSHIB (pRRL-EF1a-gwSH-IRES-BlastR)[87] or LE3FgwIP (pRRL-EF1a-3xFLAG-gw-IRES-PuroR) using LR reaction-based gateway cloning (Invitrogen). Cells infected with a corresponding empty lentiviral expression vector LEIB (pRRL-EF1a-IRES-BlastR) or LEIP (pRRL-EF1a-IRES-PuroR) served as negative control.

**Lentiviral cell line generation**. For lentivirus production, HEK293T cells were transiently transfected with psPAX2 and pMD2.G packaging plasmids (Addgene) and corresponding expression vectors. Media were changed 24 h post transfection and replaced with the target cell line-specific media. After 48 h the supernatant (containing virus) was collected, filtered through a 0.45 μm filter supplemented with 8 μg/mL protamine sulfate (Sigma-Aldrich) and added to subconfluent target cells. Cells were then spinfected at 2000 rpm for 45 min at room temperature. After 24 h the media were exchanged with fresh media and 24 h later the media were complemented with selection antibiotic for 5–7 days to select infected target cells. Knockout and overexpression efficiency was evaluated by immunoblot analysis.

**Immunoblotting**. Cells were centrifuged, washed once with cold PBS and subsequently lysed in Nonidet-40 lysis buffer (50 mM Tris-HCl pH 7.5, 150 mM NaCl, 5 mM EDTA, 1% NP-40, 1 mM PMSF and one tablet of Roche EDTA-free protease inhibitor cocktail (Sigma-Aldrich) per 50 mL) for 10 min on ice. Lysates were then centrifuged (13,000 rpm, 10 min, 4 °C) and proteins were quantified with the Bradford assay using γ-globin as a standard (Bio Rad). Cell lysates were separated by SDS-PAGE and transferred to nitrocellulose membranes Protran BA85 (GE Healthcare). The membranes were incubated with α-SLC16A1 (Santa Cruz Biotechnology; sc-365501 mouse mAb 1:500 dilution), α-SLC16A3 (Santa Cruz Biotechnology; sc-376140 mouse mAb 1:500), α-FLAG (Sigma-Aldrich; F1804 mouse mAb 1:1000), α-HSP90 (BD Biosciences; 610418 purified mouse Ab 1:2000) or α-AKT (Cell Signaling Technologies; #4685 rabbit mAb 1:1000) and visualized with goat-anti-mouse IgG (115-035-003) or goat-anti-rabbit IgG (111-035-003) horseradish peroxidase-conjugated secondary antibodies (Jackson ImmunoResearch; 1:5000) utilizing the ECL western blotting system (Thermo Fisher Scientific).

**Primary patient cells drug profiling**. Vitaly frozen mononuclear cells (MNCs) from patients with AML ($n = 13$) and CML ($n = 3$) after written informed consent were used for this study. Ethical approval was granted by the Ethics Commission of the Medical University of Vienna (Ethik Kommission 1676/2016). Patient characteristics are described in Supplementary Table 1. MNCs from bone marrow aspirates and peripheral blood were purified using Ficoll density gradient (Lymphoprep; STEMCELL Technologies). Upon thawing cells were maintained in RPMI 1640 medium (Gibco) with 10% fetal bovine serum (Gibco), penicillin (100 U/mL), and streptomycin (100 μg/mL). Patient cells were thawed, counted, and 10,000 cells per well were seeded onto pre-drugged 384-well plates. Drug screening and analysis were performed as described for the cell lines, with 70 drugs being tested in the patient samples. Mutational data were extracted from patient charts and referral reports where available.

**Reverse transcription quantitative PCR**. Total RNA was extracted using Trizol according to the manufacturer's protocol (Gibco). Quantification of *FASN* mRNA in response to pictilisib treatment and *GADPH* expression in HL-60 cells was performed using high-capacity RNA-to-cDNA kit and SYBR qPCR Green PCR Master Mix (Applied Biosystems). The following primers were used:

| | |
|---|---|
| FW primer GAPDH | GAGTCAACGGATTTGGTCGT |
| RV primer GAPDH | AATGAAGGGGTCATTGATGG |
| FW primer FASN | AGCAGTTCACGGACATGGAG |
| RV primer FASN | ATGGTACTTGGCCTTGGGTG |

**Reporting summary**. Further information on research design is available in the Nature Research Reporting Summary linked to this article.

## Data availability

Information on metabolic enzymes and drug-target annotations are available from publicly accessible databases such as KEGG (https://www.genome.jp/kegg), SMPDB (https://smpdb.ca), ChEMBL (https://www.ebi.ac.uk/chembl/), PubChem (https://pubchem.ncbi.nlm.nih.gov), and DrugBank (https://www.drugbank.ca). The manually curated CLIMET compound annotations are provided as Supplementary Data 1. Drug screening data generated in this study are provided as Supplementary Data 2 and 3. The Genomics of Drug Sensitivity in Cancer publicly available drug sensitivity data used in this study are available on the GDSC portal (https://www.cancerrxgene.org; GDSC1 data release). The DepMap cell line annotations (DepMap Public 20Q2; sample_info.csv), CCLE cell line targeted metabolomics (CCLE_metabolomics_20190502.csv; expressed as log10-transformed values cleaned up data), somatic mutation (DepMap 19Q3 Public. figshare. Dataset. https://doi.org/10.6084/m9.figshare.9201770.v1. (2019)), gene expression (DepMap 19Q3 Public. figshare. Dataset. https://doi.org/10.6084/m9.figshare.9201770.v1. (2019); and DepMap 20Q2 Public. figshare. Dataset. https://doi.org/10.6084/m9.figshare.12280541.v4. (2020)) expressed as log2 (TPM+1)), RNAi (D2_combined_gene_dep_scores.csv), and RPPA (CCLE_RPPA_20181003.csv expressed as log2 RPPA signal) data used in this study are available from the Broad DepMap portal (https://depmap.org/portal/download/). Other gene dependency datasets used are available from the Supplementary Information of the original publications (Tzelepis et al. [37], Wang et al. [43], and Wang et al. [44]). Patient sample gene expression data from TCGA can be accessed via the UCSC Xena project (http://xena.ucsc.edu) visualized with the GEPIA server (http://gepia.cancer-pku.cn/index.html). AML patient sample drug sensitivity data used in this study are available at http://www.vizome.org/aml/. Complete immunoblot scans are available in the source data file. All other data generated in this study and/or supporting the findings of this study are available within the paper, its supplementary information files and the source data accompanying the paper. Source data are provided with this paper.

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

## Acknowledgements

The authors thank the RESOLUTE consortium for providing codon-optimized *SLC16A1* and *SLC16A3* cDNAs; Anna Ringler and Kathrin Runggatscher (CeMM Platform Austria for Chemical Biology (PLACEBO)) for technical and infrastructural support; Nikolaus Krall and Noemi Mezaros for initial support in assembling the metabolic drug library; Austrian Academy of Sciences, European Research Council (ERC) grant Game of Gates 695214, Austrian Science Fund grant (FWF SFB F4711), Vienna Science and Technology Fund (LS16-034) to G.S.-F.; European Molecular Biology Organization Long Term Fellowship to T.P. (733-2016); and DOC Fellowship of the Austrian Academy of Sciences to A.L. for financial support. This project also received funding from the ERC under the

European Union's Horizon 2020 research and innovation program (grant agreement 677006, "CMIL" awarded to A.B.).

## Author contributions

T.P. and G.S.-F. designed the study, coordinated the project, performed data analysis and wrote the manuscript. T.P. assembled the metabolic drug library, annotated it, performed drug sensitivity experiments and associated data analysis. T.P., I.S., J.W.B., and F.K. performed and interpreted functional experiments. A.L. and A.B. performed and aided in data analysis and interpretation of Seahorse experiments. A.C.-R. aided in bioinformatic analysis. C.K., P.V., and P.B.S, coordinated the sampling of patient material and collection of associated clinical data. All authors discussed the results, commented and edited the manuscript.

## Competing interests

G.S.-F. is founder and shareholder of Haplogen, Allcyte, Proxygen, and Solgate. Allcyte has been acquired by Exscientia Ltd. and G.S.-F. is a shareholder of Exscientia Ltd. The remaining authors declare no conflict of interest.
