## [Peer Review File · Nature Communications]

Metabolic drug survey highlights cancer cell dependencies and vulnerabilitiesREVIEWER COMMENTS

Reviewer #1 (Remarks to the Author):

In the manuscript from Pemovska et al., the authors investigate the potential of applying a metabolic targets-related drug screen in myeloid leukemia cell lines to study the consequences of metabolic perturbations and potentially identify novel metabolic vulnerabilities. They justify the use of myeloid leukemia cells for the need of obtaining a comprehensive understanding of the molecular mechanisms associated with the disease and given previous reports showing that this disease can be sensitive to metabolic inhibition.

The authors claim that myeloid leukemia cells can be stratified in 5 groups based on their sensitivities to metabolic inhibitors, however it is not clear whether this stratification is robust enough, nor whether it has any relevance with in vivo responses. Furthermore, the associations made between genotype-metabolic sensitivity phenotype require further validation both in vitro and in vivo and more in-depth functional and mechanistic understanding.

The authors should address the following to strengthen this study

Major comments:

1) Fig 2c (and Extended figure 2): In figure 2, the authors demonstrate the stratification of myeloid leukemia cells in 5 clusters, however it is unclear how this classification is derived and/or how robust it is. BV-173 could represent a group of its own based on the dendrogram generated in extended figure 2. Clustering validation statistics and alternative clustering methods should be used to demonstrate the robustness of this classification.

Also, the black boxes in extended figure 2 indicate the compounds displaying differential effects across the cell lines. 25 compounds are highlighted, yet the authors only selected 19 to stratify the cell lines in Figure 2c. What is the rationale behind this selection? The authors have shown in extended figure 1b that some drugs show overall different responses between AML and CML cells. Out of the 8 drugs highlighted, 6 have been used for the stratification of the 5 groups. Selecting drugs that have differential effects between different disease subtypes is likely to introduce bias in unsupervised clustering.

2) Are there any mutations in Figure 3a that drive the stratification of cells in one of the 5 groups and can this stratification change to another group following loss of a tumour suppressor or introduction of an oncogenic mutation. Representative novel associations between genotype-phenotype indicated in Figure 3b should be validated in vitro and in vivo in at least two cell line models.

3) Figure 4 - For the mutation co-occurrence and mutual exclusivities, validation of these findings in a couple of cell lines and checking potential drug interactions in the form of synergistic, additive or antagonistic effects would be needed to support the conclusions.

4) Based on Figure 5b HL60 cells show a higher sensitivity to pictilisib - a class I pan-PI3K inhibitor with higher specificity to α , and δ isoforms, the expression levels of which are equivalent between HL60 cells and other cell lines (5d). HL60 cells also appear to show a higher dependency on PIK3C2A depletion (which is not a target of pictilisib). The authors claim that the lower expression levels of PI3K family members might explain the higher sensitivity of HL-60 cells to pictilisib. The authors should test whether the sensitivity of other cells to pictilisib is enhanced by depletion of individual PI3K isoforms.

5) Based on the results of the genome-wide CRISPR library screen (extended figure 7a), HL60 cells seem to be reliant on lipid-related genes SCD and ACSL4. As three SCD inhibitors were included in this study, it would be interesting to show the sensitivity of HL60 cells to these inhibitors compared to other cell lines.

6) The authors mention that there is a direct link between PIK3CA and FASN in the cellular and disease context based on a cell line specific observation and its increased sensitivity to both PI3K and FASN inhibition. It would add a lot of value to this finding to validate this connection by looking at how inhibition of PI3K signalling affects FASN expression and/or activity for example.

Minor comments:

- Fig 2b and extended figure 2: The authors have added Indisulam in figure 2c, when this is not highlighted with a black box in extended figure 2 and is not shown to be significant between different cell lines.

- The colours of the squares of figure 2c, especially for group III and group IV are difficult to distinguish.

- Extended data 4: The authors claim that mTOR sensitivity is associated with enrichment of metabolites involved in arginine and proline metabolism. However, based on extended figure 1e and similar sensitivities to mTOR inhibitor for group I, II and IV, the levels of these metabolites should be similar in all 3 groups. The same stands for group IV being more sensitive to lipid metabolism inhibitor because alpha-glycerophosphocholine levels are higher, yet group I and III have high levels of other lipid metabolism related metabolites. The authors should refrain from making such claims as they are highly speculative and likely erroneous.

- Line 244 to 251, with figure 3. Some of the correlations between genetic event and drug sensitivity are already known and even shown in AML, so this should be highlighted.

o Knapper et al, 2006, Blood, <https://doi.org/10.1182/blood-2006-04-015560> (FLT3 and Lestaurtib)

o Di Nicolantonio et al, 2010, JCI, <https://doi.org/10.1172/JCI37539> (Everolimus efficacy based on deregulation of PIK3CA and KRAS)

- Figure 7d, LAML abbreviation should be explained in the legend.

Also, figures go from Panel A to E but the legend for the last two panels indicates d-i.

Reviewer #2 (Remarks to the Author):

Pemovska et al, construct a screen of myeloid leukemia cell lines against a metabolic drug library of 243 compounds which target one or many targets involved in cell metabolism which may be preferentially utilized in leukemia cells. The authors grouped cell lines by both the capacity to induce apoptosis, and the genetic and metabolic machinery within the leukemia cells. They described distinct metabolic phenotypes for several groups of AML cell lines which emerged from unsupervised linkage clustering, and successfully went on to make efforts to illustrate relationships between mutational genotypes, morphologic phenotypes and metabolic species in each cell line.

The authors say, quite right, that "cellular metabolism must be considered in the development of new strategies." These efforts contribute, interestingly, to the working understanding of differential metabolic profiles of AML, and how to approach these differences, perhaps, ultimately, in patient samples, in a more high throughput fashion. Curiously, however, the authors make no mentions of some of the more seminal findings in leukemia metabolism made clear in the past 2-3 years, ignoring, for example the targeting of BCL2 family protein – despite the fact that BCL2 inhibition is the largest paradigm change in AML therapy since 7+3 (Line 85 refs 12-17 make small reference to metabolic effects of selective BCL2 family inhibitors and references and introduction in this section should include more focus on BCL2 family proteins). Please see - PMID: 31974170, 33028621

Further, while the authors concede the plasticity of the metabolic states, the manuscript's use of the cell line basal metabolic states without addressing the effect of specific treatment on metabolic addiction implies a lack of dynamism. Is it not likely that the metabolic dependencies are influenced by therapies as has been shown in a variety of examples in AML? Can the authors further illustrate how therapies may even "prime" cells for metabolic dependency (as a function of resistance) and then succumb to an alternative treatment?

The association made between genotype and phenotype is critical, but previously explored (PMID: 31974170, PMID: 31296572) and not referenced by the authors.

This manuscript is a thoughtful compilation – and the first to my knowledge – of systematic use of a drug library (CLIMET) to metabolically profile leukemia (or any other tumor) cells, at least to the point that one can imagine directing therapy. Would this exploration not be better served with the addition of a series of annotated patient samples? (Murine xenografts testing the compounds in the engrafted patient samples would be even better, but maybe impractical for this study). At least use of patient samples, with, ideally some attempt to associate any clinical successes to postulations which could be made by the profiling, is necessary.

Figure 2B summarizes that the most effect agents against AML cell lines are gemcitabine, toptecan, methotrexate, digoxin, among others, and while most of these agents have been in use for over 50 years, none of them are routinely used in the treatment of myeloid leukemia. This represents a disconnect between the findings and the application of the findings.

Figure 2E is a clever figure representing separation of the AML cell lines by OCAR/ECAR ratio. The meaning of this differential ratio between each leukemia, however, is not well described in the manuscript.

Reviewer #3 (Remarks to the Author):

The manuscript from Pemovska et al describes the assembly of a metabolism-biased compound library that is used to profile AML cells in hopes of identifying metabolic cancer vulnerabilities. It is an interesting approach, but overall, this reviewer found too many instances where the data were not sufficient to support the claims. I will be specific about these instances below. The most solidly supported claim is the identification of synthetic lethality between MCT1 and MCT4 in which cells with low MCT4 expression are hypersensitive to an MCT1 inhibitor. However, as is referenced in the discussion, this has already been discovered with both genetic and chemical approaches (using the same compound as is used in this manuscript).

Major notes:

- the clustering of cell lines into distinct groups in Fig2 is based on 19 compounds of the >200 in the library. Why were these selected in particular and why was the entire library not used for clustering analysis (Ext. Fig2)? Do the groups remain the same when clustering on the entire library vs on the 19 compounds?

- there are a number of claims made as possible consequences of the clustering analysis that go beyond just pointing out the differences in sensitivities among the group. e.g. that Group I may be addicted to ROS production due to their sensitivity profile. However, these claims are not addressed with experiments and the first half of the manuscript reads as a significant amount of speculation.

- on the same note, on lines 212-215, it is claimed that "Critically, the abundance of specific metabolites could be matched to the drug sensitivity defined groups, validating the notion that the similarities in functional phenotypes of the cell lines in our analysis are anchored to particular metabolic vulnerabilities." However, this is not supported by the data. Any multidimensional dataset (RNA-seq, metabolomics, proteomics, etc.) will provide an opportunity to find differences in individual transcripts/metabolites/proteins that follow some supervised categorization. It does not mean that these differences are functionally related to the categorization or the phenotype of interest. It does not mean that these metabolites are functionally related to any known or unknown metabolic vulnerabilities in these groups or that the metabolites are responsible for stratifying the groups based on their responses to the metabolic drugs. The key questions would be: can the same cell line groups be clustered into their original drug response clusters based on these metabolites? can you alter the metabolite levels and change the responses to the drugs that are uniquely active in one or more groups? those are the key questions that would support this type of claim.

- the pairwise analysis is used to suggest that the correlation between drugs addressing distinct target classes is evidence that the pathways/compounds are functionally related (Line 276) and could be used leveraged for synergy (Line 407-9). That is, that targeting PI3K and FA synthesis would be synergistic. But this is not tested. The pairwise correlations could suggest a number of other possibilities, for instance that these drugs have shared off-targets, not that the pathways of the intended targets are functionally related.

- part of the evidence for FA sensitivity are viability effects in HL-60 cells when re-testing FASN inhibitors up to 100 μ M. This is a drastically high concentration which could have a huge number of relevant off-targets and could easily be toxic to most or all cell lines in the study.

- To understand the sensitivity of HL-60 cells to pictilisib, the authors noticed low expression of PI3K family members other than PIK3CB/D (the pictilisib targets) in HL-60. This is claimed to be the basis of sensitivity in HL-60 cells, but it is not tested. A rescue experiment expressing other PIK3 family members would be required. Especially given that the expression profile of PI3K family members in HL60 cells looks very similar to other AML cell lines, like MV4-11 and MOLM-13. Claiming that this could be used to select patients for PI3K therapeutics is not appropriate.

- The most exciting data and well-supported claim in the manuscript is the identification of MCT1/4 synthetic lethality. However, this has already been discovered and reported multiple times.

Minor notes:

- the library is pitched as a way to overcome the difficulties of CRISPR in primary cells or cell lines that are difficult to work with, but all the experiments are done in AML cells, which have been extensively profiled by genome-scale CRISPR screens. Is it true that this library could easily be used to profile primary cells?

- indisulam is referenced multiple times in the manuscript. What is the basis for its annotation as a metabolic regulator? As far as I am aware, its cytotoxicity is due to degradation of the splicing factor RBM39 (science and nat chem biol papers from 2017)

Point by point reply, Pemovska et al

The authors' answers are written in **bold italics** and in **red text** are changes made in the revised manuscript, figures, and supplementary figures and tables. All changes in the revised manuscript are marked in yellow.

REVIEWER COMMENTS

Reviewer #1 (Remarks to the Author):

In the manuscript from Pemovska et al., the authors investigate the potential of applying a metabolic targets-related drug screen in myeloid leukemia cell lines to study the consequences of metabolic perturbations and potentially identify novel metabolic vulnerabilities. They justify the use of myeloid leukemia cells for the need of obtaining a comprehensive understanding of the molecular mechanisms associated with the disease and given previous reports showing that this disease can be sensitive to metabolic inhibition. The authors claim that myeloid leukemia cells can be stratified in 5 groups based on their sensitivities to metabolic inhibitors, however it is not clear whether this stratification is robust enough, nor whether it has any relevance with in vivo responses. Furthermore, the associations made between genotype-metabolic sensitivity phenotype require further validation both in vitro and in vivo and more in-depth functional and mechanistic understanding.

We thank the reviewer for her/his efforts and constructive criticism. In our revised manuscript, we have added new datasets to increase the robustness and in vivo relevance and performed additional mechanistic experiments which significantly strengthen the manuscript (see details below).

The authors should address the following to strengthen this study

Major comments:

1) Fig 2c (and Extended figure 2): In figure 2, the authors demonstrate the stratification of myeloid leukemia cells in 5 clusters, however it is unclear how this classification is derived and/or how robust is it. BV-173 could represent a group of its own based on the dendrogram generated in extended figure 2. Clustering validation statistics and alternative clustering methods should be used to demonstrate the robustness of this classification.

The classification in Figure 2c is derived by performing hierarchical clustering (complete linkage method and Spearman (drugs) and Euclidean (cell lines) distance measures) of the metabolic drug responses profiles of the 15 tested cell lines to the 77 compounds that had a cell viability inhibiting effect in at least one cell line. The Figure 2c is a snapshot of the compounds that contribute most to the classification with statistics and the complete heatmap is provided as Supplementary Fig. 2. The dendrogram clustering of the cell lines is the same between these two figures.

To address the significance of the five identified functional subgroups, we assessed their clustering reproducibility by randomly resampling bootstrap samples of the original data and then calculated the frequency that each cluster appears in the hierarchical clustering of the bootstrap replicates. This

information is now included in the updated Supplementary Fig. 2 (see below). As highlighted in the new supplementary figure, the subclusters appeared relatively robust even though some samples clearly match the subtyping pattern better than others. Overall, we expect that the future assessment of larger sample cohorts will refine and complement these subtypes.

i. See updated Supplementary Fig. 2

- ii. See edited Supplementary Fig. 2 legend: Metabolic drug sensitivity testing classifies myeloid leukemia cell lines in 5 functional groups. Unsupervised hierarchical clustering of the 77 viability affecting drugs is presented as a heatmap across 15 cell lines. Clustering was performed using the complete linkage clustering method and Euclidian and Spearman distance measures of the AUC values of cell lines and drugs, respectively. The response profiles of several compounds and compounds classes display differential effects across the cell lines tested and are outlined with black boxes. The reproducibility of the clusters was evaluated by randomly resampling 10,000 bootstrap samples of the original dataset and calculating the frequency at which each cluster appears in the hierarchical clustering of the bootstrap replicates. The numbers in the dendrogram tree of the cell lines indicate the approximately unbiased (AU) empirical frequencies (0-100%) from the multiscale bootstrap resampling implemented in the Pvcust R-package.
- iii. See edited manuscript text lines 166-169: While each cell line had a unique metabolic vulnerability profile, the activity of 18 compounds in particular stratified the cells lines into five robust and distinct functional groups (Fig. 2c and Supplementary Fig. 2).
- iv. See updated Methods section pertaining to this analysis lines 591-593: Reproducibility of the clustering was assessed with the bootstrap resampling method using the Pvcust R-package⁷².

Also, the black boxes in extended figure 2 indicate the compounds displaying differential effects across the cell lines. 25 compounds are highlighted, yet the authors only selected 19 to stratify the cell lines in Figure 2c. What is the rationale behind this selection? The authors have shown in extended figure 1b that some drugs show overall different responses between AML and CML cells. Out of the 8 drugs highlighted, 6 have been used for the stratification of the 5 groups. Selecting drugs that have differential effects between different disease subtypes is likely to introduce bias in unsupervised clustering.

We thank the reviewer for bringing up this point for further clarification. In Figure 2c only the compounds that significantly influenced the grouping of the cell lines were shown, whereas in the Supplementary Fig. 2 also compounds that had a consistent sensitivity profile within a group were highlighted albeit not reaching statistical significance. However, we acknowledge that this may be misleading, and thus we reanalyzed the data and decided to highlight only the compounds that reached statistical significance prior to multiple testing correction and show both the original and adjusted P-value (see updated Figure 2c and Supplementary Fig. 2). With respect to the disease subtype specific vulnerabilities, we would like to clarify that only Torin1, rotenone and cytarabine had a significantly increased sensitivity in CML cell lines in comparison to AML cell lines (see Rebuttal Letter Fig. 1), however after multiple testing correction this difference in response to these agents was not significant. The responses to the remaining highlighted compounds in Supplementary Fig. 1b were not statistically different between AML and CML cell lines, however a trend was observed which may hold true in an increased dataset. Therefore, we have updated Supplementary Fig. 1b to only highlight the compounds that exhibited

significant disease-specific effect. Nonetheless, the disease subtype did not play a significant role of the clustering as cell lines clustered together irrespective whether they were CML or AML. Compounds that showed preferential activity in one or the other disease subgroups did not significantly affect the clustering with the exception to Torin1.

i. See updated Figure 2c

ii. See updated Figure 2c legend: c, Heatmap illustrating the functional grouping of myeloid cancer cells in 5 taxonomic groups based on metabolic vulnerability profiles. While each cell line had an overall distinct drug sensitivity profile, the activity of 18 compounds in particular significantly functionally stratified the myeloid cancer cells lines (an ordinary one-way ANOVA analysis * $P \leq 0.05$; ** $P \leq 0.01$; *** $P \leq 0.001$; **** $P \leq 0.0001$; FDR of 10% was deemed significant; * $Padj \leq 0.1$; ** $Padj \leq 0.05$; *** $Padj \leq 0.01$; ns not significant).

iii. See Rebuttal Figure 1 and legend

Rebuttal Figure 1. Bar graphs depicting the distribution of AUC values for selected compounds between CML and AML cell lines. An unpaired T-test was performed to assess significance * $P \leq 0.05$; ** $P \leq 0.01$; ns not significant.

iv. See updated Supplementary Fig. 1b

v. See modified main text lines 154-158: The average drug sensitivity profiles between AML and CML cell lines were highly concordant ($r = 0.78$), with the exception of CML cells being more sensitive to Torin1 (mTOR inhibitor), cytarabine (nucleoside analog), and rotenone (complex I inhibitor) (Supplementary Fig. 1b). Hence, the disease subtype did not play a significant role in the cell line clustering.

2) Are there any mutations in Figure 3a that drive the stratification of cells in one of the 5 groups and can this stratification change to another group following loss of a tumour suppressor or introduction of an oncogenic mutation. Representative novel associations between genotype-phenotype indicated in Figure 3b should be validated in vitro and in vivo in at least two cell line models.

Thank you for this interesting question. The tested cell lines distribute differently based on the metabolic drug vulnerability profile in comparison to

the mutational profile (Figure 2; 3a; Rebuttal Figure 2). Clustering based on mutational profile seems to be largely driven by high frequency events such as TP53 mutations, MLL and BCR-ABL1 fusions, and AML M5 subtype. While we did not experimentally confirm this, the loss of a tumor suppressor or introduction of an oncogenic mutation could potentially lead to a change to another stratification group if the metabolic vulnerability profile changes as a result of introducing a mutation. In either regard, the mutational profiles were provided for a correlation analysis of molecular and functional profiles of the cells and the majority of clustering of drug sensitivities could not be attributed to obvious alterations in the genome of these cell lines.

i. See Rebuttal Figure 2 and legend

Rebuttal Figure 2. Clustering of AML and CML cell lines based on mutational profiles and FAB classification. The clustering was performed using the complete linkage clustering method and Euclidian distance measure. Red square indicates presence of a mutation.

ii. See modified mail text lines 251-252: The mutational status did not considerably drive the functional stratification of the cell lines

Furthermore, the representative novel phenotype to genotype associations are based on in vitro experimental drug sensitivity data (updated Supplementary Fig. 5) where we performed multiple t-tests comparing the drug sensitivity

observed in mutated and wildtype cases and significant associations were visualized in Figure 3B. We feel that *in vivo* validation of these associations is outside of the scope of the manuscript and future studies could explore their clinical impact and relevance.

iii. See updated Supplementary Fig. 5

iv. See updated Supplementary Fig. 5 legend: Significant genotype to phenotype associations. a-p, Box-plots and mean dose response curves depicting sensitivity of several metabolic modifiers that have significantly higher or lower sensitivity in *FLT3-ITD* (a-c), *RAS* (d-h), *CREBBP* (i), and *TP53* (j-p) mutant versus wild-type cell lines as depicted in Fig. 3a. Error bars signify mean \pm SD and the difference in response was assessed with a two-tailed unpaired *T*-test after multiple testing correction (FDR of 10% was deemed significant; * *P*_{adj} \leq 0.1; ** *P*_{adj} < 0.05; *** *P*_{adj} < 0.01).

3) Figure 4 - For the mutation co-occurrence and mutual exclusivities, validation of these findings in a couple of cell lines and checking potential drug interactions in the form of synergistic, additive or antagonistic effects would be needed to support the conclusions.

The correlogram depicts positive and negative drug response correlations across the entire cell line metabolic drug sensitivity dataset irrespective of the mutational profiles of the tested cell lines. The analysis identified drugs that exhibited co-occurring or mutually exclusive responses, which can come from the drug response profile of a single cell line or multiple cell lines. In order to investigate the potential drug interactions, we have performed drug synergy experiments (8x8 matrices) of 11 different drug combinations across four different cell lines (HL-60, Mono-Mac-6, BV-173, and MV4-11) that best capture the significant drug-drug correlations depicted in Figure 4. The results indicated that in majority of cases combinatorial effects were detected using the SynergyFinder tool (lanevski et al; Bioinformatics 2017; lanevski et al; Nucleic Acid Research 2020) with the Bliss independence model. However, there were a few drug combinations pairs (e.g., lestaurtinib with 5-fluorouracil) where marked single agent activity was detected but this resulted in an antagonistic combinatorial effect.

i. See new Figure 4b

ii. See new Figure 4b legend: **b**, Heatmap showing the deviation from Bliss independence score for each tested combination and cell lines. Synergy is denoted in red while antagonism is shown in blue. Analysis was performed using the SynergyFinder tool^{77,78}. The individual drug sensitivity is shown within the heatmap squares per cell line with + indicating sensitivity; +/- indicating moderate sensitivity, and - indicating no sensitivity. The first sign refers to the sensitivity of the first listed drug of the combination and the second sign to the sensitivity of the second drug.

iii. See new Supplementary Fig. 6

Mono-Mac-6

Bliss synergy score: -1.583
-30 -20 -10 0 10 20 30

Bliss synergy score: -3.075
-40 -20 0 20 40

Bliss synergy score: 2.129
-40 -20 0 20 40

Bliss synergy score: 3.316
-30 -20 -10 0 10 20 30

Bliss synergy score: -0.038
-40 -30 -20 -10 0 10 20 30 40

Bliss synergy score: -2.966
-30 -20 -10 0 10 20 30

Bliss synergy score: -5.222
-40 -20 0 20 40

Bliss synergy score: -2.178
-40 -20 0 20 40

Bliss synergy score: 3.269
-40 -20 0 20 40

Bliss synergy score: -4.162
-40 -30 -20 -10 0 10 20 30 40

Bliss synergy score: 7.966
-40 -30 -20 -10 0 10 20 30 40

BV-173

Bliss synergy score: 1.033

Bliss synergy score: -4.795

Bliss synergy score: 1.284

Bliss synergy score: 4.085

Bliss synergy score: -0.61

Bliss synergy score: 1.962

Bliss synergy score: 0.598

Bliss synergy score: 3.713

Bliss synergy score: -5.216

Bliss synergy score: -1.574

Bliss synergy score: -2.965

MV4-11

iv. See new Supplementary Fig.6 legend: 3D drug synergy visualizations for 11 different drug-drug interactions in HL-60, Mono-Mac-6, BV-173, and MV4-11 cells. Data was generated in 8x8 drug synergy matrices and analyzed using the SynergyFinder tool using the Bliss independence model with % survival values at each tested concentration relative to DMSO (negative control) and 10 μ M

bortezomib (positive control). The synergy score for a drug combination is averaged over all the dose combination measurements. The 2D and 3D synergy maps highlight synergistic and antagonistic dose regions in red and green, respectively.

- v. See edited main text lines 286-295: To investigate the identified drug interactions further we performed drug combinatorial screens in four different cell lines (HL-60, Mono-Mac-6, BV-173, and MV4-11) that best captured the significant drug-drug correlations. The results showed that in majority of cases combinatorial effects were observed as hypothesized (Fig. 4b and Supplementary Fig. 6). However, there were a few drug combination pairs (e.g., lestaurtinib and 5-fluorouracil, and dorsomorphin and rosiglitazone) that exhibited marked single agent activity but an antagonistic combinatorial effect. Taken together this analysis could facilitate elucidation of the context-specific molecular mechanisms of action and foster a rationale for context-dependent drug combinations, though alternative bases for the observed interactions should also be contemplated.
- vi. See new methods subsection lines 644-653: Drug combination screens. The significant drug-drug interactions identified in Figure 4a were evaluated by drug synergy screens in four different cell lines HL-60, Mono-Mac-6, BV-173, and MV4-11. Briefly, 8x8 synergy matrices were tested for each drug combination and cell lines with majority of drugs being tested at 5, 10, 50, 100, 500, 1000, 5000nM concentration with the exception of rosiglitazone and lestaurtinib that were tested at 1, 5, 10, 50, 100, 500, and 1000nM concentration. The experimental procedure and readouts were as described above for single agent screening. Combinatorial effects were evaluated and visualized using the SynergyFinder^{77,78} web portal (<https://synergyfinder.fimm.fi>) using the Bliss independent model.

4) Based on Figure 5b HL60 cells show a higher sensitivity to pictilisib - a class I pan-PI3K inhibitor with higher specificity to α , and δ isoforms, the expression levels of which are equivalent between HL60 cells and other cell lines (5d). HL60 cells also appear to show a higher dependency on PIK3C2A depletion (which is not a target of pictilisib). The authors claim that the lower expression levels of PI3K family members might explain the higher sensitivity of HL-60 cells to pictilisib. The authors should test whether the sensitivity of other cells to pictilisib is enhanced by depletion of individual PI3K isoforms.

To address this point, we have now generated MV4-11 cells lacking PIK3CA, PIK3CB, PIK3CD or PIK3CG using CRISPR technology with two independent guides per gene and using Renilla as control. We choose MV4-11 as the reviewer rightly points out it had the most similar gene expression profile of the class I PI3K family members to HL-60 cells. Then we tested pictilisib sensitivity and found that sensitivity to pictilisib is moderately reduced by depletion of the individual PI3K isoforms. We performed the same experiment in HL-60 cells and detected a mild gain in sensitivity in PIK3CB knockout cells, and loss of sensitivity in PIK3CD knockout cells. Due to technical challenges and cell growth disadvantage of some of the PI3K isoform knockouts, we provide this data only as a Rebuttal Figure 3.

i. See Rebuttal Figure 3 and legend

Rebuttal Figure 3. Average response to pictilisib in HL-60 or MV4-11 sgRen control infected cells, and PIK3CA, PIK3CB, PIK3CD or PIK3CG deficient cells. Error bars indicate standard deviation of the mean of technical triplicates.

We also performed the opposite experiment as asked by reviewer #3 and generated HL-60 and MV4-11 cells overexpressing either PIK3CA, PIK3CB, PIK3CD or PIK3CG and screened for pictilisib sensitivity. We found that pictilisib sensitivity was enhanced in HL-60 cells overexpressing PIK3CA, PIK3CB or PIK3CD in comparison to HL-60 cells lentivirally transduced with an empty vector. In contrast, we detected a moderate increase in pictilisib sensitivity in PIK3CA and PIK3CG MV4-11 overexpressing cells, while there was no difference in sensitivity in PIK3CB and a minor loss in sensitivity in PIK3CD overexpressing cells. These findings indicate that there are perhaps different isoforms on which sensitivity to pictilisib is dependent in these two cell lines and that no single specific PI3K isoform influences sensitivity to pictilisib. Moreover, our findings further suggest that a combination of isoforms such as PIK3CA and either PIK3CB or D determines pictilisib susceptibility. These findings do not fully support our initial hypothesis for the differential sensitivity of pictilisib in HL-60 cells to be explained by the overall lower expression levels of PI3K members in these cells. Thus, we have removed this statement in the revised manuscript.

ii. See new Supplementary Fig. 7d,e

- iii. See new Supplementary Fig. 7d,e legend: d-e, Average response to pictilisib in HL-60 and MV4-11 empty vector, and PIK3CA, PIK3CB, PIK3CD or PIK3CG overexpressing cells. Error bars indicate standard deviation of the mean of technical triplicates. Data is representative of two independent experiments. Moreover, immunoblot analysis of PIK3CA, PIK3CB, PIK3CD, and PIK3CG expression in HL-60 and MV4-11 cells lentivirally transduced with empty vector, *PIK3CA*, *PIK3CB*, *PIK3CD* or *PIK3CG*-cDNA is shown.
- iv. See updated main text lines 314-325: To explore determinants to pictilisib sensitivity, we overexpressed the different class I PI3K isoforms in HL-60 and MV4-11 cells, which have a comparable expression pattern of those genes. Pictilisib sensitivity was enhanced in HL-60 cells overexpressing *PIK3CA*, *PIK3CB* or *PIK3CD* in comparison to HL-60 cells lentivirally transduced with an empty vector (Supplementary Fig. 7d). In contrast, we detected a moderate increase in pictilisib sensitivity in *PIK3CA* and *PIK3CG* MV4-11 overexpressing cells, while there was no difference in sensitivity in *PIK3CB* and a minor loss in sensitivity in *PIK3CD* overexpressing cells (Supplementary Fig. 7e). These findings suggest that there is not one specific PI3K isoform that influences sensitivity to pictilisib in these cells, highlighting the possibility of combination of isoforms such as PIK3CA and either PIK3CB or D to determine pictilisib susceptibility.
- v. See updated methods lines 703-714: Generation of cells overexpressing *SLC16A1*, *SLC16A3*, *PIK3CA*, *PIK3CB*, *PIK3CD*, and *PIK3CG* was performed by lentiviral cDNA delivery experiments. SLC-encoding cDNAs were obtained as codon-optimized versions from Genescript and transferred into pDONR221 entry vector using BP reaction gateway cloning (Invitrogen). PI3K isoform cDNAs were ordered from Addgene (#81736, #82221, #82222, and #81843). In the case of Addgene plasmid #81843 the stop codon was removed by site directed mutagenesis to enable C-terminal tagging (E0554S, New England Biolabs, Ipswich, USA). Constructs were subsequently transferred into lentiviral

expression vectors LEgwSHIB (pRRL-EF1a-gwSH-IRES-BlastR)⁸³ or LE3FgwIP (pRRL-EF1a-3xFLAG-gw-IRES-PuroR) using LR reaction-based gateway cloning (Invitrogen). Cells infected with a corresponding empty lentiviral expression vector LEIB (pRRL-EF1a-IRES-BlastR) or LEIP (pRRL-EF1a-IRES-PuroR) served as negative control.

Lines 735-741: The membranes were incubated with α -SLC16A1 (sc-365501, Santa Cruz Biotechnology), α -SLC16A3 (sc-376140, Santa Cruz Biotechnology), α -FLAG (F1804, Sigma-Aldrich), α -HSP90 (610418, BD Biosciences) or α -AKT (#4685, Cell Signaling Technologies) and visualized with horseradish peroxidase-conjugated secondary antibodies (Jackson ImmunoResearch) utilizing the ECL Western blotting system (Thermo Fisher Scientific).

5) Based on the results of the genome-wide CRISPR library screen (extended figure 7a), HL60 cells seem to be reliant on lipid-related genes SCD and ACSL4. As three SCD inhibitors were included in this study, it would be interesting to show the sensitivity of HL60 cells to these inhibitors compared to other cell lines.

We thank the reviewer for bringing up this point. We have now added the dose response curves for these agents to the updated Supplementary Fig. 8c (see below).

i. See updated Supplementary Fig. 8c

ii. See updated Supplementary Fig.8c legend: c, Dose response of C75 and orlistat (described to target FASN) as well as 3 stearoyl-CoA desaturase 1 (SCD1) inhibitors in HL-60 cells.

iii. Updated main text lines 335-339: Although, C75 and orlistat, first generation FASN inhibitors, did not show an effect in the original drug screen, retesting of the respective compounds at an increased dose led to a viability reduction, whereas stearoyl-CoA desaturase 1 inhibitors exhibited limited activity in HL-60 cells (Supplementary Fig. 8c).

6) The authors mention that there is a direct link between PIK3CA and FASN in the cellular and disease context based on a cell line specific observation and its increased sensitivity to both PI3K and FASN inhibition. It would add a lot of value to this finding to validate this connection by looking at how inhibition of PI3K signalling affects FASN expression and/or activity for example.

Based on the observation that HL-60 cells had a distinct response profile to pictilisib and GSK2194069, we hypothesized that there might be an association between PI3K signaling and fatty acid synthesis in that particular cellular context. As recommended by the reviewer and to further explore this relationship we performed RT-qPCR for FASN expression in response to pictilisib exposure and observed a dose dependent reduction of FASN expression, suggesting that PI3K signaling affects FASN levels and possibly activity. These data are now added as the new Supplementary Figure 8d. Moreover, we performed combinatorial drug experiments and identified that pictilisib and GSK2194069 exhibit a synergistic combinatorial effect in HL-60 cells (see new Fig. 4b and Supplementary Fig. 6 shown above under point 3).

i. See new Supplementary Fig. 8d

ii. See new Supplementary Fig. 8d legend: d, FASN transcript expression measured by RT-qPCR in response to increasing concentrations of pictilisib treatment. Error bars indicate SD; (n = 3). Data is representative of two independent experiments.

iii. See GSK2194069 and pictilisib combinatorial effect in HL-60 cells

iv. See edited main text lines 339-344: We, further, detected a combinatorial effect of pictilisib and GSK2194069 in HL-60 cells (Supplementary Fig. 6). Moreover, pictilisib exposure led to a dose-dependent reduction of FASN expression (Supplementary Fig. 8d). Taken together, our data illustrate that HL-60 are dependent on PI3K signaling and *de novo* FA synthesis and

conceivably these pathways and processes are functionally linked in this cellular and disease context.

- v. See updated methods lines 757-762: Reverse transcription quantitative PCR (RT-qPCR) Total RNA was extracted using Trizol according to manufacturer's protocol (Gibco). Quantification of FASN mRNA in response to pictilisib treatment and GAPDH expression in HL-60 cells was performed using High-capacity RNA-to-cDNA kit and SYBR qPCR Green PCR Master Mix (Applied Biosystems). The following primers were used:

FW primer GAPDH	GAGTCAACGGATTTGGTCGT
RV primer GAPDH	AATGAAGGGGTCATTGATGG
FW primer FASN	AGCAGTTCACGGACATGGAG
RV primer FASN	ATGGTACTTGGCCTTGGGTG

Minor comments:

- Fig 2b and extended figure 2: The authors have added Indisulam in figure 2c, when this is not highlighted with a black box in extended figure 2 and is not shown to be significant between different cell lines.

Thank you for indicating this to us. As explained in the major point 1, we have now highlighted indisulam in both the main Figure 2c as well as Supplementary Fig. 2, given that it exhibited differential effects in subgroup I in comparison to subgroups III and V.

- The colours of the squares of figure 2c, especially for group III and group IV are difficult to distinguish.

We apologize for this, we have updated the figure such that an increased line thickness is used, which should increase discriminability.

- Extended data 4: The authors claim that mTOR sensitivity is associated with enrichment of metabolites involved in arginine and proline metabolism. However, based on extended figure 1e and similar sensitivities to mTOR inhibitor for group I, II and IV, the levels of this metabolites should be similar in all 3 groups. The same stands for group IV being more sensitive to lipid metabolism inhibitor because alpha-glycerophosphocholine levels are higher, yet group I and III have high levels of other lipid metabolism related metabolites. The authors should refrain from making such claims as they are highly speculative and likely erroneous.

We apologize for overstating these observations. In the revised text we have now removed the speculative language.

- Line 244 to 251, with figure 3. Some of the correlations between genetic event and drug sensitivity are already known and even shown in AML, so this should be highlighted.

o Knapper et al, 2006, Blood, <https://doi.org/10.1182/blood-2006-04-015560> (FLT3 and Lestaurtib)

o Di Nicolantonio et al, 2010, JCI, <https://doi.org/10.1172/JCI37539> (Everolimus efficacy based on deregulation of PIK3CA and KRAS)

Thank you for pointing these studies out, we have now referenced them in the main text and discussion (line 477).

- i. See edited main text lines 252-255: Systematic comparison of mutant vs. wild-type cases (where at least 3 mutant cell lines could be identified), revealed statistically significant novel and previously known correlations between *FLT3* mutations and sensitivity to 5-FU, lestaurtinib^{31,32}, and PF-02545920.
- ii. See updated references:
31 Knapper, S. *et al.* A phase 2 trial of the *FLT3* inhibitor lestaurtinib (CEP701) as first-line treatment for older patients with acute myeloid leukemia not considered fit for intensive chemotherapy. *Blood* **108**, 3262-3270, doi:10.1182/blood-2006-04-015560 (2006).
32 Knapper, S. *et al.* The effects of lestaurtinib (CEP701) and PKC412 on primary AML blasts: the induction of cytotoxicity varies with dependence on *FLT3* signaling in both *FLT3*-mutated and wild-type cases. *Blood* **108**, 3494-3503, doi:10.1182/blood-2006-04-015487 (2006).
47 Di Nicolantonio, F. *et al.* Deregulation of the PI3K and KRAS signaling pathways in human cancer cells determines their response to everolimus. *J Clin Invest* **120**, 2858-2866, doi:10.1172/JCI37539 (2010).

- Figure 7d, LAML abbreviation should be explained in the legend. Also, figures go from Panel A to E but the legend for the last two panels indicates d-i.

We have now included the explanation of the LAML abbreviation in the figure legend of Figure 7 and have corrected the legend to refer to the correct panels. We apologize for this omission in the initial submission.

- i. See modified Figure 7a,b legend lines 1149-1150: LAML indicates Acute Myeloid Leukemia TCGA dataset.

Reviewer #2 (Remarks to the Author):

Pemovska et al, construct a screen of myeloid leukemia cell lines against a metabolic drug library of 243 compounds which target one or many targets involved in cell metabolism which may be preferentially utilized in leukemia cells. The authors grouped cell lines by both the capacity to induce apoptosis, and the genetic and metabolic machinery within the leukemia cells. They described distinct metabolic phenotypes for several group of AML cell lines which emerged from unsupervised linkage clustering, and successfully went on to make efforts to illustrate relationships between mutational genotypes, morphologic phenotypes and metabolic species in each cell line.

We thank the reviewer for appreciating our manuscript and results. We hope we were able to address the raised concerns in this revised manuscript, as described below.

The authors say, quite right, that "cellular metabolism must be considered in the development of new strategies." These efforts contribute, interestingly, to the working understanding of differential metabolic profiles of AML, and how to approach these differences, perhaps, ultimately, in patient samples, in a more high throughput fashion. Curiously, however, the authors make no mentions of some of the more seminal findings in leukemia metabolism made clear in the past 2-3 years, ignoring, for example the targeting of BCL2 family protein – despite the fact that BCL2 inhibition is the largest paradigm change in AML therapy since 7+3 (Line 85 refs 12-17 make small reference to metabolic effects of selective BCL2 family inhibitors and references and introduction in this section should include more focus on BCL2 family proteins). Please see - PMID: 31974170, 33028621

We agree with the reviewer that targeting BCL-2 family proteins has been a game changer for several hematological malignancies, including AML particularly for the elderly unfit patients. The reason for not focusing more on BCL-2 family proteins in the introduction was that BH3 mimetics were not included in the drug library due their major involvement in cellular death and apoptotic signaling network. Nonetheless, we have now added text in the introduction and have cited relevant literature as suggested by the reviewer.

- i. See modified introduction lines 88-97: Even though, there has been substantial advancement in mapping the genetic landscapes of AML, the first approvals for targeted agents came only in the last few years^{12,13}. While the hematologists toolbox has increased, the survival of AML patients remains poor¹⁴. Moreover, prior studies have illustrated metabolic peculiarities in myeloid leukemias that can be exploited for either the development of novel therapies or for particular stratification rationales¹⁵⁻²¹. For instance, the development and clinical utility of IDH1/2 inhibitors for IDH1/2 mutant AML²²⁻²⁵ as wells as BCL-2 inhibitors²⁶ illustrates that untangling metabolic changes could provide therapeutic avenues in AML.
- ii. See updated reference list:

12 Rowe, J. M. Will new agents impact survival in AML? *Best Pract Res Clin Haematol* **32**, 101094, doi:10.1016/j.beha.2019.101094 (2019).

13 DiNardo, C. D. Which novel agents hold the greatest promise in AML? *Best Pract Res Clin Haematol* **32**, 101106, doi:10.1016/j.beha.2019.101106 (2019).

21 Pei, S. *et al.* Monocytic Subclones Confer Resistance to Venetoclax-Based Therapy in Patients with Acute Myeloid Leukemia. *Cancer Discov* **10**, 536-551, doi:10.1158/2159-8290.CD-19-0710 (2020).

22 DiNardo, C. D. *et al.* Durable Remissions with Ivosidenib in IDH1-Mutated Relapsed or Refractory AML. *N Engl J Med* **378**, 2386-2398, doi:10.1056/NEJMoa1716984 (2018).

23 Pollyea, D. A. *et al.* Enasidenib, an inhibitor of mutant IDH2 proteins, induces durable remissions in older patients with newly diagnosed acute myeloid leukemia. *Leukemia* **33**, 2575-2584, doi:10.1038/s41375-019-0472-2 (2019).

24 Roboz, G. J. *et al.* Ivosidenib induces deep durable remissions in patients with newly diagnosed IDH1-mutant acute myeloid leukemia. *Blood* **135**, 463-471, doi:10.1182/blood.2019002140 (2020).

25 Stein, E. M. *et al.* Enasidenib in mutant IDH2 relapsed or refractory acute myeloid leukemia. *Blood* **130**, 722-731, doi:10.1182/blood-2017-04-779405 (2017).

26 Pan, R. *et al.* Selective BCL-2 inhibition by ABT-199 causes on-target cell death in acute myeloid leukemia. *Cancer Discov* **4**, 362-375, doi:10.1158/2159-8290.CD-13-0609 (2014).

Further, while the authors concede the plasticity of the metabolic states, the manuscript's use of the cell line basal metabolic states without addressing the affect of specific treatment on metabolic addiction implies a lack of dynamism. Is it not likely that the metabolic dependencies are influenced by therapies as has been shown in a variety of examples in AML? Can the authors further illustrate how therapies may even "prime" cells for metabolic dependency (as a function of resistance) and then succumb to an alternative treatment?

To explore whether metabolic dependencies can be influenced by drug exposure we now tested the entire drug library in HL-60 cells in the presence of either DMSO, 100nM pictilisib (PI3Ki), 100nM GSK2194069 (FASNi), and 100nM AZD3965 (SLC16A1/MCT1i) over a 72h incubation period and measured cell viability. We found that exposure to AZD3965 led to a decreased sensitivity to rotenone targeting mitochondrial complex I. This finding suggests that upon SLC16A1 inhibition the cells underwent a metabolic switch thereby becoming less dependent on oxidative phosphorylation for their metabolism and presumably more glycolytic and dependent on SLC16A3 for export of lactate.

Moreover, we also observed an increased sensitivity to several inhibitors targeting nucleotide and lipid metabolism in response to pictilisib and AZD3965 treatment. We did not detect any pronounced differences in sensitivities in response to FASN inhibition. These results indicate that therapies can certainly

alter the metabolic profile of cells and that this can be a plausible mechanism of drug resistance.

- i. See new Figure 8

ii. See new Figure 8 legend: Metabolic inhibitors can alter metabolic dependencies. a, Spearman correlation of drug responses in HL-60 cells in response to 72h 100nM AZD3965, 100nM pictilisib, or 100 nM GSK2194069

treatment quantified as AUC values. Drugs that showed a reduced sensitivity in response to drug treatment are marked in blue, whereas drugs that showed an increased sensitivity are marked in red. b, Dose response curves of the drugs highlighted in a as percentage survival. Black curves are baseline and red curves are in response to the respective drugs. Error bars indicate standard deviation of the mean of technical duplicates.

- iii. See edited main text lines 398-413: Metabolic vulnerabilities can be altered by chemical perturbation: To determine whether metabolic vulnerabilities can be influenced by drug exposure we re-tested the entire drug library in HL-60 cells in the presence of vehicle (DMSO) or 100nM AZD3965, pictilisib, or GSK2194069 for 72 hours (Fig. 8a,b). Even though HL-60 do not exhibit sensitivity to AZD3965 at baseline, treatment with AZD3965 resulted in reduced sensitivity to the mitochondrial complex I inhibitor rotenone and increased sensitivity to anti-folates, PI3K pathway and oxidative stress inhibitors. This finding suggests that upon SLC16A1 inhibition the cells undergo a metabolic switch thereby becoming more reliant on glycolysis for their metabolic needs and SLC16A3 for lactate transport. Moreover, in response to pictilisib exposure we detected an increased sensitivity to a number of inhibitors targeting lipid metabolism (TOFA, Ro 48-8071, perhexiline, caffeic acid phenethyl ester, SKI II, lestaurtinib) and nucleotide metabolism (cladribine and cytarabine). While we did not observe any pronounced difference in drug sensitivity in response to FASN inhibition, these results indicate that drugs can modify the metabolic phenotype of cells and this could serve as a plausible mechanism of drug resistance.
- iv. See updated methods lines 597-602: Evaluation if metabolic inhibitors can alter metabolic vulnerabilities was performed by rescreening the entire drug library as described above (each drug tested in 5-different concentrations in duplicate) in HL-60 cells in the presence of either DMSO, 100nM AZD3965, 100nM pictilisib, or 100nM GSK2194069. Data analysis was the same as for the original screen with overall drug response profiles being compared with a Spearman correlation analysis performed in GraphPad Prism 9.

The association made between genotype and phenotype is critical, but previously explored (PMID: 31974170, PMID: 31296572) and not referenced by the authors.

The representative previously known and novel phenotype to genotype associations are based on the in vitro experimental drug sensitivity data generated in this study (see updated Supplementary Fig. 5) where we performed multiple t-tests comparing the drug sensitivity observed in mutated and wildtype cases and significant associations were visualized in Figure 3B. We acknowledge that some of the identified associations have been previously described in different cellular contexts, but to us, this further indicates that our approach is valid and could identify biologically meaningful associations. We have now better acknowledged previously explored associations in the main text and have cited the suggested studies from reviewer 1 and this reviewer in the discussion.

- i. See edited main text lines 252-255: Systematic comparison of mutant vs. wild-type cases (where at least 3 mutant cell lines could be identified), revealed statistically significant novel and previously known correlations between *FLT3* mutations and sensitivity to 5-FU, lestaurtinib^{31,32}, and PF-02545920.
- ii. See updated discussion lines 515-518: In fact, previous studies have linked lack of drug response or the presence of drug resistance to specific intrinsic molecular and/or metabolic cellular properties^{21,69}, that could guide the exploitation of novel metabolic vulnerabilities by alternative chemical agents.
- iii. See updated references:

21 Pei, S. *et al.* Monocytic Subclones Confer Resistance to Venetoclax-Based Therapy in Patients with Acute Myeloid Leukemia. *Cancer Discov* **10**, 536-551, doi:10.1158/2159-8290.CD-19-0710 (2020).

31 Knapper, S. *et al.* A phase 2 trial of the FLT3 inhibitor lestaurtinib (CEP701) as first-line treatment for older patients with acute myeloid leukemia not considered fit for intensive chemotherapy. *Blood* **108**, 3262-3270, doi:10.1182/blood-2006-04-015560 (2006).

32 Knapper, S. *et al.* The effects of lestaurtinib (CEP701) and PKC412 on primary AML blasts: the induction of cytotoxicity varies with dependence on FLT3 signaling in both FLT3-mutated and wild-type cases. *Blood* **108**, 3494-3503, doi:10.1182/blood-2006-04-015487 (2006).

69 Kuusanmaki, H. *et al.* Phenotype-based drug screening reveals association between venetoclax response and differentiation stage in acute myeloid leukemia. *Haematologica* **105**, 708-720, doi:10.3324/haematol.2018.214882 (2020).

This manuscript is a thoughtful compilation – and the first to my knowledge – of systematic use of a drug library (CLIMET) to metabolically profile leukemia (or any other tumor) cells, at least to the point that one can imagine directing therapy. Would this exploration not be better served with the addition of a series of annotated patient samples? (Murine xenografts testing the compounds in the engrafted patient samples would be even better, but maybe impractical for this study). At least use of patient samples, with, ideally some attempt to associate any clinical successes to postulations which could be made by the profiling, is necessary.

We agree with the reviewer and now added a metabolic drug profiling dataset across 15 different AML and CML patient samples to the manuscript. The patient characteristics are described in a new Supplementary Table 3 and the drug screening data is provided in a new Supplementary Table 4. Comparison between the cell line and patient sample data indicated comparable metabolic vulnerability profiles with higher efficacy of numerous nucleotide metabolism inhibitors in the cell lines, likely as a result of higher cell proliferation during the drug screening assay as compared to the primary patient cells. In contrast,

patient samples were more susceptible to several lipid and fatty acid metabolism drugs. Over 64% of the compounds did not display a differential response, including 14 out of the 18 inhibitors that strongly contributed to the metabolic functional stratification of the cell lines depicted in Fig. 2c. Two AML patient samples (Pat. 4 and Pat. 6) were relapsed or refractory to cytarabine prior to sampling and those patient samples exhibited limited sensitivity to cytarabine in the drug profiling, indicating that our results are generally consistent with the in vivo situation.

- i. See new Figure 9

- ii. See new Figure 9 legend: Metabolic drug sensitivity profiles are generally comparable between myeloid cancer patient samples and cell lines. a, A volcano plot depicting the comparison of 70 drug responses between 15 myeloid cancer patient samples and 15 myeloid cancer cell lines. Each point represents one drug and statistical significance was assessed by Mann-Whitney test for each drug. The \log_2 AUC (fold change) is plotted on the x-axis and negative \log_{10} of the adjusted P -values (using the two-stage step-up Benjamini, Krieger, and Yekutieli correction) is plotted on the y-axis. The horizontal dotted line signifies the 5% significance cutoff. Colored dots indicate drugs that exhibit higher sensitivity in either patient samples (red) or cell lines (blue). b, Spearman correlation of drug response profiles between myeloid leukemia patient samples and cell lines. Drugs that were highlighted to strongly contribute to the cell line grouping are marked in green. c, Heatmap of metabolic drug sensitivity profiles expressed as AUC in 12 AML and 3 CML patient samples. Clustering was performed with the complete linkage method and Spearman (drugs) and Euclidean (patient samples) distance measures. The color bars to the left and above show sample annotations with grey box depicting data not available and white box depicting no mutation detected. Drug AUC distribution (min to max) box plot is shown on the right side of the graph with the black line in the middle of box depicting the median (the mean is indicated with a + sign).
- iii. See update methods lines 743-755: Primary patient cells drug profiling. Vitaly frozen mononuclear cells (MNCs) from patients with AML ($n = 13$) and CML ($n = 3$) after written informed consent were used for this study. Ethical approval was granted by the Ethics Commission of the Medical University of Vienna (Ethik Kommission 1676/2016). Patient characteristics are described in Supplementary Table 3. Mononuclear cells (MNCs) from bone marrow aspirates and peripheral blood were purified using Ficoll density gradient (Lymphoprep; STEMCELL Technologies). Upon thawing cells were maintained in RPMI 1640 medium (Gibco) with 10% fetal bovine serum (Gibco), penicillin (100 U/mL), and streptomycin (100 $\mu\text{g}/\text{mL}$). Patient cells were thawed, counted, and 10 000 cells per well were seeded onto pre-drugged 384-well plates. Drug screening and analysis were performed as described for the cell lines, with 70 drugs being tested in the patient samples. Mutational data was extracted from patient charts and referral reports where available.

Figure 2B summarizes that the most effect agents again AML cell lines are gemcitabine, toptecan, methotrexate, digoxin, among others, and while most of these agents have been in use for over 50 years, none of them are routinely used in the treatment of myeloid leukemia. This represents a disconnect between the findings and the application of the findings.

Figure 2b summarizes the average drug response detected in the cell line panel tested and therefore certain drugs are highlighted that may exhibit an overall cell toxicity profile as is the case of gemcitabine, topotecan, digoxin, digitoxigenin. We have previously shown that chemotherapeutics have a narrower therapeutic window and their efficacy in vivo is more challenging to predict based on ex vivo drug sensitivity data (see also Pemovska et al, Cancer Discovery 2013). We have also plotted the top selectively effective compounds

for the reviewer's discretion based on analysis depicted in Figure 5a (Rebuttal Figure 4). From the 25 compounds shown in Figure 2b 14 (56%) exhibit selective drug responses. If the reviewer and editor feel strong to have the top selectively effective compounds better in the main figure instead of the top effective compounds we would do so, however, we feel that in the first instance it is better to provide an overview of the data distribution without delving too much into selectivity, an aspect that is introduced and followed up later in the manuscript.

i. See Rebuttal Figure 4 and legend

Rebuttal Figure 4. Scatter dot plot showing the average top selectively effective compounds in the myeloid cell lines tested with each circle representing one cell line and bars labeled per metabolic process the corresponding compounds target. Graph shows mean \pm SD. The compounds were chosen based on comparison of the average AUC values to ones detected in each individual cell line (sAUC) with hits ranked according to sAUC value (cutoff ≥ 0.16).

Figure 2E is a clever figure representing separation of the AML cell lines by OCAR/ECAR ratio. The meaning of this differential ratio between each leukemia, however, is not well described in the manuscript.

We thank the reviewer for appreciating the data presentation of the overall basal metabolic profiles of the cell lines studied. The plotting of basal ECAR and OCR levels measured with the Seahorse XF Cell Mito Stress Test Kit provides a snapshot of the bioenergetics profiles of the cell lines. In addition, in Supplementary Fig. 3c the OCR/ECAR ratio at baseline represents the mean OCR at baseline (of 8 replicates) divided by the mean ECAR at baseline (of 8 replicates) for each individual cell line from the second baseline timepoint at approximately 8

minutes after assay start. We have now provided the source data for each figure and have clarified the meaning of the ratio in the figure legend.

- i. See edited Supplementary Fig. 3c,d legend: c-d, Box and whiskers plot showing the basal bioenergetics state (the mean OCR at baseline divided by the mean ECAR at baseline for each individual cell line from the second baseline timepoint at approximately 8 minutes after assay start) and percentage spare respiratory capacity of the myeloid cell lines.

Reviewer #3 (Remarks to the Author):

The manuscript from Pemovska et al describes the assembly of a metabolism-biased compound library that is used to profile AML cells in hopes of identifying metabolic cancer vulnerabilities. It is an interesting approach, but overall, this reviewer found too many instances where the data were not sufficient to support the claims. I will be specific about these instances below. The most solidly supported claim is the identification of synthetic lethality between MCT1 and MCT4 in which cells with low MCT4 expression are hypersensitive to an MCT1 inhibitor. However, as is referenced in the discussion, this has already been discovered with both genetic and chemical approaches (using the same compound as is used in this manuscript).

We thank the reviewer for finding our approach interesting. The case example for determinants of sensitivity to AZD3965, while previously shown in certain cellular contexts provides proof-of-principle of our approach and highlights that biologically meaningful cellular vulnerabilities can be identified by drug sensitivity testing. Moreover, we have further mechanistically investigated the vulnerability to PI3K and FASN inhibition. The main focus of the manuscript is the description of a novel metabolic drug library and its utility in identifying metabolic vulnerabilities and we hope that we have sufficiently addressed the raised concerns in the revised manuscript and the rebuttal letter.

Major notes:

- the clustering of cell lines into distinct groups in Fig2 is based on 19 compounds of the >200 in the library. Why were these selected in particular and why was the entire library not used for clustering analysis (Ext. Fig2)? Do the groups remain the same when clustering on the entire library vs on the 19 compounds?

As explaining previously to major point 1&2 of reviewer 1 the classification in Figure 2c is derived by performing hierarchical clustering (complete linkage method and Spearman (drugs) and Euclidean (cell lines) distance measures) of the metabolic drug responses profiles of the 15 tested cell lines to the 77 compounds that had a cell viability inhibiting effect in at least one cell line. The Figure 2c is a snapshot of the compounds that contribute the most to the classification with statistics and the complete heatmap is provided as Supplementary Fig. 2. The dendrogram clustering of the cell lines is the same between these two figures. In the Figure 2c only the compounds that significantly influenced the grouping of the cell lines were shown, whereas in the Supplementary Fig. 2 also compounds that had a consistent sensitivity profile within a group were highlighted albeit not reaching statistical significance. However, we acknowledge that this may be misleading, and thus we reanalyzed the data and decided to highlight only the compounds that reached statistical significance prior to multiple testing correction and show both the original and adjusted P-value (see updated Figure 2 and updated Supplementary Fig. 2).

- i. See updated Figure 2c

- ii. See updated Figure 2c legend: c, Heatmap illustrating the functional grouping of myeloid cancer cells in 5 taxonomic groups based on metabolic vulnerability profiles. While each cell line had an overall distinct drug sensitivity profile, the activity of 18 compounds in particular significantly functionally stratified the myeloid cancer cells lines (an ordinary one-way ANOVA analysis * $P \leq 0.05$; ** $P \leq 0.01$; *** $P \leq 0.001$; **** $P \leq 0.0001$; FDR of 10% was deemed significant; * $Padj \leq 0.1$; ** $Padj \leq 0.05$; *** $Padj \leq 0.01$; ns not significant).
- iii. See updated Supplementary Fig. 2

- iv. See edited Supplementary Fig. 2 legend: Metabolic drug sensitivity testing classifies myeloid leukemia cell lines in 5 functional groups. Unsupervised hierarchical clustering of the 77 viability affecting drugs is presented as a heatmap across 15 cell lines. Clustering was performed using the complete linkage clustering method and Euclidian and Spearman distance measures of the AUC values of cell lines and drugs, respectively. The response profiles of

several compounds and compounds classes display differential effects across the cell lines tested and are outlined with black boxes. The reproducibility of the clusters was evaluated by randomly resampling 10,000 bootstrap samples of the original dataset and calculating the frequency at which each cluster appears in the hierarchical clustering of the bootstrap replicates. The numbers in the dendrogram tree of the cell lines indicate the approximately unbiased (AU) empirical frequencies (0-100%) from the multiscale bootstrap resampling implemented in the Pvcust R-package.

- there are a number of claims made as possible consequences of the clustering analysis that go beyond just pointing out the differences in sensitivities among the group. e.g. that Group I may be addicted to ROS production due to their sensitivity profile. However, these claims are not addressed with experiments and the first half of the manuscript reads as a significant amount of speculation.

We agree with the reviewer that in the initial manuscript there were certain unsubstantiated claims like the one mentioned above and we apologize for overstating some of these observations. In the revised text we have now removed the speculative language. Nonetheless, we also investigated the significance of the five identified functional subgroups by assessing their clustering reproducibility by randomly resampling bootstrap samples of the original data and then calculating the frequency that each cluster appears in the hierarchical clustering of the bootstrap replicates. This information is now included in the updated Supplementary Figure 2. As highlighted in the new supplementary figure, the subclusters appeared relatively robust even though some samples clearly match the subtyping pattern better than others. Overall, we expect that the future assessment of larger sample cohorts will refine and complement these subtypes.

- i. See updated Figure 2 and Supplementary Fig. 2 above
- ii. See edited manuscript text lines 166-169: While each cell line had a unique metabolic vulnerability profile, the activity of 18 compounds in particular stratified the cells lines into five robust and distinct functional groups (Fig. 2c and Supplementary Fig. 2).
- iii. See updated Methods section pertaining to this analysis lines 591-593: Reproducibility of the clustering was assessed with the bootstrap resampling method using the Pvcust R-package⁷².

- on the same note, on lines 212-215, it is claimed that “Critically, the abundance of specific metabolites could be matched to the drug sensitivity defined groups, validating the notion that the similarities in functional phenotypes of the cell lines in our analysis are anchored to particular metabolic vulnerabilities.” However, this is not supported by the data. Any multidimensional dataset (RNA-seq, metabolomics, proteomics, etc.) will provide an opportunity to find differences in individual transcripts/metabolites/proteins that follow some supervised categorization. It does not mean that these differences are functionally related to the categorization or the phenotype of interest. It does not mean that these metabolites are functionally related to any known or unknown metabolic vulnerabilities in these groups or that the metabolites are responsible for stratifying the groups based on their responses to the

metabolic drugs. The key questions would be: can the same cell line groups be clustered into their original drug response clusters based on these metabolites? can you alter the metabolite levels and change the responses to the drugs that are uniquely active in one or more groups? those are the key questions that would support this type of claim.

The analysis for identifying metabolites that are more or less abundant in the previously defined metabolic drug sensitivity functional groups is based on statistics that is corrected for multiple testing analysis. Of course, this analysis does not provide direct evidence for correlation or causality between the drug sensitivity and metabolomics data. However, we performed metabolite enrichment analysis using Metaboanalyst 4.0 (<https://www.metaboanalyst.ca>) and integrated the information of the drug targets of the sensitive drugs in a particular group as well as the pathway/process in which the metabolites play a role to interpret the findings. Therefore, one can state that those findings can be associated with the observed drug sensitivity profiles. Thus, we have changed the wording of the sentence to tone down the conclusions.

- i. See edited main text lines 234-237: Taken together, the identified cell line groups have distinct metabolic phenotypes, which suggest different nutrient acquisition dependencies that could be associated to the drug sensitivity profiles.

To address whether the same cell line groups can be clustered into their original drug response clusters based on the metabolomics data we have now re-clustered the cells lines based on the significantly altered metabolites depicted in Supplementary Fig. 4. As it can be seen in the Rebuttal Figure 5, overall, this is not the case, despite some cell lines still clustering together as per the drug screening data. Several reasons can be contemplated: that the clustering is based on the metabolomics profiles available for 13 cell lines rather than 15, and on the levels of the selected 57 metabolites levels of which were found to be significantly different between at least two groups, and that drug sensitivity to metabolic and/or other inhibitors may be dependent on additional features. Therefore, in the revised manuscript we have removed unsubstantiated statements.

- ii. See Rebuttal Figure 5 and legend

Rebuttal Figure 5. Hierarchical clustering across 13 cell lines based on the levels of 57 metabolites, represented as Z-scores, and visualized with a heatmap, which were significantly different between at least two of the drug sensitivity pre-defined groups with a two-way ANOVA analysis (FDR of 10% was deemed significant). For analysis the clean imputed Log₂ transformed values provided in the original publication were used. Clustering was performed using the complete linkage clustering method and Euclidian (cell lines) and Spearman (metabolites) distance measures.

Moreover, as also touched upon by reviewer 2, to explore whether metabolic dependencies can be influenced by drug exposure we tested the entire drug library (each drug in 5 different concentrations in duplicate as the original screen) in HL-60 cells in the presence of either DMSO, 100nM pictilisib (PI3Ki),

100nM GSK2194069 (FASNi), and 100nM AZD3965 (SLC16A1/MCT1i) over a 72h incubation period and measured cell viability. We found that exposure to AZD3965 led to a decreased sensitivity to rotenone, which targets the mitochondrial complex I. This finding suggests that upon SLC16A1 inhibition the cells underwent a metabolic switch thereby becoming less dependent on oxidative phosphorylation for their metabolism and presumably more glycolytic and dependent on SLC16A3 for export of lactate.

Moreover, we also observed an increased sensitivity to several inhibitors targeting nucleotide and lipid metabolism in response to pictilisib and AZD3965 treatment. We did not detect any pronounced differences in sensitivities in response to FASN inhibition. These results indicate that therapies can certainly alter the metabolic profile of cells and that this can be a plausible mechanism of drug resistance.

iii. See new Figure 8

iv. See new Figure 8 legend: Metabolic inhibitors can alter metabolic dependencies. a, Spearman correlation of drug responses in HL-60 cells in response to 72h 100nM AZD3965, 100nM pictilisib, or 100 nM GSK2194069 treatment quantified as AUC values. Drugs that showed a reduced sensitivity in response to drug treatment are marked in blue, whereas drugs that showed an increased sensitivity are marked in red. b, Dose response

curves of the drugs highlighted in a as percentage survival. Black curves are baseline and red curves are in response to the respective drugs. Error bars indicate standard deviation of the mean of technical duplicates.

- v. See edited main text lines 398-413: Metabolic vulnerabilities can be altered by chemical perturbation. To determine whether metabolic vulnerabilities can be influenced by drug exposure we re-tested the entire drug library in HL-60 cells in the presence of vehicle (DMSO) or 100nM AZD3965, pictilisib, or GSK2194069 for 72 hours (Fig. 8a,b). Even though HL-60 do not exhibit sensitivity to AZD3965 at baseline, treatment with AZD3965 resulted in reduced sensitivity to the mitochondrial complex I inhibitor rotenone and increased sensitivity to anti-folates, PI3K pathway and oxidative stress inhibitors. This finding suggests that upon SLC16A1 inhibition the cells undergo a metabolic switch thereby becoming more reliant on glycolysis for their metabolic needs and SLC16A3 for lactate transport. Moreover, in response to pictilisib exposure we detected an increased sensitivity to a number of inhibitors targeting lipid metabolism (TOFA, Ro 48-8071, perhexiline, caffeic acid phenethyl ester, SKI II, lestaurtinib) and nucleotide metabolism (cladribine and cytarabine). While we did not observe any pronounced difference in drug sensitivity in response to FASN inhibition, these results indicate that drugs can modify the metabolic phenotype of cells and this could serve as a plausible mechanism of drug resistance.
- vi. See updated methods lines 597-602: Evaluation if metabolic inhibitors can alter metabolic vulnerabilities was performed by rescreening the entire drug library as described above (each drug tested in 5-different concentrations in duplicate) in HL-60 cells in the presence of either DMSO, 100nM AZD3965, 100nM pictilisib, or 100nM GSK2194069. Data analysis was the same as for the original screen with overall drug response profiles being compared with a Spearman correlation analysis performed in GraphPad Prism 9.

- the pairwise analysis is used to suggest that the correlation between drugs addressing distinct target classes is evidence that the pathways/compounds are functionally related (Line 276) and could be used leveraged for synergy (Line 407-9). That is, that targeting PI3K and FA synthesis would be synergistic. But this is not tested. The pairwise correlations could suggest a number of other possibilities, for instance that these drugs have shared off-targets, not that the pathways of the intended targets are functionally related.

The correlogram depicts positive and negative drug response correlations across the entire cell line metabolic drug sensitivity dataset. This analysis identified drugs that exhibited co-occurring or mutually exclusive responses, which can come from the drug response profile of a single cell line or multiple cell lines. In the initial manuscript we speculated that drugs with co-occurring sensitivity patterns could have a combinatorial effect in the right cellular context. We agree with the reviewer that other reasons for the interactions are possible and have altered the manuscript to reflect that. Nevertheless, we have now further investigated the potential drug interactions by performing drug synergy experiments (8x8 matrices) of 11 different drug combinations across four different cell lines (HL-60, Mono-Mac-6, BV-173, and MV4-11) that best

capture the significant drug-drug correlations depicted in Figure 4. The results indicated that in majority of cases combinatorial effects were detected using the SynergyFinder tool (lanevski et al; Bioinformatics 2017; lanevski et al; Nucleic Acid Research 2020) with the Bliss independence model. However, there were a few drug combinations pairs (e.g., lestaurtinib with 5-fluorouracil) where marked single agent activity was detected but this resulted in an antagonistic combinatorial effect. We also tested the combinatorial effect of pictilisib and GSK2194069 in this set of experiments and found a synergistic combinatorial effect in HL-60 cells.

i. See new Figure 4b

ii. See new Figure 4b legend: b, Heatmap showing the deviation from Bliss independence score for each tested combination and cell lines. Synergy is denoted in red while antagonism is shown in blue. Analysis was performed using the SynergyFinder tool^{77,78}. The individual drug sensitivity is shown within the heatmap squares per cell line with + indicating sensitivity; +/- indicating moderate sensitivity, and - indicating no sensitivity. The first sign refers to the sensitivity of the first listed drug of the combination and the second sign to the sensitivity of the second drug.

iii. See new Supplementary Fig. 6

HL-60

Mono-Mac-6

Bliss synergy score: **-1.583**
 -30 -20 -10 0 10 20 30

Bliss synergy score: **-3.075**
 -40 -20 0 20 40

Bliss synergy score: **2.129**
 -40 -20 0 20 40

Bliss synergy score: **3.316**
 -30 -20 -10 0 10 20 30

Bliss synergy score: **-0.038**
 -40 -30 -20 -10 0 10 20 30 40

Bliss synergy score: **-2.966**
 -30 -20 -10 0 10 20 30

Bliss synergy score: **-5.222**
 -40 -20 0 20 40

Bliss synergy score: **-2.178**
 -40 -20 0 20 40

Bliss synergy score: **3.269**
 -40 -20 0 20 40

Bliss synergy score: **-4.162**
 -40 -30 -20 -10 0 10 20 30 40

Bliss synergy score: **7.966**
 -40 -30 -20 -10 0 10 20 30 40

BV-173

Bliss synergy score: 1.033 Bliss synergy score: -4.795 Bliss synergy score: 1.284

Bliss synergy score: 4.085 Bliss synergy score: -0.61 Bliss synergy score: 1.962

Bliss synergy score: 0.598 Bliss synergy score: 3.713 Bliss synergy score: -5.216

Bliss synergy score: -1.574 Bliss synergy score: -2.965

MV4-11

iv. See new Supplementary Fig.6 legend: 3D drug synergy visualizations for 11 different drug-drug interactions in HL-60, Mono-Mac-6, BV-173, and MV4-11 cells. Data was generated in 8x8 drug synergy matrices and analyzed using the SynergyFinder tool using the Bliss independence model with % survival values at each tested concentration relative to DMSO (negative control) and 10 μ M

bortezomib (positive control). The synergy score for a drug combination is averaged over all the dose combination measurements. The 2D and 3D synergy maps highlight synergistic and antagonistic dose regions in red and green, respectively.

- v. See edited main text lines 286-295: To investigate the identified drug interactions further we performed drug combinatorial screens in four different cell lines (HL-60, Mono-Mac-6, BV-173, and MV4-11) that best captured the significant drug-drug correlations. The results showed that in majority of cases combinatorial effects were observed as hypothesized (Fig. 4b and Supplementary Fig. 6). However, there were a few drug combination pairs (e.g., lestaurtinib and 5-fluorouracil, and dorsomorphin and rosiglitazone) that exhibited marked single agent activity but an antagonistic combinatorial effect. Taken together this analysis could facilitate elucidation of the context-specific molecular mechanisms of action and foster a rationale for context-dependent drug combinations, though alternative bases for the observed interactions should also be contemplated.
- vi. See edited main text lines 339-344: We, further, detected a combinatorial effect of pictilisib and GSK2194069 in HL-60 cells (Supplementary Fig. 6). Moreover, pictilisib exposure led to a dose-dependent reduction of *FASN* expression (Supplementary Fig. 8d). Taken together, our data illustrate that HL-60 are dependent on PI3K signaling and *de novo* FA synthesis and conceivably these pathways and processes are functionally linked in this cellular and disease context.
- vii. See new methods subsection lines 644-653: Drug combination screens. The significant drug-drug interactions identified in Figure 4a were evaluated by drug synergy screens in four different cell lines HL-60, Mono-Mac-6, BV-173, and MV4-11. Briefly, 8x8 synergy matrices were tested for each drug combination and cell lines with majority of drugs being tested at 5, 10, 50, 100, 500, 1000, 5000nM concentration with the exception of rosiglitazone and lestaurtinib that were tested at 1, 5, 10, 50, 100, 500, and 1000nM concentration. The experimental procedure and readouts were as described above for single agent screening. Combinatorial effects were evaluated and visualized using the SynergyFinder^{77,78} web portal (<https://synergyfinder.fimm.fi>) using the Bliss independent model.

- part of the evidence for FA sensitivity are viability effects in HL-60 cells when re-testing *FASN* inhibitors up to 100 μ M. This is a drastically high concentration which could have a huge number of relevant off-targets and could easily be toxic to most or all cell lines in the study.

This data was added to illustrate why in the initial screen we did not detect a response to orlistat and c75, two inhibitors frequently used in literature as FASN inhibitors. However, they are not very potent compounds and exhibit activity in the high μ M range (see Rae C, et al. Radiat Res. 2015 Nov;184(5):482-93 and Kridel SJ, et al. Cancer Res. 2004 Mar 15;64(6):2070-5). If the reviewer and editor feel that we should better remove this data, we would do so.

- To understand the sensitivity of HL-60 cells to pictilisib, the authors noticed low expression of PI3K family members other than PIK3CB/D (the pictilisib targets) in HL-60. This is claimed to be the basis of sensitivity in HL-60 cells, but it is not tested. A rescue experiment expressing other PIK3 family members would be required. Especially given that the expression profile of PI3K family members in HL60 cells looks very similar to other AML cell lines, like MV4-11 and MOLM-13. Claiming that this could be used to select patients for PI3K therapeutics is not appropriate.

To address this point, we have now generated HL-60 and MV4-11 class I PI3K overexpression cells and screened for pictilisib sensitivity. We choose MV4-11 as the reviewer rightly points out it had the most similar gene expression profile of the class I PI3K family members to HL-60 cells. We found that pictilisib sensitivity was enhanced in HL-60 cells overexpressing PIK3CA, PIK3CB or PIK3CD in comparison to HL-60 cells lentivirally transduced with an empty vector. In contrast, we detected a moderate increase in pictilisib sensitivity in PIK3CA and PIK3CG MV4-11 overexpressing cells, while there was no difference in sensitivity in PIK3CB and a minor loss in sensitivity in PIK3CD overexpressing cells. These findings indicate that there are perhaps different isoforms on which sensitivity to pictilisib is dependent in these two cell lines and that no one specific PI3K isoform influences sensitivity to pictilisib. Moreover, our findings further suggest that a combination of isoforms such as PIK3CA and either PIK3CB or D to determine pictilisib susceptibility. We agree with the reviewer that the statement that lower expression of PI3K family members could be used for selecting patients for PI3K inhibitor treatment was too strong and have removed it in the revised manuscript.

i. See new Supplementary Fig. 7d,e

ii. See new Supplementary Fig. 7d,e legend: d-e, Average response to pictilisib in HL-60 and MV4-11 empty vector, and PIK3CA, PIK3CB, PIK3CD or PIK3CG

overexpressing cells. Error bars indicate standard deviation of the mean of technical triplicates. Data is representative of two independent experiments. Moreover, immunoblot analysis of PIK3CA, PIK3CB, PIK3CD, and PIK3CG expression in HL-60 and MV4-11 cells lentivirally transduced with empty vector, *PIK3CA*, *PIK3CB*, *PIK3CD* or *PIK3CG*-cDNA is shown.

- iii. See updated main text lines 314-325: To explore determinants to pictilisib sensitivity, we overexpressed the different class I PI3K isoforms in HL-60 and MV4-11 cells, which have a comparable expression pattern of those genes. Pictilisib sensitivity was enhanced in HL-60 cells overexpressing *PIK3CA*, *PIK3CB* or *PIK3CD* in comparison to HL-60 cells lentivirally transduced with an empty vector (Supplementary Fig. 7d). In contrast, we detected a moderate increase in pictilisib sensitivity in *PIK3CA* and *PIK3CG* MV4-11 overexpressing cells, while there was no difference in sensitivity in *PIK3CB* and a minor loss in sensitivity in *PIK3CD* overexpressing cells (Supplementary Fig. 7e). These findings suggest that there is not one specific PI3K isoform that influences sensitivity to pictilisib in these cells, highlighting the possibility of combination of isoforms such as PIK3CA and either PIK3CB or D to determine pictilisib susceptibility.
- vi. See updated methods lines 703-714: Generation of cells overexpressing *SLC16A1*, *SLC16A3*, *PIK3CA*, *PIK3CB*, *PIK3CD*, and *PIK3CG* was performed by lentiviral cDNA delivery experiments. SLC-encoding cDNAs were obtained as codon-optimized versions from Genescript and transferred into pDONR221 entry vector using BP reaction gateway cloning (Invitrogen). PI3K isoform cDNAs were ordered from Addgene (#81736, #82221, #82222, and #81843). In the case of Addgene plasmid #81843 the stop codon was removed by site directed mutagenesis to enable C-terminal tagging (E0554S, New England Biolabs, Ipswich, USA). Constructs were subsequently transferred into lentiviral expression vectors LEgwSHIB (pRRL-EF1a-gwSH-IRES-BlastR)⁸³ or LE3FgwIP (pRRL-EF1a-3xFLAG-gw-IRES-PuroR) using LR reaction-based gateway cloning (Invitrogen). Cells infected with a corresponding empty lentiviral expression vector LEIB (pRRL-EF1a-IRES-BlastR) or LEIP (pRRL-EF1a-IRES-PuroR) served as negative control.

Lines 735-741: The membranes were incubated with α -SLC16A1 (sc-365501, Santa Cruz Biotechnology), α -SLC16A3 (sc-376140, Santa Cruz Biotechnology), α -FLAG (F1804, Sigma-Aldrich), α -HSP90 (610418, BD Biosciences) or α -AKT (#4685, Cell Signaling Technologies) and visualized with horseradish peroxidase-conjugated secondary antibodies (Jackson ImmunoResearch) utilizing the ECL Western blotting system (Thermo Fisher Scientific).

- The most exciting data and well-supported claim in the manuscript is the identification of MCT1/4 synthetic lethality. However, this has already been discovered and reported multiple times.

As stated above, the work up of AZD3965 as SLC16A1 (MCT1) inhibitor was used as a case study to illustrate the utility of the metabolic drug library in identifying

relevant metabolic vulnerabilities and as an example that synthetic lethality relationships could be uncovered by drug screening.

Minor notes:

- the library is pitched as a way to overcome the difficulties of CRISPR in primary cells or cell lines that are difficult to work with, but all the experiments are done in AML cells, which have been extensively profiled by genome-scale CRISPR screens. Is it true that this library could easily be used to profile primary cells?

We agree with the reviewer and how now also added a metabolic drug profiling dataset across 15 different AML and CML patient samples to the manuscript. The patient characteristics are described in a new Supplementary Table 3 and the drug screening data is provided in a new Supplementary Table 4. Comparison between the cell line and patient sample data indicated comparable metabolic vulnerability profiles with higher efficacy of numerous nucleotide metabolism inhibitors in the cell lines, likely as a result of higher cell proliferation during the drug screening assay as compared to the primary patient cells. In contrast, patient samples were more susceptible to several lipid and fatty acid metabolism drugs. Over 64% of the compounds did not display a differential response, including 14 out of the 18 inhibitors that strongly contributed to the metabolic functional stratification of the cell lines depicted in Fig. 2c. Two AML patient samples (Pat. 4 and Pat. 6) were relapsed or refractory to cytarabine prior to sampling and those patient samples exhibited limited sensitivity to cytarabine in the drug profiling, indicating that our results are generally consistent with the in vivo situation.

i. See new Figure 9

- ii. See new Figure 9 legend: Metabolic drug sensitivity profiles are generally comparable between myeloid cancer patient samples and cell lines. a, A volcano plot depicting the comparison of 70 drug responses between 15 myeloid cancer patient samples and 15 myeloid cancer cell lines. Each point represents one drug and statistical significance was assessed by Mann-Whitney test for each drug. The \log_2 AUC (fold change) is plotted on the x-axis and negative \log_{10} of the adjusted P -values (using the two-stage step-up Benjamini, Krieger, and Yekutieli correction) is plotted on the y-axis. The horizontal dotted line signifies the 5% significance cutoff. Colored dots indicate drugs that exhibit higher sensitivity in either patient samples (red) or cell lines (blue). b, Spearman correlation of drug response profiles between myeloid leukemia patient samples and cell lines. Drugs that were highlighted to strongly contribute to the cell line grouping are marked in green. c, Heatmap of metabolic drug sensitivity profiles expressed as AUC in 12 AML and 3 CML patient samples. Clustering was performed with the complete linkage method and Spearman (drugs) and Euclidean (patient samples) distance measures. The color bars to the left and above show sample annotations with grey box depicting data not available and white box depicting no mutation detected. Drug AUC distribution (min to max) box plot is shown on the right side of the graph with the black line in the middle of box depicting the median (the mean is indicated with a + sign).
- iii. See update methods lines 743-755: Primary patient cells drug profiling. Vially frozen mononuclear cells (MNCs) from patients with AML ($n = 13$) and CML ($n = 3$) after written informed consent were used for this study. Ethical approval was granted by the Ethics Commission of the Medical University of Vienna (Ethik Kommission 1676/2016). Patient characteristics are described in Supplementary Table 3. Mononuclear cells (MNCs) from bone marrow aspirates and peripheral blood were purified using Ficoll density gradient (Lymphoprep; STEMCELL Technologies). Upon thawing cells were maintained in RPMI 1640 medium (Gibco) with 10% fetal bovine serum (Gibco), penicillin (100 U/mL), and streptomycin (100 $\mu\text{g/mL}$). Patient cells were thawed, counted, and 10 000 cells per well were seeded onto pre-drugged 384-well plates. Drug screening and analysis were performed as described for the cell lines, with 70 drugs being tested in the patient samples. Mutational data was extracted from patient charts and referral reports where available.

- indisulam is referenced multiple times in the manuscript. What is the basis for its annotation as a metabolic regulator? As far as I am aware, it's cytotoxicity is due to degradation of the splicing factor RBM39 (science and nat chem biol papers from 2017).

Indisulam is annotated as an carbonic anhydrase IX inhibitor in ChEMBL, CHEBI, PubChem and thus targeting/affecting acid-base balance and energy metabolism (see Supplementary Table 1). Of course we cannot exclude that the sensitivity observed in this study is not due to activity reported in the papers the reviewer is citing.

REVIEWER COMMENTS

Reviewer #1 (Remarks to the Author):

Comments to the authors:

The functional significance and clinical relevance of the suggested stratification is still low and requires further validation. Unless the authors can demonstrate a novel genotype/molecular subtype-metabolic dependency interaction that can be validated in vivo this study would be of limited scope and translational potential.

For comment 1)

Figure 2c. The authors mention that the method used is hierarchical complete linkage clustering. However, this computational method merges different clusters based on shortest distance between observations. While Group I, II, III and V make sense with this approach based on a selected threshold, group IV cannot be justified solely based on the distance between the cell lines. KG1 and K562 cell lines would be closer to Groups I, II and III with this approach. The robustness of the dendrogram is not the issue. On the other hand, the numbers of clusters identified requires further validation. Using a method such as the Average silhouette or Elbow method would answer this question. In addition, the authors should use another unsupervised computational method e.g. non-negative matrix factorization (NMF) to validate A) the optimal number of clusters, B) the overlap with the hierarchical clustering, and C) the drugs that contribute the most to the stratification.

For comment 2)

While the Authors mention in line 255 that they identify novel correlations between mutational status and sensitivity to a drug, all this information is correlative and there are no data to support that the results are not cell line specific, nor an in vitro artefact. Validating these findings with a few experiments like those that were performed to address comment 3) would provide higher impact to this study and potentially identify new therapeutic targets of high relevance in myeloid leukemia. For example, using isogenic models or overexpressing FLT3 in a cell line with WT-FLT3 and testing its sensitivity to both known and newly identified compounds could serve to validate the correlations identified. The same could be done for RAS or TP53 that the authors describe in lines 258-260. These models could also be used to assess whether any of these driver mutations could drive the stratification of the different clusters, i.e. whether the introduction of a genetic alteration in a WT cell line setting could change its classification from one cluster to another. Finally, given that the screen was performed in unphysiologically relevant media, and it aims to identify metabolic vulnerabilities, this study would be of limited scope and translational potential unless the major findings are validated in vivo.

For comment 3)

Comment has been addressed. Validation of drug-response associations in cell lines highlights the discrepancy between predicted and real response to combination therapy but also provides a good tool for synergistic drug response discovery.

For comment 4)

Comment has been addressed and results of follow-up experiments show the non-specificity of the response.

For comment 5)

While the dose response curves have been added for the 5 drugs mentioned, it would be useful to demonstrate if this sensitivity to FASN inhibitors is cell line dependent or is observed across all the cell lines of cluster III compared to cells from other clusters. Even if non-significant across different clusters using the initial concentrations, the cluster containing HL60 might still show higher sensitivity to FASN inhibitors and therefore demonstrate a higher impact of the classification.

For comment 6)

Comment has been addressed. New data indicate that PIK3CA inhibition (albeit at concentration that largely affects cell viability) reduces FASN expression at the mRNA level. It would be expected that KCL-22, the only cell line with PIK3CA mutation would also be more sensitive to FASN inhibition and/or show synergism with pictisilib.

Minor comment: Fig. 5 has been relabelled fig.4 in the revised manuscript.

Reviewer #3 (Remarks to the Author):

I thank the authors for their detailed responses to my previous comments. Clearly a great amount of effort has gone into the initial and revised manuscripts. However, I still find several claims to be unsupported by the data.

In the first submission, the authors claimed that lower overall expression of PI3K genes formed the basis of the heightened sensitivity of HL60 cells to pictilisib. I suggested that to make this claim would require overexpressing PI3K genes in HL60 cells to test whether sensitivity is diminished. The authors have attempted to do this but have not found that the low expression of PI3K genes is responsible for the increased sensitivity in HL60s. I have several concerns with these new data.

First, in the new supplementary 7d,e, HL60 empty vector control cells are no more sensitive than MV4-11 empty vector control cells to pictilisib suggesting that the heightened sensitivity of HL60 observed in Fig 4b may not be reproducible.

Additionally, the authors have included sensitivity data for HL60 overexpressing PIK3CA and PIK3CG, but neither of these genes appear to be successfully expressed by western blot.

Finally, if lower expression of PI3K genes is responsible for increased sensitivity in HL60 cells, then how would the authors explain the left-shifted curves for HL60 overexpressing PIK3CA, B, and C? If anything, this would suggest lower expression would diminish sensitivity to pictilisib, not increase it. Nevertheless, the authors do not acknowledge that low PI3K expression conferred increased pictilisib sensitivity appears to be incorrect.

REVIEWER COMMENTS

Reviewer #1 (Remarks to the Author):

Comments to the authors:

The functional significance and clinical relevance of the suggested stratification is still low and requires further validation. Unless the authors can demonstrate a novel genotype/molecular subtype-metabolic dependency interaction that can be validated in vivo this study would be of limited scope and translational potential.

In this second revision, we have provided further validation of the proposed stratification and identified phenotype to genotype relationships. We hope that the reviewer's concerns and comments are now sufficiently addressed as detailed below.

For comment 1)

Figure 2c. The authors mention that the method used is hierarchical complete linkage clustering. However, this computational method merges different clusters based on shortest distance between observations. While Group I, II, III and V make sense with this approach based on a selected threshold, group IV cannot be justified solely based on the distance between the cell lines. KG1 and K562 cell lines would be closer to Groups I, II and III with this approach. The robustness of the dendrogram is not the issue. On the other hand, the numbers of clusters identified requires further validation. Using a method such as the Average silhouette or Elbow method would answer this question. In addition, the authors should use another unsupervised computational method e.g. non-negative matrix factorization (NMF) to validate A) the optimal number of clusters, B) the overlap with the hierarchical clustering, and C) the drugs that contribute the most to the stratification.

We thank the reviewer for prompting us to further explore the clustering of the cell lines based on metabolic vulnerabilities. In the previous revision round, we did state that "some samples clearly match the subtyping pattern better than others". Hence, we do acknowledge that cluster IV (KG-1, K-562, and BV-173) is not ideal, given that BV-173 could also be on its own. This was also evident from the bootstrapping and robustness analysis we performed already where this cluster had the lowest reproducibility (71% (BV-173); 77% (K-562 and KG-1). However, upon deeper analysis we identified that drug response profiles of 38 drugs out of the 77 analyzed (49%) were fairly concordant either in sensitivity or not sensitivity (see Rebuttal Figure 1 below).

Rebuttal Figure 1. A Venn diagram depicting overlapping and non-overlapping metabolic drug responses among cell lines falling in cluster IV (KG-1, K-562, and BV-173).

We next evaluated the optimal number of clusters with the elbow and non-negative matrix factorization (NMF) methods as suggested by the reviewer, using the *factoextra* and *NMF R*-packages, respectively. Both the elbow method and several measures of the default NMF algorithm (brunet; Brunet et al. PNAS 2004) indicated that 4 clusters would be an optimal cluster number for our dataset (Rebuttal Figure 2).

Rebuttal Figure 2. Optimal number of clusters for the myeloid leukemia cell lines based on metabolic drug vulnerabilities as determined by the elbow method (*factoextra R* package) or brunet non-negative matrix factorization algorithm (*NMF R* package). The elbow method picture shown used *hcut* as clustering function and within cluster sums of squares (wss) method. Same results were obtained with different clustering functions such as *kmeans* or *cluster::pam*. The NMF rank survey shown above depicts several measures provided in the *NMF R* package based on which the *r* (optimal number of clusters) could be determined. For the cophenetic coefficient (a metric of how closely a dendrogram conserves the pairwise distances between the original unmodeled data points), it is suggested to take the first value where the cophenetic coefficient starts to decrease (in our case the highest cophenetic coefficient value corresponds with rank 4). Similarly for the dispersion and the consensus silhouette coefficients. In contrast, for the residuals and *rss* it is proposed to take the first value where the curve exhibits an inflection point (Hutchins et al. Bioinformatics 2008).

We then visualized the consensus matrix with rank coefficient 4 with a heatmap, which displayed the NMF-proposed cell line clusters (Rebuttal Figure 3). Majority of cell lines (12/15 80%) retained their clusters as provided in the initial and revised manuscripts based on Euclidean distance and complete linkage clustering method. The cell lines that clustered differently based on NMF were Mono-Mac-6 (now clustered with THP-1 and SHI-1), HL-60 (now clustered with MV4-11 and MOLM-13), and KCL-22 (now clustered with K-562, KG-1, and BV-173). Next, we evaluated how well the NMF-based clustering fitted the metabolic drug sensitivity dataset. We specifically compared the drug response similarity of cell lines within a cluster in relation to their original cluster (Rebuttal Figure 4) by counting drugs with standard deviation of the AUC ≤ 0.051 as concordant.

For instance, MOLM-13 and MV4-11 have a concordant drug sensitivity profile to 54/77 (70%) drugs, whereas MOLM-13, MV4-11, and HL-60 have 42/77 (55%) drug responses in common with 29 (54%; original cluster I) and 17 (40%; NMF cluster 4) compounds exhibiting sensitivity. Moreover, HL-60 cells display differential sensitivity to 19 compounds in relation to MOLM-13 and MV4-11, 6 of which significantly contributed to the original clustering (5-fluorouracil, econazole, Torin1, lestaurtinib, PF-02545920, and GW 4064). In comparison, HL-60 exhibits differential sensitivity to 7 compounds in relation to the remaining members of original cluster III (ML-2, HEL, NOMO-1) with only 1 contributing to the original clustering (LY-294002).

Similarly, SHI-1 and THP-1 cells have a comparable metabolic drug response profile to 56/77 (73%) compounds in contrast to 45/77 (58%) when the profile of Mono-Mac-6 is included with 17 (30%; original cluster V) and 13 (29%; NMF cluster 3) drugs exhibiting sensitivity. Moreover, Mono-Mac-6 cells display differential sensitivity to 18 compounds from SHI-1 and THP-1 cells, 7 of which significantly contributed to the original cell line stratification (indisulam, everolimus, 5-fluorouracil, lestaurtinib, daporinad, STF-31, and temsirolimus), in contrast to differential sensitivity to 12 compounds in relation to the remaining cell lines from original cluster II (KU-812, LAMA-84, and KCL-22) with 5 contributing to the original clustering (5-fluorouracil, disulfiram, lestaurtinib, daporinad, and STF-31).

Lastly, KCL-22 displays analogous drug responses to 36/77 (47%) metabolic agents with K-562, KG-1, and BV-173 in comparison to 38/77 (49%) of original cluster IV alone with 12 (33%; NMF cluster 1) and 17 (45%; original cluster IV) compounds affecting cell viability. In particular, KCL-22 has a differential response to 4 compounds with respect to original cluster IV and to 10 compounds in relation to the remaining original cluster II cell lines, one of which influenced the original clustering in each instance (disulfiram and everolimus, respectively). Moreover, the original cluster IV cell lines are also grouped together based on the NMF method thereby further supporting our initial clustering.

Rebuttal Figure 3. A heatmap of the consensus matrix obtained from 100 random runs of the brunet NMF method with the NMF R package.

Rebuttal Figure 4. Venn diagrams depicting similarity and dissimilarity in metabolic drug sensitivity between original Clusters I, IV, and V and NMF-proposed clusters 1, 3, and 4.

Next, we performed one-way ANOVA analysis comparing the metabolic drug responses between the NMF-proposed clusters to identify the drugs that contributed most to the stratification. This analysis revealed 15 compounds contributing to the stratification, 10 of which also contributed to the original stratification (Rebuttal Figure 5). However, after multiple testing correction with an FDR of 10% only daporinad remained significant.

Rebuttal Figure 5. Heatmap illustrating the functional grouping of myeloid cancer cells in 4 taxonomic groups based on metabolic vulnerability profiles according to the NMF brunet method. Fifteen compounds contributed to the functional stratification of the myeloid cancer cells lines (an ordinary one-way ANOVA analysis * $P \leq 0.05$; ** $P \leq 0.01$; *** $P \leq 0.001$; FDR of 10% was deemed significant; * $Padj \leq 0.1$; ** $Padj \leq 0.05$; ns not significant).

Based on all of the findings from the clustering evaluation analysis, we believe that our original clustering optimally represents the obtained metabolic drug sensitivity data. Therefore, we would not revise the cell line stratification based on the NMF output given that this method appears to prioritize the mean drug response per cluster without consideration of the consistency of a particular drug response within a cluster. Nevertheless, we have now added a new Supplementary Figure 2b depicting the overlapping and non-overlapping drug responses of our original clusters (see below).

- i. See new Supplementary Fig. 2b
- b**

- ii. See new Supplementary Fig. 2b Legend: Venn diagrams depicting concordant and discordant drug responses within the functional taxonomic clusters defined in Fig. 2c.
- iii. See referral to figure in main text lines 166-169: While each cell line had a unique metabolic vulnerability profile, the activity of 18 compounds in particular stratified the cells lines into five robust and distinct functional groups (Fig. 2c and Supplementary Fig. 2a,b).
- iv. See updated Methods section pertaining to this analysis lines 611-614: Similarity of drug sensitivity was evaluated by SD of AUC values for a particular compound within a cluster with $SD \leq 0.051$ indicating concordance. The results were visualized with Venn diagrams with the Venn webtool from the University of Gent (<http://bioinformatics.psb.ugent.be/webtools/Venn/>).

For comment 2) While the Authors mention in line 255 that they identify novel correlations between mutational status and sensitivity to a drug, all this information is correlative and there are no data to support that the results are not cell line specific, nor an in vitro artefact. Validating these findings with a few experiments like those that were performed to address comment 3) would provide higher impact to this study and potentially identify new therapeutic targets of high relevance in myeloid leukemia. For example, using isogenic models or overexpressing FLT3 in a cell line with WT-FLT3 and testing its sensitivity to both known and newly identified compounds could serve to validate the correlations identified. The same could be done for RAS or TP53 that the authors describe in lines 258-260. These models could also

be used to assess whether any of these driver mutations could drive the stratification of the different clusters, i.e. whether the introduction of a genetic alteration in a WT cell line setting could change its classification from one cluster to another. Finally, given that the screen was performed in unphysiologically relevant media, and it aims to identify metabolic vulnerabilities, this study would be of limited scope and translational potential unless the major findings are validated in vivo.

The reviewer is right that findings presented in Figure 3b and Supplementary Fig. 5 are correlative in nature. Nevertheless, they can still have translational relevance. We would like to emphasize again that in vivo validation of certain phenotype to genotype correlations would be out of the scope of the manuscript and the key message of the paper in agreement with the editor. Instead, we have opted to cross-validate some of the phenotype to genotype associations with publicly available cell line and patient sample datasets to further test the findings. To that end, we utilized the following datasets: Genomics of Drug Sensitivity in Cancer (GDSC1) and Iorio et al., Cell 2016; Malani et al., Leukemia 2020, and the Beat AML study (Tyner et al., Nature 2018 (data retrieved from <http://www.vizome.org/aml/>)). This analysis showed that the findings of this study could be cross-validated with independent datasets pertaining to FLT3 mutations and 5-fluorouracil and FLT3-inhibitor sensitivity, RAS and TP53 mutations and reduced sensitivity to mTOR inhibitors, and PI3K and topoisomerase I inhibitors, respectively (Rebuttal Fig. 6). Moreover, we have now added the AML patient samples data from the Beat AML study as a new Fig. 3c, which highlights that translational potential of some of our findings.

FLT3 mutations

Data from Genomics of Drug Sensitivity in Cancer (GDSC1) & Iorio et al; *Cell* 2016 (AML and CML cell lines)

Data from Malani et al; *Leukemia* 2020 (AML cell lines)

Data from Tyner et al; *Nature* 2018 & vizome.org (AML patient samples)

Scoring here is opposite to above, higher values indicate lack of sensitivity and lower values indicate sensitivity

RAS mutations

Data from Genomics of Drug Sensitivity in Cancer (GDSC1) & Iorio et al; *Cell* 2016 (AML and CML cell lines)

Data from Malani et al; *Leukemia* 2020 (AML cell lines)

Data from Tyner et al; *Nature* 2018 & vizome.org (AML patient samples)

Scoring here is opposite to above, higher values indicate lack of sensitivity and lower values indicate sensitivity

TP53 mutations

Data from Genomics of Drug Sensitivity in Cancer (GDSC1) & Iorio et al; *Cell* 2016 (AML and CML cell lines)

Data from Malani et al; *Leukemia* 2020 (AML cell lines)

Data from Tyner et al; *Nature* 2018 & vizome.org (AML patient samples)

Rebuttal Figure 6. Cross-validation of selected phenotype to genotype associations. Box-plots depicting sensitivity of several inhibitors that have significantly higher or lower sensitivity in FLT3, RAS, and TP53 mutant versus wild-type AML and CML cell lines and patient samples. Error bars signify mean \pm SD and the difference in response was assessed with either a two-tailed unpaired T or Mann-Whitney test (* $P \leq 0.05$; ** $P \leq 0.01$; *** $P \leq 0.001$; **** $P \leq 0.0001$). Higher values of AUC from GDSC1 and Iorio et al. and of DSS (drug sensitivity score; Yadav et al., *Sci Rep* 2014) indicate sensitivity and lower values indicate lack of sensitivity. This is the other way around for AUC values from the Beat AML study. FKBF12 – FK506-binding protein 12; TS – thymidylate synthase; i – inhibitor; CPT – camptothecin.

i. See new Fig. 3c

c Data from Tyner et al; *Nature* 2018 & vizome.org (AML patient samples)

ii. See new Fig. 3c figure legend: **c**, Validation of selected phenotype to genotype associations (where data available) from the Beat AML study³³ (data retrieved from <http://www.vizome.org/aml/>). Box-plots showing sensitivity of several metabolic modifiers that have significantly higher or lower sensitivity in *FLT3*, *RAS*, and *TP53* mutant versus wild-type AML patient samples. Higher values, here, indicate lack of sensitivity and lower values indicate sensitivity. Error bars signify mean \pm SD and the difference in response was assessed with either a two-tailed unpaired *T*- or Mann-Whitney test (* $P \leq 0.05$; ** $P \leq 0.01$; *** $P \leq 0.001$; **** $P \leq 0.0001$).

iii. See modified main text lines 259-261: Several of these associations could be confirmed in an independent AML patient sample dataset³³ (Fig. 3c).

For comment 3)

Comment has been addressed. Validation of drug-response associations in cell lines highlights the discrepancy between predicted and real response to combination therapy but also provides a good tool for synergistic drug response discovery.

This issue has been resolved in the previous revision round.

For comment 4)

Comment has been addressed and results of follow-up experiments show the non-specificity of the response.

This issue has been resolved in the previous revision round.

For comment 5)

While the dose response curves have been added for the 5 drugs mentioned, it would be useful to demonstrate if this sensitivity to FASN inhibitors is cell line dependent or is

observed across all the cell lines of cluster III compared to cells from other clusters. Even if non-significant across different clusters using the initial concentrations, the cluster containing HL60 might still show higher sensitivity to FASN inhibitors and therefore demonstrate a higher impact of the classification.

The individual dose responses curves of inhibitors targeting fatty acid synthesis related enzymes illustrate that the dependency is cell line dependent and not cluster dependent. From the graphs below (Rebuttal Fig. 7) one can deduce that HL-60 cells and to a lesser extent MOLM-13 cells exhibit vulnerability to FASN inhibition, whereas MOLM-13 cells display marked susceptibility to SCD1 inhibition. This was highlighted in the spider plot in Fig. 5a that depicted the differential metabolic drug responses observed in each tested cell line. In contrast, the other cell lines falling in group III (NOMO-1, HEL, and ML-2) or group I (MV4-11) do not exhibit sensitivity to these inhibitors (Rebuttal Fig. 7).

Rebuttal Figure 7. Sensitivity to inhibitors targeting fatty acid synthesis metabolism. Dose response, AUC, and mean dose response data of GSK2194069, MF-438, MK-8245, and PluriSin 1 in the myeloid leukemia cell lines with the dose response curves and bars colored per the group the cell lines were classified to. Error bars indicate SD either of technical replicate (top panel) or other cell lines belonging to the same group (bottom panel).

For comment 6)

Comment has been addressed. New data indicate that PIK3CA inhibition (albeit at concentration that largely affects cell viability) reduces FASN expression at the mRNA level. It would be expected that KCL-22, the only cell line with PIK3CA mutation would also be more sensitive to FASN inhibition and/or show synergism with pictisilib.

This issue has been resolved in the previous revision round.

Minor comment: Fig. 5 has been relabelled fig.4 in the revised manuscript.

Thank you for pointing this issue out, we apologize for the oversight. In the revised manuscript all the figures have been relabeled accordingly.

Reviewer #3 (Remarks to the Author):

I thank the authors for their detailed responses to my previous comments. Clearly a great amount of effort has gone into the initial and revised manuscripts. However, I still find several claims to be unsupported by the data.

We thank the reviewer for appreciating our work and the response to the previous revision round. We have now provided further clarifications and support of our claims that we hope would be acceptable for the reviewer.

In the first submission, the authors claimed that lower overall expression of PI3K genes formed the basis of the heightened sensitivity of HL60 cells to pictilisib. I suggested that to make this claim would require overexpressing PI3K genes in HL60 cells to test whether sensitivity is diminished. The authors have attempted to do this but have not found that the low expression of PI3K genes is responsible for the increased sensitivity in HL60s. I have several concerns with these new data.

First, in the new supplementary 7d,e, HL60 empty vector control cells are no more sensitive than MV4-11 empty vector control cells to pictilisib suggesting that the heightened sensitivity of HL60 observed in Fig 4b may not be reproducible.

During the revision process we have retested the sensitivity to pictilisib in HL-60 cells several times and found that the sensitivity originally observed is reproducible (Rebuttal Fig. 8). The sensitivity is also in accordance to previously published data as we have shown in Supplementary Figure 7. This observation of HL-60 empty vector controls not being more sensitive than MV4-11 is likely due to the cells not adapting very well following transfection.

Rebuttal Figure 8. Biological replicates of pictilisib dose response data in HL-60 cells in two or three technical replicates (original screen, rep1 – start of first revision, rep2 – during PI3K knock out and over expression experiments). Error bars indicate SD.

Additionally, the authors have included sensitivity data for HL60 overexpressing PIK3CA and PIK3CG, but neither of these genes appear to be successfully expressed by western blot.

Given that the same constructs were used in both MV4-11 and HL-60 cells, it appears that in HL-60 cells PIK3CA and PIK3CG are modified postranslationally and their expression cannot be detected successfully by western blot. However, as we detected changes in pictilisib sensitivity in HL-60 cells overexpressing different PI3K gene variants in relation to empty vector controls it is plausible that overexpression of these genes occurred.

Finally, if lower expression of PI3K genes is responsible for increased sensitivity in HL60 cells, then how would the authors explain the left-shifted curves for HL60 overexpressing PIK3CA, B, and C? If anything, this would suggest lower expression would diminish sensitivity to pictilisib, not increase it. Nevertheless, the authors do not acknowledge that low PI3K expression conferred increased pictilisib sensitivity appears to be incorrect.

Given those findings, we had modified the manuscript to state that we first hypothesized that lower expression of different PIK3 genes is responsible for the increased sensitivity in HL-60 cells. However, upon mechanistic exploration, this hypothesis, as the reviewer rightly points out, did not appear to hold. Thus, in the previous rebuttal letter we already indicated that we have removed the statement from the original submission that this feature could be used to select patients for PI3K inhibitor therapy. Since we observed different PIK3 genes influencing pictilisib sensitivity in HL-60 and MV4-11 cells, we postulated that there are different isoforms on which sensitivity to pictilisib is dependent in these two cell lines and that no one specific PI3K isoform influences sensitivity to pictilisib. Thus, it is plausible that a combination of isoforms determines pictilisib susceptibility, and this is technically challenging to model and investigate. In the revised manuscript we now stated more clearly that our original hypothesis, that lower expression of PIK3 genes determines pictilisib sensitivity, is not correct.

- i. See modified main text lines 315-317: Thus, we initially hypothesized this to likely explain the vulnerability of HL-60 cells to PI3K inhibition as pictilisib is not effective in inhibiting PIK3C2A³⁹.
- ii. See modified main text lines 327-330: These findings suggest that our original postulate was incorrect and that there may not be one specific PI3K isoform that influences sensitivity to pictilisib in these cells. Rather, it is likely that the combined activity of several isoforms may determine pictilisib susceptibility.

To further explore the sensitivity profile of pictilisib we performed additional analysis of publicly available gene essentiality (Tzelepis et al; Cell Rep 2016), gene expression (DepMap; 20Q2 data release), reverse phase protein array (CCLE; Ghandi et al; Nature 2019), and drug sensitivity (GDSC1) data in HL-60, MV4-11, and MOLM-13 cells on the MAPK and PI3K/mTOR pathway. Across all four datasets HL-60 cells clustered separately than MV4-11 and MOLM-13 (new Supplementary Fig. 7f-i). All three cell lines carry activating mutations in genes involved in growth factor signaling, NRAS (HL-60) and FLT3-ITD (MV4-11 and MOLM-13). HL-60 exhibits higher dependency and activity on the MAPK pathway as supported by enhanced essentiality to NRAS, RAF1, and RPS6KA1 and higher RPPA signal to N-ras, MEK1, ERK2, RSK1/2/3, p90RSK, and phospho B-Raf and C-Raf. Moreover, HL-60 cells appear to have increased reliance on the mTORC2 complex and its signaling given the dependency on MAPKAP1 (mSin1), MLST8, and partially RICTOR; higher expression of TSC1/2 and RPS6, lower expression and RPPA signal of several mTORC1 complex members and its downstream effectors; and greater RPPA signal for AKT and AKT_pS473 (indicative of mTORC2 complex activity and signaling), TSC1, eIF4E, and total and phospho PKC $\alpha/\beta/\delta$ (see Supplementary Fig. 7f-h).

Last but not least, as it was also shown in this manuscript HL-60 cells are less sensitive to both rapalogs (indirectly targeting mTORC1) and ATP-competitive mTOR inhibitors (targeting mTORC1 and mTORC2) in line with the finding that RAS mutation confer reduced sensitivity to mTOR inhibitors. With drug sensitivity data from the GDSC1 resource, we confirmed this and also show that HL-60 cells are also less sensitive to pan-AKT inhibitors, but display strong sensitivity to pan-PI3K (Supplementary Fig. 7i). Sensitivity patterns to PI3K isoform specific inhibitors did not provide insights into

which isoform is responsible for the selective sensitivity to HL-60 cells. However, what could be a plausible scenario is that the more pronounced mTORC1 activation in MV4-11 and MOLM-13 cells in comparison to HL-60 cells, resulted in lower sensitivity to pictilisib in these cells. mTORC1 activation has been described to be a crucial event in the resistance to PI3K inhibitors in many tumor types, possibly due to its function downstream of PI3K, as cancers that have residual mTORC1 activity do not respond strongly to PI3K inhibition (Ilagan E; Trends Cancer 2016; Elkabets M; Cancer Discov 2013). A recent study looking at determinants of drug sensitivity and resistance showed that gene expression data does not significantly predict sensitivity to pictilisib (Tognetti et al; Cell Syst 2021). They found that signaling models based on multiplex single-cell mass cytometry data were more predictive with S6K activation by mTOR, STAT1/3 activation by EGFR and SRC, and cellular variability of MKK4, p90RSK, and MKK3/6 determining sensitivity to pictilisib. Thus, the mechanism of sensitivity to pictilisib is clearly more complex than reliance on a single gene/protein, but rather is multifaceted and context-dependent. We have now modified the main text and discussion to reflect these arguments.

iii. See new Supplementary Fig. 7f-i

iv. See new Supplementary Fig. 7f-i legend: f-i, Gene essentiality³⁷, gene expression⁴⁰, reverse phase protein array (RPPA)²⁹, and drug sensitivity data³⁶, respectively, related to the PI3K/AKT/mTOR signaling in HL-60, MV4-11, and MOLM-13 cells.

- v. See modified main text lines 330-338: Comparison of gene essentiality³⁷, gene expression⁴⁰, reverse phase protein array (RPPA)²⁹, and drug sensitivity data³⁶ in HL-60 cells and cluster I cell lines (MV4-11 and MOLM-13) revealed that HL-60 cells exhibit higher dependency and activity on the MAPK pathway and mTORC2 complex signaling resulting in reduced sensitivity to AKT and mTOR inhibitors (Supplementary Fig. 7f-i). These features corresponded with lower gene expression and RPPA signal of several mTORC1 complex members and its downstream effectors, suggesting that mTORC1 activation status influences sensitivity to pictilisib as shown previously in the context of breast cancer^{41,42}.
- vi. See modified discussion lines 495-500: The sensitivity of the PI3K inhibitor pictilisib could be associated with lower activation of the mTORC1 complex, as mTORC1 activation has been described to be a crucial event in the resistance to PI3K inhibitors in many tumor types^{41,42}. A recent study found that signaling models based on multiplex single-cell mass cytometry data were more predictive of pictilisib sensitivity than gene expression alone in breast cancer cell lines⁵⁷.

REVIEWER COMMENTS

Reviewer #1 (Remarks to the Author):

The authors have adequately addressed most of my comments and this paper is worthy of publication.

REVIEWER COMMENTS

Reviewer #1 (Remarks to the Author):

The authors have adequately addressed most of my comments and this paper is worthy of publication.

We thank the reviewer for the constructive comments to the manuscript.